

# Quantifying the Impacts of Compound Extremes on Agriculture and Irrigation Water Demand

Iman Haqiqi[1], Danielle S. Grogan[2], Thomas W. Hertel[1,3], and Wolfram Schlenker[4]

[1] Department of Agricultural Economics, Purdue University, West Lafayette, IN, USA.
[2] Institute for the Study of Earth, Oceans, and Space, University of New Hampshire, Durham, NH, USA.
[3] Purdue Climate Change Research Center, Purdue University, West Lafayette, IN, USA.
[4] School of International and Public Affairs, Columbia University, New York City, NY, USA.

*Correspondence to*: Iman Haqiqi (ihaqiqi@purdue.edu)

**Abstract.** Agricultural production and food prices are affected by hydroclimatic extremes. There has been a large literature
measuring the impacts of individual extreme events (heat stress or water stress) on agricultural and human systems. Yet, we lack
a comprehensive understanding of the significance and the magnitude of the impacts of compound extremes. Here, we combine a
high-resolution weather product with fine-scale outputs of a hydrological model to construct functional indicators of compound
hydroclimatic extremes for agriculture. Then, we measure the impacts of individual and compound extremes on crop yields
focusing on the United States during the 1981-2015 period. Supported by statistical evidence, we confirm that wet heat is more
damaging than dry heat for crops. We show that the average damage from heat stress has been up to four times more severe when
combined with water stress; and the value of water experiences a four-fold increase on hot days. In a robust framework with only
a few parameters of compound extremes, this paper also improves our understanding of the conditional marginal value (or damage)
of water in crop production. This value is critically important for irrigation water demand and farmer decision-making – particularly
in the context of supplemental irrigation and sub-surface drainage. **Keywords**. agriculture; climate impacts; water balance model;
extreme heat; extreme drought.

## 1 Introduction

In this paper, we quantify the response of agricultural production to individual and compound extremes. This study employs high-resolution daily estimates of soil moisture from a hydrological model, in combination with fine-scale daily weather data and a county-level panel of crop yields across the United States (US). By considering metrics of individual and compound extremes in
a statistical framework from the literature estimating climate impacts on agriculture, we explore the importance of compound hydroclimatic extremes versus individual extremes. We show that the compound-extremes approach provides a significantly better prediction model compared to the individual-extreme approach that is currently employed in the literature. The proposed framework also allows us to estimate the conditional marginal value of water (economic water demand) in agricultural production – a key metric for producers making supplemental irrigation decisions. We show the implications of our findings for irrigation
water demand and subsurface drainage. This paper adds important nuances to the climate-agriculture-water links and therefore helps to refine projections of future impacts of anthropogenic climate change.

We construct various indicators of individual and compound extremes appropriate for agricultural studies. In agricultural production, water and heat extremes are key determinants of yield. They affect agricultural yields, farm revenues, and crop markets. The relationship between extreme heat, cumulative seasonal precipitation, and crop yields has been well-documented, particularly
across the US and particularly for corn (Schlenker and Roberts, 2009; Urban et al., 2012; Diffenbaugh et al., 2012; Roberts et al., 2013; Lobell et al., 2013; Urban et al., 2015; Wing et al., 2015; Burke and Emerick, 2016). However, the indicators of water



conditions are either mean or cumulative measures calculated over the growing season or stages of crop growth. Other metrics of extreme water conditions have been only minimally explored (Fishman, 2016). Current statistical studies had limited success in statistically capturing the yield response to soil moisture metrics (Bradford et al., 2017; Peichl et al., 2018; Siebert et al., 2017).

The recent work by Ortiz-Bobea et al. is a notable exception. It highlights the importance of mean soil moisture metrics for estimating crop yields in the US (Ortiz-Bobea et al., 2019). However, it ignores the daily interaction of soil moisture and heat and the significant role played by soil types.

In the physical sciences, researchers have discussed the impact of climate change on soil moisture (Feng and Zhang, 2015; Jung et al., 2010; Marshall et al., 2015; McDonald and Girvetz, 2013; Rodell et al., 2018; Taylor et al., 2013). However, a key unknown

is the extent of the benefits of soil moisture in buffering the heat damage to yields. Despite existing theoretical frameworks and controlled experiments, we currently lack a comprehensive understanding of the conditional marginal impact of heat on yields while controlling for water (Bradford et al., 2017; Ortiz-Bobea et al., 2019). The problem is that current studies tend to separate the impact of heat from water stress. There is no robust predictive framework that captures the implications of daily interactions of soil moisture and heat in the determination of national crop yields. Also, the current literature is focused mainly on the impacts

of dry-heat and ignoring the impacts of wet-heat stress. In short, standard measures of heat and water stress are missing important temporal, spatial, and vertical dynamics. In this paper, we investigate the significance of the compound heat and water conditions in predicting crop yields including dry-heat and wet-heat. We focus on corn as the major field crop in the US. We also compare the indicators of compound extremes versus individual extremes (i.e. only heat stress or only water stress).

Technically, we extend the models in Schlenker and Roberts (2009) and Ortiz-Bobea et al. (2019) by assuming the growth effects

of heat and water are mutually interdependent. The model captures the impacts of compound extremes (e.g. hot-dry or hot-wet conditions) as well as individual extremes (excess heat, excess water, and water deficit). We use detailed soil moisture information available from recent developments in the Water Balance Model (Grogan, 2016; Wisser et al., 2010), hereafter WBM. We show that the coefficient on excess heat in the estimation of corn yield is significantly different when we consider the daily interaction with soil moisture. This study also demonstrates the advantages of using soil moisture metrics over current proxy variables in

capturing climate-driven variations in heat and moisture availability. We find that the soil moisture index, and its daily interaction with heat, perform better in predicting corn yields compared to the commonly used proxy variables such as cumulative precipitation. Specifically, we estimate 1) the marginal impacts of heat stress on crop yields; 2) the marginal impact of daily soil moisture extremes on crop yields, and 3) the conditional marginal impact of heat and soil moisture on crop yields.

This study improves our understanding of the value of water management in crop production. It contributes to the socio-hydrology

literature by providing a robust framework for studying the climate-agriculture-water links (Ertsen et al., 2013; Fernald et al., 2015; Van Emmerik et al., 2014; Di Baldassarre et al., 2019). It also serves to bridge the gap between statistical studies of climate impacts on crops and their biophysical counterparts. Understanding the true value of water management for agriculture is critical in the face of a warming climate. Fluctuations in precipitation can lead to drought or flooding. They account for more than 70% of crop indemnities in the US during the 2001-2015 period (USDA-RMA). As economic agents, farmers may choose to adjust to climate

change depending on the likely benefits and costs of alternative options. Current studies suggest a significant impact from climate change on rainfed agriculture (Ortiz-Bobea and Just, 2013; Annan and Schlenker, 2015; Liu et al., 2017; Sesmero et al., 2017; Hsiang and Kopp, 2018; McCarl and Hertel, 2018). This study sheds light on the benefits of adaptation options including full-scale irrigation or supplemental irrigation by showing how water can reduce heat damages to crops. Although converting to complete irrigation is sometimes an attractive solution, a more challenging question involves the likely benefits and costs of supplemental

irrigation. While the biophysical information necessary for these calculations is offered, at least in part, by the hydroclimatic, biophysical, geospatial, earth, and atmospheric sciences, this study transforms this information into economic terms that are useful



for both farmers and policymakers. In this paper, we will show how the results can be used to economically quantify the marginal value of water, in the form of soil moisture, for corn production in the US under different hydroclimatic conditions.

The remainder of this paper is organized as follows. The next section provides an overview of the empirical concerns in this type
of study. Then we introduce two models to explain the significance of soil moisture for estimating crop yields. We also describe different data sets used in the study. Section 4 provides estimation results. Section 5 contains a discussion on the implications of the findings for climate impact research. And Sect. 6 concludes.

## 2 Empirical concerns

In this section, we will review some of the empirical concerns in investigating the individual and compound impacts of heat and
water on crop yields in statistical models. Before starting our discussion, we will briefly describe the basic model as introduced by Schlenker and Roberts (2009). The model assumes that the effects of heat on corn yields are cumulative over the growing season. In other words, the end-of-season yield is the integral of daily heat impacts over the growing season. This relationship can be demonstrated via Eq. (1):

$$y_{it} = \int_{\underline{h}}^{\overline{h}} g(h)\varphi_{it}(h)dh + z_{it}\boldsymbol{\delta} + c_i + \epsilon_{it} \tag{1}$$

where $\varphi_{it}(h)$ is the time distribution of heat ($h$) over the growing season in county $i$ and year $t$, while the heat ranges between the lower bound $\underline{h}$ and the upper bound $\overline{h}$; indicators of water availability and other control factors are denoted as $z_{it}$, and $c_i$ is a time-invariant county fixed effect.

When using high-resolution data on heat and water, there are empirical concerns regarding the data generating processes as well as estimation itself. Here we will briefly talk about choosing the variables, the level of aggregation, endogeneity issues,
measurement errors, model specifications, and standardization.

### 2.1 Degree of spatial and temporal aggregation

The impact of water on crop production depends on local hydroclimatic conditions. There is considerable heterogeneity in the geographical distribution of water resources and the location of producers. There is also significant spatial heterogeneity in soil properties, which modulate temperature and precipitation signals through their capacity to hold water as soil moisture. It is
important to have a sample with extreme conditions and a pattern of spatial heterogeneity which is representative of the entire market (population).

An important aspect of estimation is the spatial scope of the study. A simple spatial aggregation can eliminate the extreme conditions (tails of the distribution) from the sample. On the other hand, farm-level data is limited to specific locations. Although estimates based on geographically limited observations can be informative for those locations, a more comprehensive analysis of
yield response to climate is necessary for market predictions (Kucharik, 2003; Kudamatsu, 2018; Martin, 2018). We construct our database at the county level in the US. To ensure a correct data generating process, the hydroclimate data are constructed such that nonlinear transformations are taken at the grid cell level before being aggregated across time and space (Hsiang, 2016). The county-level aggregation has a couple of benefits. First, the corn yield data reported by USDA at the county level and we do not need to have heavy data processing on the yield data. Also, this will provide enough heterogeneity for local market analysis and local
climate impact reports.

Another empirical challenge is that yields data are reported annually, while weather data have a higher temporal resolution (e.g. hourly, daily, or weekly). Consequently, some empirical studies employ annual or monthly weather indicators like average



temperature. However, many studies utilize daily climate information by introducing growing degree days and harmful degree days through the growing season (D'Agostino and Schlenker, 2016; Mueller et al., 2012). One standard solution is the use of the

growing degree days approach along with an index of cumulative rainfall to proxy for water availability as in Schlenker and Roberts (2009); or as mean monthly or seasonal soil moisture in Ortiz-Bobea et al. (2019).

However, cumulative indices, monthly mean, or seasonal average metrics do not capture extreme events during the season (e.g. early-season floods and late-season droughts can cancel out when taking the average). The mean variable can be misleading as the plants respond to day to day variability. Furthermore, the mean water index may not represent hydrological extremes (D'Odorico

and Porporato, 2004; Lobell and Burke, 2010; Schaffer et al., 2015; Werner and Cannon, 2016). While the average conditions are important, exposure to extreme water stress can cause permanent unrecoverable damage to the plant (Denmead and Shaw, 1960). In addition, too much water can cause flooding, waterlogging, or may wash out soil nutrients and fertilizers (Kaur et al., 2018; Schmidt et al., 2011; Urban et al., 2015). Therefore, it is necessary to introduce indicators of extreme soil moisture stress. This will be even more important in the future, as climate scientists are predicting more extreme drought and precipitation events (Myhre et

al., 2019). In other words, mean variables can create biases in future climate impact analysis by ignoring the extreme events. It is important to introduce different metrics of daily water availability to measure the value of water at the time which is most needed. Panel (a) in Fig. 1 visualizes four soil moisture conditions that are unfavorable for crop yield. Both too much water [i] and intense moisture stress [ii] can cause severe damage to crop yields. Similarly, a long period of mild moisture stress [iii] or a short period of severe moisture stress [iv] can also cause significant yield loss. These measures can help to understand the need for artificial

drainage or irrigation as shown in panel (b).

## 2.2 Water availability index

While soil moisture plays a crucial role in determining climate impacts on agricultural yields, there have been only a few successful statistical studies in measuring this relationship. Many researchers have acknowledged the need for soil moisture data to predict the response of crop yields to variations in water availability. Some studies also highlight the need for irrigation to compensate for

soil moisture deficits (Li et al., 2017; McDonald and Girvetz, 2013; Meng et al., 2016; Williams et al., 2016). One barrier has been limited availability of daily fine-scale soil moisture data and inconsistency of soil moisture data with heat information. It has become a standard practice for current studies either to focus on a limited geographical area (Rizzo et al., 2018; Wang et al., 2017) or to employ a proxy variable like precipitation, evapotranspiration, or vapor pressure deficit estimates (Comas et al., 2019; Roberts et al., 2013). While cumulative precipitation is significant in previous studies, it may not be a good representation of available

water for plants in many places due to irrigation, runoff, or evaporation. Indeed, it is only relevant if the precipitation is stored in the soil for plant use during the season. Following these studies, one might be able to estimate the marginal impact of change in "mean precipitation" or "mean evapotranspiration". However, this will not necessarily provide appropriate coefficients for future climate impacts as the distribution of precipitation across space and time is estimated to change, leading to more frequent extreme events (Myhre et al., 2019). To undertake climate impact analyses of water availability required further information.

In this study, we will show that, although cumulative precipitation and mean soil moisture are correlated, their performance can be different in predicting corn yields. We will show how an empirically validated, high-resolution hydrological model, such as WBM, can provide valuable information for estimating the marginal value of water.

## 2.3 Interaction of soil moisture and heat

To accurately measure the marginal impact of soil moisture, we need to draw on biogeochemistry, hydrology, and plant physiology

perspectives on crop yields and soil moisture. We treat soil moisture as an integrative variable that contains information on



precipitation, temperature, and soil types, as well as the behavior of the crops themselves. Crop yields depend on daily growth during the season (Hatfield and Prueger, 2015). Plants require water for germination, transpiration, nutrient transport, and to buffer against temperature fluctuations (Maharjan et al., 2016; Teixeira et al., 2014). Therefore, timely irrigation can play an important role in boosting yields (Carter et al., 2016; Siebert et al., 2017; Tack et al., 2017; Troy et al., 2015).

However, the growth effects of heat and soil moisture are mutually interdependent. Beneficial heat is less beneficial without sufficient soil moisture. On the other hand, soil moisture is not beneficial without sufficient heat for plant growth. Harmful heat can be less harmful when there is enough soil moisture (Hauser et al., 2018). While the amount of daily water requirement depends on the biophysical properties of soil and crop, it changes with temperature, solar radiation, humidity, and wind speed. In this framework, climate change can affect both soil moisture supply and demand by altering the abundance and frequency of
precipitation and by increasing the water required to compensate evapotranspiration and evaporation. If the temperature is high and there is not enough soil moisture for a long period (drought conditions), this may cause severe damage to crops (Denmead and Shaw, 1960). Therefore, consideration of the daily interaction of soil moisture and heat is necessary to capture the impacts on natural supply and plant demand for soil moisture.

**2.4 Importance of soils**

In a large literature in the statistical estimation of corn yields in the US, water availability is represented by cumulative precipitation and its square term in a fixed effect panel regression. The estimated coefficients suggest a positive impact from cumulative precipitation and a negative impact from its square term. This leads to a universal optimum precipitation level, $\hat{p}*$, which is the same for all the observation locations. However, this is not necessarily equal to the true optimum level of water for production in each location. According to the agronomic literature, the optimum amount of water depends on the moisture stored in the soil, soil
type, and heat (Fang and Su, 2019). Thus, many studies utilize other metrics of water availability including estimated evapotranspiration, standardized precipitation, and drought indices.

On the other hand, standard measures of volumetric soil moisture are not the best indicator of water availability. In the agronomic literature, the water available to plants depends on volumetric soil moisture as well as soil type. for the same volume of soil moisture, different soil types imply different wilting points and different field capacity which result in different water availability
to crops. Figure 2 shows the difference between soil moisture content, water available to plants, and unavailable water. This illustrates that sand and sandy-loam soil types have the lowest field capacity (and water availability) while clay and clay-loam have the highest. As soil moisture metrics (volumetric or fraction) vary over the space, we need to look at soil type, crop cover, and other biophysical variables. Generally, soil moisture thresholds are defined in terms of the soil available water for plants, or soil wilting point, not a constant depth of water.

As a simple solution, one can capture the differences in soil type by introducing dummy variables. However, at the county level aggregation which has been chosen by many studies, it is challenging to select a soil type for a county. While a dominant soil type can work, it is not necessarily the best option. As we prefer to take care of differences at the grid cell level before aggregation. Another solution is to standardize the soil moisture indicator. Introducing the soil moisture fraction can help as it takes the ratio of soil moisture content to the field capacity. However, interpretation of the results is not straightforward. A better measure is the soil
moisture deviation from normal. This is defined as daily soil moisture deviation from historical average soil moisture at each location. In a standard Schlenker-Roberts type model, the coefficient on this indicator would show the percentage change in corn yields in response to one mm higher soil moisture deficit (or surplus). We use deviation from normal levels as this can remove the location-specific features of soil moisture. While irrigation is taken into calculations, variation in this metric is higher in non-irrigated areas and is lower in irrigated areas as the irrigating farmers try to keep the soil moisture around a normal range.



### 2.5 Measurement errors and endogeneity concerns

While there exist remotely sensed metrics of soil moisture (e.g. NASA's Gravity Recovery and Climate Experiment or the European Space Agency's Climate Change Initiative), they are coarse in spatial and temporal resolution. Also, they are relatively new and therefore give rise to a short length for the panel data. Also, there is in situ observed soil moisture data that suffer from missing data points and requires a significant amount of interpolation as the stations are irregularly scattered in time and space (Ford and Quiring, 2019).

On the other hand, simulated soil moisture data from hydrological models can be problematic in different ways. Also, if the simulation involves a time-varying yield input, the estimations will be biased due to serious endogeneity problems. Besides, if a model employs a simulation framework based on specific parameters and functional forms, there is a likely systematic measurement error due to the correlation of unobservable determinants of soil moisture over space and time. For studies covering the continental US, encompassing both a highly irrigated West and a less-irrigated East, irrigation is also important in estimating the soil moisture. If the simulated soil moisture metric ignores the irrigation inputs, the estimation will suffer from a key omitted variable. Irrigation inputs will be correlated with the soil moisture, as irrigation water will be applied when precipitation inputs are insufficient for optimal crop growth. Thus, when soil moisture is low, irrigation is more likely to be high. This challenge is the reason that most of the literature linking corn yields to temperature and precipitation across the US has relied only on counties east of the 100th meridian, where corn is rarely irrigated.

The average soil moisture output from WBM is informed mainly by daily soil moisture memory, exogenous heat data, time-invariant crop cover, time-invariant soil features, precipitation, irrigation, and carefully calibrated parameters. Furthermore, the output from the WBM has been validated against observational data (Grogan et al., 2017). This ensures that the model performs well in replicating the observations. Also, as it includes irrigation in generating the soil moisture, it is a reliable data source for both Eastern and Western US (in which irrigation is dominant). As the model does not use yield data, the soil moisture is invariant to changes in yield. That said, we will test the performance of soil moisture in predicting corn yields.

Typically, clustering the standard errors (for example by state) is a standard practice to minimize the remaining concerns. We will also use clusters by state. In addition, we will employ the soil moisture deviation from normal, to select the bins and generate the soil moisture extreme metrics. This deviation can eliminate the likely systematic measurement errors in soil moisture data which can happen due to simulation. As discussed in Sect 2.4, we will introduce metrics for soil moisture deficit and soil moisture surplus by calculating the daily deviation from normal soil moisture levels. This will tackle two problems at the same time: choosing a water availability index that provides extreme conditions while taking into account different soil types.

### 3 Methods

In this section, we introduce two models. For each model, we consider different parameterizations of heat and water to estimate the impacts of *water* on yields of corn in the US. Model 1 considers metrics of heat stress and different measures of water availability. Model 2 considers interactions of heat and soil moisture with individual and compound extremes. For each model, we will describe the relevant variables and their measurement. After describing all the models, data sources are introduced in detail. We construct the models taking into account the fact that plants generate biomass each day using the available resources like heat and water (we assume no change in soil nutrients).





### 3.1 Model (1) cumulative precipitation, mean soil moisture, and individual extremes

For model (1), we follow Schlenker and Roberts (2009). However, we include different representations of water variables. In summary, we consider four main representations of water measures. In Model (1-a), $z_{it}$ includes cumulative precipitation from the first day of April to the last day of September as well as its square term. In Model (1-b), we consider the seasonal mean soil moisture index and its square term. This is calculated as average volumetric soil moisture as reported by WBM. We keep the same heat metrics as Model (1-a). In Model (1-c), this includes some indicators of soil moisture below or above normal levels. This will show the significance of soil moisture conditions in predicting yield. Model (1-d) includes the number of days with low soil moisture as well as the number of days with high soil moisture as other indicators of extreme soil moisture conditions. We describe these metrics in Sect. 3.3. Supplementary Materials provide further metrics including the mean evapotranspiration and the mean of soil moisture fraction. The estimation strategy is described in Sect. 3.4.

### 3.2 Model (2) compound extremes and daily interaction of soil moisture and heat

Here, we focus on the daily interaction of available water and heat as major indicators of plant growth. This will show the significance of the conditional marginal impact of heat and water on crop yields:

$$y_{it} = \int_{\underline{m}}^{\overline{m}} \int_{\underline{h}}^{\overline{h}} g(h,m)\varphi(h,m)dhdm + z'_{it}\boldsymbol{\delta} + c_i + \epsilon_{it} \tag{2}$$

where the crop growth is different for each combination of soil moisture level, *m,* and heat, *h,* other control factors are denoted as $z_{it}$, and $c_i$ is a time-invariant county fixed effect. Here, we do not separate the impact of heat from water. In other words, the marginal impact of heat depends on water; and the marginal impact of water depends on heat. We will consider different approaches to estimate this model as described in Sect. 3.4.

### 3.3 Data

In estimating the marginal impact of soil moisture on corn yields, we employ information about soil moisture, temperature, precipitation, and corn yields for counties of the United States for the 1981-2015 period as summarized in Table 1. The data on yield is obtained from USDA-NASS (United States Department of Agriculture-National Agricultural Statistics Service) at the county level. The yield is defined as the corn production (in bushels) divided by harvested area (in acres). Precipitation is defined in millimeters as accumulated rainfall during the growing season (Apr-Sep). It is calculated based on PRISM (Parameter-elevation Regressions on Independent Slopes Model) daily information at 2.5 x 2.5 arcmin grid cells over the continental US for 1981-2015. It is aggregated to each county according to cropland area weights. Daily interaction of heat and soil moisture is also calculated daily at the gridded level. Then we aggregate the metrics to the growing season and county level.

### 3.3.1 Degree days (index for heat)

Following D'Agostino and Schlenker (2015), the daily distribution of temperatures is approximated assuming a cosine function between the daily minimum and maximum temperature. Let $\bar{t} = \mathrm{acos}\left(\frac{2b - T_{max} - T_{min}}{T_{max} - T_{min}}\right)$, then degree days at each day is defined using



$$D(b) = \begin{cases} \dfrac{(T_{max} + T_{min})}{2} - b & \text{if } b \leq T_{min} \\ \dfrac{\bar{t}}{\pi}\left[\dfrac{(T_{max} + T_{min})}{2} - b\right] + \dfrac{(T_{max} - T_{min})}{2\pi}sin(\bar{t}) & \text{if } T_{min} < b \leq T_{max} \\ 0 & \text{if } T_{max} < b \end{cases}$$

where $b$ is the base for calculating degree days and can take the base values as well as critical values. we consider a piecewise-linear function to aggregate the degree days. The major assumption is that plant growth is approximately linear between two bounds. Degree days between two bounds is simply degree days above the smaller bound minus degree days above the larger

bound.

We calculate county-level seasonal degree days based on daily weather information. The weather information on daily maximum and minimum temperature are obtained from PRISM at 2.5 x 2.5 arcmin grid cells over the continental US for 1981-2015. Degree days are initially calculated for each day at each 2.5 x 2.5 arcmin grid cell during the growing season (Apr-Sep). Then they are aggregated for the whole growing season from the first day of April through the last day of September. Finally, they are aggregated

to the county level using cropland area weights.

### 3.3.2 Soil moisture (index for water availability)

Daily soil moisture content and soil moisture fraction are obtained from the Water Balance Model (Grogan, 2016; Wisser et al., 2010) based on daily simulations using PRISM data at 6 x 6 arcmin grid cells for the 1981-2015 period over the continental US. Here, we briefly describe WBM's soil moisture module. However, the model is much more complex and employs a large list of

inputs. Full documentation for WBM can be found in Wisser et al. (2010) with updates in Grogan (2016). In WBM, crop-specific soil moisture balance within each grid cell is calculated with an accounting system that tracks a location's water inputs and outputs and is limited by the soil moisture pool's water holding capacity.

$$\frac{\delta W_s}{\delta t} = \begin{cases} g(W_s)(I - PET) & \text{if } I < PET \\ I - PET & \text{if } PET \leq I \text{ and } (I - PET) < (W_{cap} - W_s) \\ W_{cap} - W_s & \text{if } PET \leq I \text{ and } (W_{cap} - W_s) \leq (I - PET) \end{cases}$$

where $W_s$ is soil moisture, $t$ is time, $I$ is the sum of all water inputs to the soil moisture pool, $PET$ is potential evapotranspiration,

and $W_{cap}$ is available water capacity. Water inputs to the soil come in the form of precipitation as rain and as snowmelt. Water intercepted by the canopy reduces precipitation reaching the soil. Here, we use the Hamon method for estimating $PET$ (Hamon, 1963; Federer et al. 1996), and $g(W_s)$ is 1 for all crops. Crop-specific potential evapotranspiration values, PETc, are calculated following the FAO-recommended crop-modeling methodology outlined in Allen et al (1998):

$$PET_c = k_c \cdot PET$$

where $k_c$ [-] is a crop-specific, time-varying scalar. Crop scalar values are from Siebert and Döll (2010), and crop maps that identify the area of each rainfed crop type within a grid cell are from the Crop Data Layer (CDL, USDA NASS, 2017). When soil moisture is insufficient for crops to extract water equal to $PET_c$, actual crop evapotranspiration is limited to available soil water volumes. Available water capacity, $W_{cap}$, is a function of vegetation-specific rooting depth, a crop-specific depletion factor, soil field capacity, and soil wilting point:

$$W_{cap} = D_c R_c (F - W_p)$$

where $D_c$ is the depletion factor for crop $c$, $R_c$ is the rooting depth of crop $c$, $F$ is the soil field capacity, and $W_p$ is the soil wilting point. Here we use the Harmonized World Soil Database (Fischer et al. 2008) as model input for all soil properties. Corn rooting





depth is set to 1 meter and the corn depletion factor is 0.55; and the depletion factor is 0.5, following Siebert and Döll (2010). Once the soil moisture content reaches field capacity, no further water is added to the soil moisture pool; excess inputs move to

the groundwater pool via percolation and the river system via runoff.

Fig. 3 shows the scatter plot of cumulative precipitation versus mean soil moisture for US counties for the growing season from 1981 to 2015. The simple correlation coefficient between them is 0.44. This ensures the soil moisture output is not a simple linear transformation of precipitation data. We have also investigated other correlations including the correlation between mean soil moisture and evapotranspiration as illustrated in Fig. A2; and the correlation between mean soil moisture and mean daily soil

moisture fraction as shown in Fig. A3.

One limitation for historical analysis is the inconsistency of WBM and PRISM grid cells as they have different extent, different resolution, and non-matching centroids. Therefore, we interpolate WBM to PRISM using nearest neighbor and bilinear methods. The main regression results are reported for bilinear interpolation. However, the regression results using the nearest neighbor interpolation method are very similar (Table A6). The interpolation provides soil moisture information at 2.5 x 2.5 arc-minute grid

cells.

Soil moisture in the model is calculated as the mean of soil moisture content (in mm for the 1000 mm topsoil) during the growing season (Apr-Sep) for each 2.5 x 2.5 arcmin grid cell. For the interaction of soil moisture and heat, we sum up degree days for each temperature interval (5ºC) for each soil moisture deviation interval (10 mm) for each 2.5 x 2.5 arcmin grid cells for the 1981-2015 period. These fine-scale metrics are checked with satellite scans of cropland area to exclude grid cells with no cropland. Finally,

we aggregate all the grids in each county using cropland area weight.

Fig. 4 displays "normal" soil moisture, which is the temporal average of daily soil moisture data over Apr-Sep over 1981-2015, for the Continental US. This map shows the rich heterogeneity of these data across the nation. However, there are distinct regional patterns. For the Corn Belt, the soil moisture level is relatively high compared to other regions. To operationalize this metric, we consider soil moisture deviation from normal. Soil moisture deviation is defined as daily soil moisture minus the normal soil

moisture levels. The soil moisture level is considered critical if it is 25 mm below/above normal condition. The threshold is obtained by testing the impacts of 5-mm intervals of soil moisture deviation from normal.

This implies that the variation in seasonal mean soil moisture may not follow the variation in seasonal mean precipitation. Figure 5 illustrates the year-on-year variation of the precipitation and soil moisture indexes aggregated over the corn growing areas in the US. In general, variation in soil moisture average is higher than in that of precipitation.

Fig. 6 shows the bivariate density of daily temperature and soil moisture ratio to normal for all the grid cells in the Corn Belt for 1981-2015 by the month of the year, capturing the daily variation of the heat and soil moisture combinations. The data shows significant month-to-month variation, with the second half of the season facing hotter and drier days. Also, July has the highest variation in soil moisture deviation.

### 3.4 Estimation strategy

For Model (1), we assume a piece-wise linear form for $g(h)$. We include degree days above 29˚C (as an indicator of extreme heat) as well as degree days from 10 to 29˚C (as an indicator of beneficial heat). Considering the exposure to each temperature interval to capture the marginal impact of heat and water on crop yields, we estimate the following:

$$y_{it} = \alpha D_{it}^{10-29C} + \beta D_{it}^{29} + z_{it}\boldsymbol{\delta} + \lambda_s T_t + \lambda_s T_t^2 + c_i + \epsilon_{it} \qquad (1')$$

where $i$ is an index for counties, $t$ is the index of time, $s$ is the index for states, $y_{it}$ shows the log corn yields, $D_{it}$ represents

degree day variables, $z_{it}$ includes indicators of water conditions, $T$ shows the time trend variable ($T$ = year – 1950), $c_i$ is a time-





invariant county fixed effect, and $\alpha$, $\beta$, $\delta$, $\lambda$ are the regression parameters showing the marginal impacts. We assume the errors are serially correlated due to unobservable and systematic measurement errors, and we consider clustering US counties by the state which has been a standard approach in the literature (Blanc and Schlenker, 2017; Hsiang, 2016; Lobell and Burke, 2010). In this study, the models are estimated using a panel fixed-effect approach. The panel consists of 35 years (1981-2015) for all US counties

with corn production. For purposes of model comparison, we provide adjusted $R^2$, Akaike's information criterion (AIC) and Bayesian information criterion (BIC).

For Model (2), we consider the daily interaction of heat and soil moisture. The key empirical challenge arises when estimating the model with daily interaction of heat and soil moisture. A simple multiplicative interaction of soil moisture and heat will be problematic (Hainmueller et al., 2019). It implies a linear interaction effect that changes at a constant rate with heat. However, as

will be shown below, soil moisture has a non-linear marginal effect. We take two approaches here to calculate the conditional marginal impact of heat on corn yields.

First, we construct a binning estimator based on daily interaction on heat and soil moisture. We define several intervals of soil moisture represented by daily dummy variables and we interact these dummy variables with daily excess heat index. We considered 29°C a critical temperature for heat. Also, we take 25 mm intervals for soil moisture deviation from normal. In other words, we

split the degree days into degree days conditional to soil moisture conditions. This includes dday29°C & SM 75+ mm below normal (extreme deficit), dday29°C & SM 25-75 mm below normal (deficit), dday29°C & SM 0-25 mm around normal (normal), dday29°C & SM 25-75 mm above normal (surplus), and dday29°C & SM 75+ mm above normal (extreme surplus). We estimate a coefficient for each combination of excess heat and soil moisture. In other words, we will estimate a model with indicators of degree days while controlling for soil moisture. The model will provide the conditional marginal impact of excess heat as:

$$y_{it} = \alpha D_{it}^{10-29} + \left\{ \sum_m \beta_m D_{mit}^{29} \right\} + \delta M_{it} + \delta' M_{it}^2 + \lambda_s T_t + \lambda_s' T_t^2 + c_i + \varepsilon_{it} \qquad \text{(2-a)}$$

where $i$ is the county index, $t$ is the time index, $m$ is an index of soil moisture condition (high, low, normal), $s$ is an index for states, $y$ shows average corn yields, $D$ represents conditional growing degree day variables, $M$ shows the seasonal mean soil moisture content, $T$ stands for the time trend variable, $c_i$ is a time-invariant county fixed effect. Here, $\beta$ is indexed by $m$. In other words, the marginal impact of heat is conditional to soil moisture conditions.

Second, we estimate a model with indicators of soil moisture while controlling for temperature. We define an index of soil moisture when the temperature is above the threshold and an index of soil moisture when the temperature is below the threshold. In this model, the soil moisture is separated by a temperature threshold $H^*$.

$$y_{it} = \alpha D_{it}^{10-29} + \beta D_{it}^{29} + \left\{ \sum_m \delta_m M_{mit}\big|_{H<H^*} + \delta_m' M_{mit}\big|_{H>H^*} \right\} + \lambda_s T_t + \lambda_s' T_t^2 + c_i + \varepsilon_{it} \qquad \text{(2-b)}$$

where $i$ is the county index, $t$ is the time index, $m$ is an index of soil moisture condition, $s$ is an index for states, $y$ shows average

corn yields, $D$ represents growing degree day variables, $M$ shows conditional seasonal mean soil moisture, $T$ stands for the time trend variable, $H$ is the average daily temperature, $H^*$ is the temperature threshold, and $c_i$ is a time-invariant county fixed effect. Here, we define $\delta$ and $\delta'$ to test whether the marginal impact of soil moisture depends on heat. The soil moisture indicators are calculated from daily gridded data and aggregated to county and growing season. This includes the index of normal soil moisture ($SM$ 0-25+ mm around normal) when $H > H^*$, the index of normal soil moisture when $H < H^*$, the index of moisture deficit ($SM$

25+ mm below normal) when $H > H^*$, index of moisture deficit when $H < H^*$, the index of moisture surplus ($SM$ 25+ mm above normal) when $H > H^*$, and the index of moisture surplus when $H < H^*$.



## 4 Results

This section provides estimation results for different representations of Model (1). We will discuss the implications of these results in Sect. 5. Regression coefficients, standard errors, R-squared, AIC, and BIC values for Models (1-a), (1-b), (1-c), and (1-d) are

reported in Table 2. The first column (1-a) shows a strong relationship between corn yields and heat and precipitation. The marginal impact of a degree-day within 10-29°C is significantly positive while that from an additional degree day above 29°C is strongly negative, confirming the seminal findings of Schlenker and Roberts (2009). The second column, excluding precipitation, shows the marginal relationship with soil moisture is also significant. It shows that the marginal relationship with soil moisture is increasing up to ~92 mm and decreasing for higher values.

In Model (1-c), we consider the number of days that soil moisture is either too high or too low. The model with metrics of soil moisture extremes further improves the fit, revealing a negative marginal relationship associated with the number of days with low/high soil moisture. Regarding Model (1-c), the coefficient on the number of days with low moisture is also significant and negative. Our estimation sample shows 26 days of high soil moisture and 27 days of low soil moisture on average. The implication is that eliminating 25 days of high soil moisture and 25 days of low soil moisture can improve the corn yields by up to 12.6%.

Model (1-d) shows the estimated coefficients when considering surplus and deficit (soil moisture deviation from normal) instead of average seasonal soil moisture. Here, we consider two thresholds for low and high soil moisture. Returning to Fig. 1, we evaluate the area of all blue bars and the area of all red bars. It shows that the marginal impact of the moisture deficit (cumulative negative soil moisture deviation) is significant and positive. This indicates the positive contribution of additional soil moisture when the soil moisture levels are below normal. On the other hand, the marginal impact of additional soil moisture in a wet period – i.e., a

positive soil moisture deviation -- is negative. In other words, this measure captures the fact that plants will benefit from reductions in soil moisture when the soil moisture levels are above normal. This is an indicator of the value of sub-surface drainage for agriculture. Note that the Model (1-d) decreases the marginal relationship with extreme heat (DD29°C). However, this effect is not statistically different from that produced by the first model. A central finding is that metrics of soil moisture extremes are statistically significant.

The coefficient of the deficit in Model (1-d) is significant and positive. On the other hand, the coefficient of the extreme deficit is also significant and positive. The estimation sample shows this indicator is around 2300 mm on average. It indicates that reducing the deficit and by 2300 mm and reducing the surplus by the same amount can improve the corn yield by up to 21.2% on average. Note the mean soil moisture can stay unchanged in this scenario.

We introduce heat-soil moisture interactions to test whether soil moisture availability changes the marginal impact of heat. In

Model (2-a), we estimate a model while splitting the heat stress index according to soil moisture conditions. Here we construct the heat index for different intervals of soil moisture deviation. Table 3 shows the estimation results. As shown in this table, the average marginal impacts of dday29°Cs are all significant. The coefficient on dday29°C (heat stress) combined with the extreme deficit is -0.0082. The coefficient of ddays29°C (heat stress) combined with extreme water surplus is -0.0140. These figures are significantly different compared to Model (1).

Figure 7 illustrates these coefficients while they are translated to the percentage change in corn yields assuming additional 10 degree-days above 29°C and no change in mean soil moisture. The error bars show the 95% confidence interval when simply taking the standard errors of the estimation. The shaded area is the 95% confidence interval when using Model 2. The figure shows that Model (1) would significantly underestimate the damage for conditions with extreme water surplus or extreme water deficit. Finally, we estimate a model with soil moisture while controlling for temperature (2-b). The results are presented in Table 4. The

coefficient of degree days from 10°C to 29°C is significant and positive. This is not significantly different from previous models. The coefficient on degree days above 29°C is significant and negative. It is close to the estimated values from Model (2-a) but



slightly lower than Model (1). This indicates that the average damage from extreme heat index (dday29˚C) is around 25% lower than Model (1). The coefficient on normal soil moisture conditional to hot weather is 0.00012. The coefficient on normal soil moisture conditional to moderate weather is 0.00003. This indicates that water is up to four times more valuable in hot weather.

The marginal impact on soil moisture deficit index is 0.00009 in hot weather and is 0.00002 in moderate weather. This also supports the finding that water is up to four times more valuable in hot weather. Also, the results suggest that the damage from excess water is up to two times bigger in hot weather.

## 5 Discussion and robustness checks

We have presented estimated coefficients for different models for predicting corn yields with individual and compound extremes.

We find that the coefficients on the soil moisture metrics are significant and with expected signs. We also find that *the average damage from excess heat has been up to four times more severe when combined with water stress.* Comparing the models' performance suggests that Model (1-b), with mean soil moisture, performs better than the Model (1-a), with cumulative precipitation. Also, Model (1-d), with the extreme soil moisture metrics, outperforms both previous models (with cumulative precipitation or with mean soil moisture). Finally, the best performance is from Models (2-a) and (2-b), considering compound

extremes through the daily interaction of heat and soil moisture.

The appendix of this paper provides several robustness checks. Overall, the findings remain robust to alternative soil moisture indicators from WBM including the mean of soil moisture fraction (soil moisture content divided by field capacity), the mean of evapotranspiration as well as within season standard deviation of them. We also report the results for an alternative interpolation of WBM data to PRISM resolution (nearest neighbor versus bilinear). To test for time separability, we estimate Model (1-b) for

two-month intervals (Apr-May, Jun-Jul, Aug-Sep) as well as for the whole season. Finally, we have provided different interaction models. Here we discuss the implications of our findings for climate-agriculture-water studies, as well as in broader literature of climate impact studies.

### 5.1 Should we replace precipitation with soil moisture in climate-related studies?

Cumulative precipitation and mean soil moisture can be strongly or weakly correlated. The main factors in the difference between

the two are runoff, drainage, and irrigation. If the runoff is rare and there is little or no irrigation, there is a high chance that cumulative precipitation and mean soil moisture are strongly correlated. If they are expected to stay highly correlated, then adding soil moisture to the model may not benefit the researchers. Precipitation may still be a valid measure of future water availability for locations with small runoff, drainage, and irrigation.

However, as drainage becomes more attractive in the Eastern US and irrigation dominates in the Western US, the correlation

between cumulative precipitation and mean soil moisture weakens considerably. A quick test for a given locality would involve looking at the share of irrigated area, the area equipped with drainage systems, the share of intensive precipitation, or the number of days without precipitation. If any of these are dominant or expected to be dominant in the future, we recommend the use of soil moisture data or other metrics of water availability as suggested in the literature but not cumulative precipitation.

The studies using degree days above a critical threshold may capture part of the damages from low soil moisture. This can be due

to feedbacks and dynamics of temperature and soil moisture as abundant soil moisture can reduce the temperature and extreme heat can reduce the soil moisture (D'Odorico and Porporato, 2004; Seneviratne et al., 2010). While previous studies are unable to capture the different impacts of dry heat versus wet heat (Feng and Zhang, 2015; Schoof et al., 2017), our study suggests that the impact of excess heat can be significantly different while considering soil moisture.





### 5.2 Should we use mean soil moisture or deficit-surplus metrics?

Does it make a difference to consider seasonal mean soil moisture or metrics of extreme conditions? Figure 8 illustrates the difference by comparing the impacts of soil moisture on log corn yield using the estimated coefficients. The black curve in Fig. (8-a) shows the relationship between soil moisture and log corn yield from Model (1-b), without the extreme conditions and the interaction term. This indicates a more general relationship that looks like an envelope to local functions. Panel b and c show the relationships considering the deviation from normal in Model (1-d) drawn for a clay soil type (c) and sandy soil type (b). In other

words, when parametrizing the soil moisture as a deviation from normal, we get a specific piece-wise linear yield response to water depending on soil types (and normal levels of soil moisture).

In addition, an examination of extreme conditions can improve our understandings of climate impacts with intensive extreme events. Generally, if the location of the study does not expect a significant change in the within-season distribution of the soil moisture, a mean soil moisture index will work. However, if there is an expected change in this distribution, using the mean variable

will create a bias.

Here is an example from our sample. Consider the case of Bureau County in Illinois (FIPS: 17011). Our data show that the mean soil moisture is almost the same in 1983 and 1992 and is around 125 mm. However, soil moisture deficit and surplus metrics are quite different. They are -5,934 mm and +4,937 mm for 1983, and -2,451 mm and +2,137 mm for 1992, respectively. As a result, Model (2) predicts almost no impact from the change in soil moisture, while Model (3) predicts a 24.7% increase in corn yield due

to changes in soil moisture. While this may be counted as a rare incidence historically, this may not be the case in the future. As climate models are warning about significant changes in the frequency and intensity of extreme precipitations (Myhre et al., 2019), the mean metrics of water availability are not enough in capturing the impacts of water on yields. It is necessary to consider the metrics of extreme events in the models.

### 5.3 Should we use daily interaction of soil moisture with heat metrics?

Recall Fig. 6 which illustrates the historical dynamics of change in the daily distribution of heat and simulated soil moisture over the Corn Belt for the 1981-2015 period. Throughout the growing season, the density moves in the direction of lower soil moisture and warmer conditions. If a location is expected to face minimal changes in the bivariate distribution of heat and water availability, adding interaction terms will benefit the analysis relatively little. However, if a significant change is expected in compound extreme events, then the use of models with interaction terms is inevitable.

There is an increasing body of literature in climate sciences about the changes in the likelihood of compound extreme events (Zscheischler et al., 2018; Manning et al., 2019; Bevacqua et al., 2019; Poschlod et al., 2020; Potopová et al., 2020; Wehner, 2019). Therefore, this research is critical for evaluating the impacts of future climate change as we found that the coefficient on extreme heat is significantly different when considering soil moisture. We have not investigated the size of the overestimation or underestimation of climate impact studies. However, predictions of significant changes in precipitation and soil moisture within

the growing season suggest that the impact could be substantial.

However, applying this framework to climate impact studies may face a key challenge —namely projecting the future compound extremes. It requires collaboration between hydrologists, climate scientists, and statisticians (Zscheischler et al., 2020). For future projections, we need reliable future projections of daily temperature (maximum and minimum). Unfortunately, to the best of our knowledge, available data sets including predictions of future soil moisture have a relatively coarse spatial and temporal resolution.

While various climate products are projecting future daily temperatures, the choice of climate model requires extreme caution and should be compatible with the special needs of each study. Although there are some projections of future levels of soil moisture,





there is a great deal of inconsistency among the models regarding this variable. Further research is required to improve the ability of climate models in projecting the bivariate distribution of heat-moisture (Sarhadi et al., 2018).

In places predicted to face higher mean precipitation coupled with more extreme water stress, adaptation through soil moisture management will be beneficial to farmers. This may motivate investments in supplemental irrigation. Also, farm management practices such as no-till farming, cover cropping, and soil conservation can increase soil moisture without (or in addition to) irrigation. Farmers may also consider improvements in water use efficiency, both by crops and by irrigation systems, as one way to address the need for increased irrigation. However, the expansion of irrigation can increase the stress on global water resources.

**5.4 Implications for irrigation water demand and subsurface drainage**

Considering the estimated coefficients for Model (2), we construct the daily marginal value product of soil moisture conditional on a given soil type and temperature. The economic literature on the value of water offers a variety of techniques to estimate the value of irrigation water in agriculture (Aubuchon and Morley, 2013; García Suárez et al., 2018; Gemma and Tsur, 2007; Griffin, 2016; Mesa-Jurado et al., 2010; Mukherjee and Schwabe, 2014; Rigby et al., 2010; Young, 2010). The agronomic literature also considers deficit and supplemental irrigation (Hargreaves et al., 1989; Hargreaves and Samani, 1984) and crop water productivity

(Kang et al., 2009; Zwart and Bastiaanssen, 2004). Here we employ the marginal value product (MVP) approach to estimate the value of water (Costanza et al., 1997; Griffin, 2016; Young, 2010). In this approach, the impacts of the change in the water input are estimated while assuming other inputs are constant. Assuming a general form of production function as $Y = Y(L, W, H)$ where $Y$ is the output, $L$ is augmented land, $W$ is water, and $H$ is heat. The MVP of water is given by:

$$P_Y \frac{\partial Y(L, W, H)}{\partial W} = MVP_w$$

where $P_y$ shows the price of output. If farmers make their decisions about other inputs before the planting date, then the variation in $Y$ is mainly due to variation in $W$ and $H$. Figure 9 shows one example of this marginal productivity assuming clay soil type (with the normal moisture around 200 mm) and temperature around 25C. In the left panel, the marginal contribution is displayed. In the right panel, the marginal value product is illustrated assuming the price of corn is $3.5 per bushel. Note that even at normal soil moisture levels, the value of water is positive.

**5.5 Decomposing the variation in US corn yields**

We find that *the average damage from excess heat has been up to four times more severe when combined with water stress.* To illustrate the significance of this finding, we have decomposed the changes in the US corn yields from 1981 to 2015 considering soil moisture and heat. Figure 10 illustrates a decomposition based on our findings while aggregated for the whole US. With no climate variation, the US corn yield is expected to have a smooth positive trend as shown in green color. The deviation from the

trend occurs due to changes in water and heat stressors. The blue bars are showing the expected changes in US corn yields due to changes in the water stress while the orange bars are demonstrating the expected yield changes due to changes in heat stress. While there have been years in which the stressors have moved together (e.g. 2011 and 2012), for several years water and heat have offset each other's benefit or damage. For example, in 1992 the damage from heat is partially offset by benefits from water. Or in 2010, the damage from water stress is partially offset by benefits from heat.



### 5.6 Implications for climate studies

The results emphasize the value of soil moisture management as an effective means of adaptation to climate change. This adaptation can moderate production damages from a hot future climate. Thus, we predict that supplemental irrigation will be more beneficial to farmers. However, the expansion of irrigation in many areas may lead to further increases in unsustainable groundwater withdrawals. Such trade-offs are inevitable as environmental stresses in agriculture increase in the future. Furthermore, we confirm that excess soil moisture is damaging for corn and it is intensified when combined with heat stress. This emphasizes the importance of subsurface drainage for crop production in the future.

We have examined the possible impacts of climate change on global corn yields by the mid-century. We employ information from NASA Earth Exchange (NEX) Global Daily Downscaled Projections (NEX-GDDP) product for Representative Concentration Pathway (RCP) 8.5 scenario at 15 x 15 arcmin at the global level. We consider grid-specific growing season according to SAGE growing calendar (Sacks et al., 2010) and carefully calculate the degree days considering leap years. According to this projection, heat stress conditions are expected to increase sharply in the US by the mid-century. Figure 11 shows the climate impact on maize yields at the global level for irrigated and non-irrigated corn based on the CCSM4 model. This figure shows a heterogenous impact around the world. The critical finding is that adaptation through irrigation can significantly reduce the damage of heat stress on corn yields.

These findings are important for assessing the regional resilience of agroecosystems, global food security, and as well as future climate impacts. This framework can help farmers quantify the daily importance of soil moisture for future climate adaptation which can indirectly enhance food security. At the policy level, this study improves our understanding of the implications of compound hydroclimatic extremes which are critical to economic assessments undertaken at the local, national, and global levels. The estimation framework also provides a better measurement of climate-related variables which is also valuable for economic studies. Our findings also provide a significant contribution to the climate impact literature through the estimation of the monetary value of damages from compound hydroclimatic extremes for agriculture.

Finally, this paper demonstrates the value of fine-scale hydroclimatic information for research in the economics of climate change, global environmental changes, and coupled human and environmental systems. A strength of our findings is that they can be used widely by the research community, as many hydrology and land surface models can simulate soil moisture. Also, this method can be tailored for use with different climate model outputs as well as different soil maps. It can also accommodate the analysis of hypothetical situations (e.g., drought) which may vary by study location and research question at hand.

### 6 Conclusions

This study serves to bridge the gap between statistical studies of climate impacts on crops and their biophysical counterparts and underscores that findings of statistical models based on county-level data are in line with experimental agronomic studies (Lobell and Asseng, 2017). We employ a fine-scale dataset to investigate the conditional marginal value of soil moisture and heat in US corn yields for the 1981-2015 period employing a statistical framework. The major contribution of this study is showing that the coefficient on extreme heat (DD29˚C) is significantly different while considering daily interactions with soil moisture emphasizing the importance of compound hydroclimatic conditions.

Our first key finding is that seasonal mean soil moisture performs well in statistically predicting corn yield. While the majority of current empirical studies employ precipitation as a proxy of water availability for crops, we show that the precipitation coefficient may not be always an appropriate measure of water availability. This study suggests that soil moisture content should be used in



estimating crop yields instead of cumulative rainfall for locations with high runoff, drainage, or irrigation (e.g. Western and Central US).

Also, the indicators of soil moisture extremes can explain a portion of the damages to corn yield. On average, farmers can improve corn yields by up to 24% only by avoiding extreme water stress. We also find that the coefficient of excess soil moisture is negative. This is in line with the current agronomic literature (Torbert et al., 1993; Urban et al., 2015) which points out that high soil moisture content can result in nutrient loss through excess water flows. In addition, at high humidity, the plants may have difficulty remaining cool at high temperatures. There is also a risk of waterlogging soils. With a few notable exceptions (e.g., rice), most crops do not grow well in inundated conditions as the plant roots need oxygen, so the direct impact of excess water stress is because

of the anoxic conditions.

Finally, the marginal impact of heat index on crop yields depends on the soil moisture level. We show the average damage from heat stress has been up to four times more severe when combined with water stress; and the value of water has been up to four times bigger on hot days.

**Appendix**

This appendix provides some robustness checks on the results and the model variables. First, we illustrate the relationship between mean soil moisture and other seasonal variables in this study. This includes mean seasonal evapotranspiration, mean seasonal soil moisture fraction, and the degree days above 10˚C. We provide some examples to demonstrate the seasonal mean soil moisture shows no linear relationship with the seasonal heat index (degree days above 10˚C). However, it has a positive correlation with evapotranspiration and soil moisture fraction. Then we provide alternative models controlling for irrigation, growth periods, spatial

scope of the study, and other measures of individual and compound extremes.

**A.1. Correlation of mean seasonal soil moisture and other variables**

The soil moisture output from WBM is informed mainly by soil moisture memory, heat, precipitation, and many other time-variant and time-invariant information. We have taken two other variables from WBM including soil moisture fraction and evapotranspiration (ET). Also, we have interpolated WBM soil moisture using an alternative method (nearest neighbor method).

Section A.6 will provide the estimation results when using these variables to show the robustness of the results to variable selection. Here we plot these variables against the volumetric soil moisture content to illustrate the correlation and differences. As shown in Fig. A1 two interpolations of soil moisture are closely correlated by R= 0.9997. Figures A2 and A3 are the scatter plots of seasonal ET and seasonal mean soil moisture fraction against volumetric soil moisture. The figures show the seasonal variables are not following a simple linear relationship. Figure A4 shows the scatter plot of cumulative growing degree days above 10˚C versus

mean soil moisture for US counties for the growing season from 1981 to 2015. This indicates the soil moisture output is not a simple linear transformation of heat data.

**A.2. Robustness check: controlling for normal soil moisture (2-c)**

Here we introduce Model (2-c) trying to control for compound stresses. The idea is that the daily excess heat may come with or without water stress. Thus, the coefficient on the excess heat indicator (DD29˚C) shows the impact of heat stress which usually

involves some water stress. We estimate a model with the interaction of heat indicators and index of soil moisture defined as the "share of heat at normal moisture".





$$y_{it} = \alpha D_{it}^{10-29} + \beta D_{it}^{29} + \alpha' S_1 D_{it}^{10-29} + \beta' S_2 D_{it}^{29} + \left\{ \sum_m \eta_m MD_{mit} \right\} + \lambda_s T_t + \lambda_s' T_t^2 + c_i + \varepsilon_{it} \qquad (2\text{-c})$$

where $i$ is the index for counties, t is an index for year, $y_{it}$ shows log corn yields by county and year, $m$ is an index of soil moisture bins (high, low, normal), $s$ is an index for the US states, $D$ represents growing degree day variables, $MD$ shows soil moisture
metrics, $T$ is a time trend index. For calculating $S$, we split the degree days according to soil moisture bins. Then, $S$ is defined as the share of degree days around normal soil moisture over seasonal degree days.

$$S_1 = \frac{\sum_d D_d^{10-29}\big|_{M \sim \bar{M}}}{\sum_d D_d^{10-29}}, \qquad S_2 = \frac{\sum_d D_d^{29}\big|_{M \sim \bar{M}}}{\sum_d D_d^{29}}$$

Here, S1 is the share of DD10-29˚C with normal soil moisture for all DD10-29˚C calculated by county and by growing season; and S2 is the share of DD29˚C with normal soil moisture for all DD29˚C at each county each growing season. This model makes
the interpretation easier for decomposing heat stress from water stress. With no water stress, $S_1 = S_2 = 1$, and soil moisture is always around normal levels. Thus, $\alpha + \alpha'$ will show the marginal impact of additional DD10-29˚C; and $\beta + \beta'$ will show the marginal impact of an additional DD29˚C. On the other hand, for water-stressed corn, $S_1 = S_2 = 0$, and soil moisture is not around the normal levels. Therefore, $\alpha$ will show the marginal impact of additional DD10-29˚C combined with water stress; and $\beta$ will show the marginal impact of an additional DD29˚C combined with water stress. We will discuss this while looking at the estimation results.
Table A1. provides the estimated coefficients, standard errors, R-square, and AIC and BIC for Model (2-c). The coefficient on the beneficial heat (DD10-29˚C) is significant and positive. The coefficient on its interaction with soil moisture (share of heat at normal soil moisture) is also significant and positive. With no water stress (S1= 1), the point estimate of the marginal impact of the beneficial heat is 0.00036 (+0.00025 + 0.00011). The marginal impact will be 0.00025 when the soil moisture is not at normal levels. The finding that heat is less beneficial with soil moisture deficit is not a surprise and is in line with agronomic literature.
However, it has crucial implications for climate impact studies as it suggests a likely overestimation of the benefits of global warming in the context of more erratic and concentrated rainfall events.

The coefficient on the extreme heat (DD29˚C) is significant and negative. However, the coefficient on its interaction with soil moisture (share of heat at normal soil moisture) is significant and positive. In other words, with no water stress (S2= 1), the marginal impact of extreme heat is -0.0040 (-0.0057 + 0.0017). While with water stress, it is -0.0057. This is another critical finding and
approves the results from Models (2-a) and (2-b). It shows when soil moisture is not at normal levels, the average damage from excess heat is around 43% more severe than normal conditions of soil moisture. Note that this is the average damage and the actual damage can be more/less severe depending on the degree of water stress as we show in Model (2-a).

The estimated coefficients on soil moisture metrics (sum of daily deviations below and/or above a threshold) are significant. This coefficient is -0.000023 for soil moisture surplus and is +0.000050 for normal soil moisture. The important finding is that the signs
are as expected and consistent with other models.

**A.3. Robustness check: controlling for irrigation in Model (2-d)**

Here we estimate Model (2-d) controlling for irrigation. While irrigation ensures the soil moisture at normal levels, it can have additional benefits than just providing water. One major impact is the *cooling effect* of irrigation technologies as used in the Western US. Sprinkler irrigation not only reduces the water stress but also makes the air temperature lower near the surface. This will reduce
heat stress and is not captured by GDD29˚C based on PRISM. Another point is that the panel focused on the West includes a little





variation in soil moisture metrics as suggested by various water deficit and surplus indicators in Table A3. As a result, the soil moisture data for the Western US provides little information on soil moisture variations.

Here, we investigate whether irrigation has other benefits than just providing soil moisture. To control for irrigation, we re-estimate Model (2) including a term for interaction of DD29˚C and share of irrigated area. Specifically, we estimate this model:

$$y_{it} = \alpha D_{it}^{10-29} + \beta D_{it}^{29} + \beta' S_2 D_{it}^{29} + \beta'' S_3 D_{it}^{29} + \left\{ \sum_m \eta_m M D_{mit} \right\} + \lambda_s T_t + \lambda_s' T_t^2 + c_i + \varepsilon_{it} \qquad (2\text{-}d)$$

Here, if S2 =1, S3=0 then $\beta + \beta'$ will show the marginal impact of an additional DD29˚C at normal soil moisture for non-irrigated corn; if S2 =1, S3=1 then $\beta + \beta' + \beta''$ will show the marginal impact of an additional DD29˚C at normal soil moisture for irrigated corn. Table A2 shows the results. All the estimated coefficients are significant. In summary, the marginal impact of additional DD29˚C: for non-irrigated corn and with water stress is -0.0062; for non-irrigated corn and without water stress is -0.0044; and for irrigated corn without water stress is -0.0020; In other words, we strongly reject the hypothesis that irrigation benefits are limited to providing soil moisture. This also explains the differences between Western and Eastern US as discussed in Sect. S.5.

### A.4. Robustness check: West versus East in Model (1)

In this section, we estimate the main models separately for Eastern and Western US. Those counties with centroids on the left of 100th meridian are considered West. The idea is that water stress is less severe in the Western US as it is mostly irrigated. Table A3. provides the main descriptive statistics to compare these regions. Overall, Western US experiences more excess heat by 82 versus 58 DD29˚C in the East. On average, Eastern US receives 601 mm of cumulative precipitation while it is only 271 mm in the Western US. On the other hand, within-season SD of soil moisture is 39 mm in the East while it is 13 mm in the west. Looking at the number of days with high/low soil moisture, only 11 days in the West soil moisture is not at normal levels, while this is 59 days in the East.

Table A4. shows the estimated coefficients, standard errors, adjusted R-squared, AIC, and BIC statistics for four models for Eastern US. Model (1-a) includes cumulative precipitation. Model (a-2) includes mean soil moisture metrics. The third model, similar to Model (3b), considers soil moisture extremes. And Model (4) considers the interaction terms. The results suggest that the coefficient on the extreme heat is not significantly different from the estimations for the whole US.

Table A5. shows the estimated coefficients, standard errors, adjusted R-squared, AIC, and BIC statistics for four models for the Western US. The results suggest that the coefficients on the extreme heat are significantly different from the estimations for the whole US and the Eastern US. For example, the coefficient on DD29˚C is -0.0020 in Model (1) for the West, while it was estimated -0.0056 for the East. This is around 65% lower damage for a given degree day above 29˚C. Also, the AIC and BIC statistics would reject the hypothesis that models with interaction perform better compared to the model with cumulative precipitation. The difference can be a result of the "*cooling effect*" as discussed in Sect. S.4.

### A.5. Robustness check: bi-monthly metrics of soil moisture

Table A6. provides the estimation coefficients, standard errors, AIC, BIC, and R-squares statistics for Model (1-b) for Eastern, Western, and the continental US with bi-monthly mean soil moisture. The results suggest that the coefficients on extreme heat (DD29˚C) are not significantly different from the model with seasonal mean soil moisture.

The results suggest that the marginal impact of mean soil moisture is higher in June-July. This is in line with agronomic literature as it suggests the water stress during pollination and the silking stage is more damaging. These stages are the most critical stage of





development for corn. Water stress during this stage can cause higher yield loss than almost any other stage in the crop's development.

The marginal impact of mean soil moisture is not significant in August-September. This suggests that additional soil moisture can have a positive or negative impact on yield. This also makes sense as a high level of moisture can hurt the maturity and drying

stage. High soil moisture at the end of the growing season can cause delayed grain maturity and may lead to delay in the harvest. In Addition, the marginal impact of mean soil moisture in April-May is negative for the whole US and the Western US and significant at 90% confidence interval. This can be a result of the negative impacts of excess soil moisture on germination and early crop developments as a result of flooding and waterlogging.

### A.6. Robustness check: other metrics from WBM outputs (soil moisture fraction and ET)

Here, we re-estimate Model (1) with other related metrics of water availability to crops including simulated daily evapotranspiration of rainfed corn (ET) from WBM; daily soil moisture fraction (SMF) from WBM; and soil moisture content from different spatial interpolation of WBM grid cells to PRISM (nearest neighbor method versus original bilinear method).

The soil moisture fraction index considers the volumetric soil moisture content divided by field capacity. We have also considered the within-season standard deviation of ET and SMF. Note that we keep the degree days above 29°C as an indicator of heat stress

and the degree days from 10°C to 29°C as an indicator of beneficial heat to corn.

Table A7. reports regression results for these models. Columns 1 and 2 show a significant relationship with the mean of soil moisture fraction, its square term, and its within season standard deviation. Columns 3 and 4 with mean ET and within-season SD of ET also show a significant relationship. Column 5 shows that the other interpolation of soil moisture has a very close marginal coefficient and standard error compared to our original Model (1). The important finding is the marginal relationship for beneficial

and harmful heat remains significant and not significantly different from Model (1).

### A.7. Robustness check: East and West in Model (2)

Her we re-estimate Model (2). The results are presented in Table A8 and A.9 for the US, West, and East. We see a similar pattern for East versus West. The coefficient on heat stress is smaller for the West which can be a result of the cooling effect.

The results of Model (2-a) are presented in Table A8. Column 1 shows the results for the whole US while columns 2 and 3 contain

the results for the Western US and Eastern US, respectively. According to column 2, the coefficient on dday29°C and the extreme deficit is -0.0074 in the Western US which is significantly different from all other estimations for the Western US. This is another evidence of the cooling effect. These results indicate that, even in the Western US, *the damage from heat stress can be up to four times higher when combined with water stress*. The coefficient on excess heat and the extreme surplus is not significant (note that this is a very rare condition in the West).

As in column (3) of Table A9, the coefficient on normal soil moisture conditional to hot weather is 0.00010. The coefficient on normal soil moisture conditional to moderate weather is 0.00002. This indicates that water is up to four times more valuable in hot weather. The marginal impact on soil moisture deficit index is 0.00008 in hot weather and is 0.00002 in moderate weather. This also supports the finding that water is up to four times more valuable in hot weather. Also, the results suggest that the damage from excess water is up to two times bigger in hot weather.


*Code availability*. The codes are available at DOI:10.4231/Q07D-J369.



*Data availability*. The historical weather data (PRISM) is available at http://www.prism.oregonstate.edu. The future weather data (NEX-GDDP) is available at https://www.nccs.nasa.gov/services/data-collections/land-based-products/nex-gddp. The input data

for estimations are available at DOI:10.4231/0M14-EY38.

*Author contribution*. All authors contributed to conceptualization, methodology, formal analysis, and writing- review & editing. IH and DSG collected model input, performed the simulations, and contributed to the investigation, resources, software, and validation. IH contributed to writing the original draft and visualization. TWH contributed to supervision and funding acquisition.


*Competing interests*. The authors declare that they have no conflicts of interest.

*Acknowledgement*. This work was supported by the U.S. Department of Energy, Office of Science, Biological and Environmental Research Program, Earth and Environmental Systems Modeling, MultiSector Dynamics, Contract No. DE-SC0016162. The

database construction process is done employing high-performance computing solutions of Institute for CyberScience Advanced CyberInfrastructure (ICS-ACI) at Penn State University. Any opinions, findings, and conclusions or recommendations expressed in this material are those of the authors and do not necessarily reflect the views of the US Department of Energy.





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





**Table 1. Yield, heat, and water metrics for 1981-2015 (Apr-Sep)**

| Variable | Mean | Std.Dev | Min | Max |
|---|---|---|---|---|
| Corn yield (bushels / acre) | 109.8 | 37.8 | 4.5 | 246.0 |
| Cumulative precipitation (mm) | 564 | 183 | 1 | 1469 |
| Mean daily soil moisture content (mm) | 47 | 39 | 0.1 | 262 |
| in Apr-May (mm) | 60 | 48 | 0 | 270 |
| in Jun-Jul (mm) | 48 | 45 | 0 | 270 |
| in Aug-Sep (mm) | 30 | 30 | 0 | 264 |
| Degree days from 10˚C to 29˚C | 1848 | 434 | 693 | 3083 |
| when soil moisture is low | 397 | 430 | 0 | 2629 |
| when soil moisture is normal | 1112 | 572 | 0 | 3044 |
| when soil moisture is high | 330 | 346 | 0 | 2665 |
| Degree days above 29˚C | 61 | 61 | 0 | 723 |
| when soil moisture is low | 18 | 31 | 0 | 400 |
| when soil moisture is normal | 37 | 48 | 0 | 680 |
| when soil moisture is high | 5 | 9 | 0 | 140 |
| Index of soil moisture above normal levels (mm) | 2370 | 2135 | 0 | 20319 |
| Index of soil moisture below normal levels (mm) | -2384 | 2147 | -23978 | 0 |
| Number of days with moisture deficit > 25 mm | 27 | 30 | 0 | 182 |
| Number of days with moisture surplus > 25 mm | 27 | 35 | 0 | 183 |
| Mean daily soil moisture fraction | 0.71 | 0.18 | 0.01 | 1.00 |
| Mean daily evapotranspiration (mm) | 0.55 | 0.58 | 0.00 | 2.95 |
| Number of observations | 69923 | | | |

**Notes: Table reports descriptive statistics for major variables in this study. The mean and standard deviations are calculated over US counties for the 1981-2015 period. All the weather data are calculated for each 2.5 x 2.5 arcmin grids, averaged over the time interval, and then averaged to counties using cropland area weights. Soil moisture seasonal normal is defined as the average of 1981-2015 daily soil moisture level from the first day of April to the last day of September.**





**Table 2. Corn yield estimation without the interaction of heat and soil moisture**

|  | (1-a) Log CornYield | (1-b) Log CornYield | (1-c) Log CornYield | (1-d) Log CornYield |
|---|---|---|---|---|
| Degree Days 10-29°C Apr-Sep | .000336*** (.000087) | .000343*** (.00008) | .0003486*** (.0000725) | .0003083*** (.0000683) |
| Degree Days above 29°C Apr-Sep | -.005307*** (.000673) | -.005114*** (.000691) | -.005277*** (.0006678) | -.005041*** (.0005999) |
| Precipitation Apr-Sep | .000658** (.000254) |  |  |  |
| Precipitation Apr-Sep Square | -5.16e-07** (-9.35e-07) |  |  |  |
| Seasonal Mean Soil Moisture Content |  | .003593*** (.000664) |  |  |
| Seasonal Mean Soil Moisture Content Square |  | -.000017*** (3.000e-06) |  |  |
| Number of days with SM 25+ mm above normal |  |  | -.001838*** (.0003816) |  |
| Number of days with SM 25+ mm below normal |  |  | -.002089*** (.0002817) |  |
| Index of Soil Moisture above Normal (mm) |  |  |  | -.000040*** (2.800e-06) |
| Index of Soil Moisture below Normal (mm) |  |  |  | .000044*** (7.100e-06) |
| Obs. | 69923 | 69923 | 69923 | 69923 |
| R-squared | 0.4686 | 0.4714 | 0.4795 | 0.4914 |
| AIC (Akaike's information criterion) | -21238.1 | -21612.3 | -22696.8 | -24303.4 |
| BIC (Bayesian information criterion) | -21201.4 | -21575.7 | -22660.2 | -24266.8 |

Standard errors are in parenthesis & adjusted for
state clusters
*** $p<0.01$, ** $p<0.05$, * $p<0.1$

**Notes: Table lists regression coefficients and shows standard errors in brackets. Temperature is in degree Celsius, precipitation in mm, soil moisture in mm in 1000 mm topsoil. The soil moisture is obtained from WBM at 6 arcmin output and interpolated to 2.5 arcmin, while precipitation and temperature are taken from PRISM at 2.5 arcmin. They are aggregated from grid cells to counties based on crop area weight. Yield data is acquired from the USDA. The constant term and coefficients on the interaction of each state and time trends are not** 910 **reported.**





**Table 3. Corn yield estimation while splitting heat stress index**

|  | (2-a) Log CornYield |
| --- | --- |
| Degree days from 10˚C to 29˚C | .0003083*** |
|  | (.0000685) |
| dday29˚C & SM 75+ mm below normal (extreme deficit) | -.0082398*** |
|  | (.0014372) |
| dday29˚C & SM 25-75 mm below normal (deficit) | -.0062069*** |
|  | (.0009793) |
| dday29˚C & SM 0-25 mm around normal (normal) | -.0037559*** |
|  | (.0004045) |
| dday29˚C & SM 25-75 mm above normal (surplus) | -.0055709*** |
|  | (.0012041) |
| dday29˚C & SM 75+ mm above normal (extreme surplus) | -.0140295*** |
|  | (.0019083) |
| Mean daily soil moisture content (mm) | .0026635*** |
|  | (.0008153) |
| Square of mean daily soil moisture content | -.0000161*** |
|  | (2.600e-06) |
| Observations | 69923 |
| R-squared | .4921 |
| Akaike's Crit | -24401.6 |
| Bayesian Crit | -24328.3 |

Standard errors are in parenthesis & adjusted for state clusters
*** p<0.01, ** p<0.05, * p<0.1

**Notes:** Table lists regression coefficients and shows standard errors in brackets. Temperature is in degree Celsius and soil moisture in mm in 1000 mm topsoil. The soil moisture is obtained from WBM at 6 arcmin output while precipitation and temperature are taken from PRISM at 2.5 arcmin. They are aggregated from grid cells to counties based on crop area weight. Yield data is acquired from the USDA. The constant term and coefficients on the interaction of each state and time trends are not reported.






**Table 4. Estimation of corn yields while splitting the soil moisture indicators**

| | (2-b) log CornYield |
|---|---|
| Degree days from 10°C to 29°C | .0003154*** (.0000689) |
| Degree days above 29°C | -.004044*** (.0005384) |
| Index of normal soil moisture when T > T* | .0001199*** (.0000342) |
| Index of extreme moisture surplus when T > T* | -.0000628*** (.0000151) |
| Index of extreme moisture deficit when T > T* | .000092*** (.0000234) |
| Index of extreme moisture deficit when T < T* | .0000209*** (7.100e-06) |
| Index of extreme moisture surplus when T < T* | -.0000326*** (3.200e-06) |
| Index of normal soil moisture when T < T* | .000028** (.0000105) |
| Observations | 69923 |
| R-squared | .5006 |
| Akaike's Crit | -25582.4 |
| Bayesian Crit | -25509.2 |

Standard errors are in parenthesis & adjusted for state clusters
*** $p<0.01$, ** $p<0.05$, * $p<0.1$

**Notes: Table lists regression coefficients and shows standard errors in brackets. Temperature is in degree Celsius and soil moisture in mm in 1000 mm topsoil. The soil moisture is obtained from WBM at 6 arcmin output while precipitation and temperature are taken from PRISM at 2.5 arcmin. They are aggregated from grid cells to counties based on crop area weight. Yield data is acquired from the USDA. The constant term and coefficients on the interaction of each state and time trends are not reported.**





**Table A1. Corn yield estimation controlling for normal soil moisture**

|  | (2-c)<br>Log CornYield |
|---|---|
| Degree days from 10˚C to 29˚C x S1 (share of heat at normal moisture) | .000111*<br>(.0000584) |
| Degree days from 10˚C to 29˚C | .0002513***<br>(.0000827) |
| Degree days above 29˚C x S2 (share of heat at normal moisture) | .0017202**<br>(.0008421) |
| Degree days above 29˚C | -.0057224***<br>(.0009421) |
| Index of normal soil moisture | .0000504***<br>(.0000108) |
| Index of extreme moisture surplus (sum of positive deviations if > +25 mm) | -.0000233***<br>(6.700e-06) |
| Index of extreme moisture deficit (sum of negative deviations if < -25 mm) | .0000187***<br>(5.300e-06) |
| Observations | 69923 |
| R-squared | .502883 |
| AIC (Akaike's information criterion) | -25900.4 |
| BIC (Bayesian information criterion) | -25836.4 |

Standard errors are in parenthesis & adjusted for state clusters
*** p<0.01, ** p<0.05, * p<0.1

**Notes: Table lists regression coefficients and shows standard errors in brackets. Temperature is in degree Celsius and soil moisture in mm in 1000 mm topsoil. The soil moisture is obtained from WBM at 6 arcmin output while precipitation and temperature are taken from PRISM at 2.5 arcmin. They are aggregated from grid cells to counties based on crop area weight. Yield data is acquired from the USDA. The constant term and coefficients on the interaction of each state and time trends are not reported.**






**Table A2. Estimation of Model (2-d) controlling with irrigation**

|  | (2-d)<br>log CornYield |
|---|---|
| Degree days from 10˚C to 29˚C, $\alpha$ | .0003027***<br>(.0000676) |
| Degree days above 29˚C, $\beta$ | -.0061523***<br>(.0009143) |
| Degree days above 29˚C x S2 (share of heat at normal moisture), $\beta'$ | .0017508***<br>(.0006243) |
| Degree days above 29˚C x S3 (area share of irrigated corn), $\beta''$ | .0023809***<br>(.0007472) |
| Index of deficit (sum of negative deviations if $< -25$ mm), $\eta_{lo}$ | .0000287***<br>(5.200e-06) |
| Index of surplus (sum of positive deviations if $> +25$ mm), $\eta_{hi}$ | -.0000345***<br>(2.800e-06) |
| Index of normal soil moisture, $\eta_{nl}$ | .000046***<br>(.0000101) |
| Observations | 69923 |
| R-squared | .5095193 |
| Akaike's Crit | -26840.2 |
| Bayesian Crit | -26776.1 |

Standard errors are in parenthesis
*** p<0.01, ** p<0.05, * p<0.1

**Notes: Table lists regression coefficients and shows standard errors in brackets. Temperature is in degree Celsius and soil moisture in mm in 1000 mm topsoil. The soil moisture is obtained from WBM at 6 arcmin output while precipitation and temperature are taken from PRISM at 2.5 arcmin. They are aggregated from grid cells to counties based on crop area weight. Yield data is acquired from the USDA. The constant term and coefficients on the interaction of each state and time trends are not reported.**






**Table A3. Descriptive statistics of main variables for Eastern and Western US**

| Variables | East | | West | |
|---|---|---|---|---|
| | Mean | Std. Dev. | Mean | Std. Dev. |
| Degree days from 10°C to 29°C | 1877.79 | 433.54 | 1612.74 | 363.57 |
| Degree days above 29°C | 58.01 | 57.13 | 82.11 | 80.29 |
| Cumulative precipitation Apr-Sep (mm) | 601.13 | 153.31 | 271.69 | 132.12 |
| Mean daily soil moisture content (mm) | 50.49 | 39.49 | 15.15 | 13.17 |
| Number of days with high soil moisture | 28.89 | 30.38 | 8.69 | 11.57 |
| Number of days with low soil moisture | 30.39 | 35.46 | 2.97 | 7.1 |
| Surplus (sum of positive daily deviation, mm) | 2546.95 | 2177.62 | 964.98 | 938.69 |
| Deficit (sum of negative daily deviation, mm) | -2563.43 | 2200.22 | -962.27 | 699.6 |
| Degree days from 10°C to 29°C & low soil moisture | 442.94 | 433.88 | 29.08 | 94.27 |
| Degree days from 10°C to 29°C & high soil moisture | 364 | 351.68 | 62.88 | 90.52 |
| Degree days from 10°C to 29°C & normal soil moisture | 1067.65 | 573.28 | 1462.24 | 426.27 |
| Degree days above 29°C & low soil moisture | 20.19 | 32.55 | .85 | 3.22 |
| Degree days above 29°C & high soil moisture | 5.17 | 9.34 | .76 | 2.41 |
| Degree days above 29°C & normal soil moisture | 32.24 | 41.87 | 72.91 | 72.8 |
| Index of extreme deficit | -1823.19 | 2339.6 | -160.91 | 597.29 |
| Index of extreme surplus | 1942.11 | 2207.68 | 482.25 | 770.15 |
| Index of normal soil moisture | -194.99 | 516.76 | -406.16 | 434.96 |
| Mean daily evapotranspiration (mm) | .6 | .59 | .15 | .19 |
| Mean daily soil moisture fraction | .71 | .18 | .68 | .2 |
| Mean daily soil moisture content (mm), alternative | 50.52 | 39.41 | 15.17 | 13.2 |
| Mean daily soil moisture content (mm), Apr-May | 21.82 | 16.5 | 6.29 | 6.75 |
| Mean daily soil moisture content (mm), Jun-Jul | 17.7 | 15.77 | 5.14 | 4.53 |
| Mean daily soil moisture content (mm), Aug-Sep | 10.98 | 10.74 | 3.72 | 3.27 |
| Observations | 62094 | 62094 | 7829 | 7829 |





**Table A4. Estimation of Model (1) for the East**

|  | (1-a)<br>Log<br>CornYield | (1-b)<br>Log<br>CornYield | (1-d')<br>Log<br>CornYield | (2-c)<br>Log<br>CornYield |
|---|---|---|---|---|
| Degree days from 10°C to 29°C | .0003108***<br>(.0000936) | .0003152***<br>(.0000868) | .0003072***<br>(.0000724) | .0002308**<br>(.000088) |
| Degree days above 29°C | -.0056293***<br>(.0007259) | -.0054707***<br>(.0007343) | -.0052882***<br>(.0006442) | -.0056523***<br>(.000946) |
| Cumulative precipitation Apr-Sep (mm) | .0009245***<br>(.0002502) |  |  |  |
| Square of cumulative precipitation Apr-Sep | -7.000e-07***<br>(2.000e-07) |  |  |  |
| Mean daily soil moisture content (mm) |  | .00319***<br>(.0006763) |  |  |
| Square of mean daily soil moisture content |  | -.0000158***<br>(3.000e-06) |  |  |
| Index of extreme deficit |  |  | .0000379***<br>(5.700e-06) | .0000183***<br>(5.300e-06) |
| Index of extreme surplus |  |  | -.0000381***<br>(2.700e-06) | -.0000225***<br>(6.800e-06) |
| Index of normal soil moisture |  |  | .0000292**<br>(.0000112) | .0000433***<br>(.0000107) |
| Degree days from 10°C to 29°C x S1 |  |  |  | .0001296**<br>(.00006) |
| Degree days above 29°C x S2 |  |  |  | .0010785<br>(.000888) |
| Observations | 62094 | 62094 | 62094 | 62094 |
| R-squared | .4997799 | .4989592 | .5205428 | .5277292 |
| Akaike's Crit | -20126.6 | -20024.8 | -22756.9 | -23690.7 |
| Bayesian Crit | -20090.4 | -19988.6 | -22711.8 | -23627.5 |

Standard errors in parenthesis
*** p<0.01, ** p<0.05, * p<0.1

**Notes: Table lists regression coefficients and shows standard errors in brackets. Temperature is in degree Celsius and soil moisture in mm in 1000 mm topsoil. The soil moisture is obtained from WBM at 6 arcmin output while precipitation and temperature are taken from PRISM at 2.5 arcmin. They are aggregated from grid cells to counties based on crop area weight. Yield data is acquired from the USDA. The constant term and coefficients on the interaction of each state and time trends are not reported.**



**Table A5. Estimation of the Model (1) for the West**

| | (1-a) Log CornYield | (1-b) Log CornYield | (1-d') Log CornYield | (2-c) Log CornYield |
|---|---|---|---|---|
| Degree days from 10˚C to 29˚C | .0004426*** (.0000829) | .0004484*** (.0000823) | .0004539*** (.0000862) | .0004289*** (.000084) |
| Degree days above 29˚C | -.0020381*** (.000423) | -.0023744*** (.0004911) | -.0022938*** (.0004752) | -.0020711 (.0013393) |
| Cumulative precipitation Apr-Sep (mm) | .0005768 (.0003372) | | | |
| Square of cumulative precipitation Apr-Sep | -3.000e-07 (5.000e-07) | | | |
| Mean daily soil moisture content (mm) | | .0078908** (.0027432) | | |
| Square of mean daily soil moisture content | | -.0000848** (.0000326) | | |
| Index of extreme deficit | | | .0000255 (.0000271) | .0000217 (.0000342) |
| Index of extreme surplus | | | -9.800e-06 (7.600e-06) | -7.100e-06 (.0000123) |
| Index of normal soil moisture | | | .0000762** (.0000309) | .000077** (.0000334) |
| Degree days from 10˚C to 29˚C x S1 | | | | .0000281 (.0001257) |
| Degree days above 29˚C x S2 | | | | -.000238 (.001542) |
| Observations | 7829 | 7829 | 7829 | 7829˚C |
| R-squared | .2784229 | .2768284 | .2772401 | .2772526 |
| Akaike's Crit | -3050.8 | -3033.5 | -3035.9 | -3032.1 |
| Bayesian Crit | -3022.9 | -3005.6 | -3001.1 | -2983.3 |

Standard errors are in parenthesis
*** p<0.01, ** p<0.05, * p<0.1

**Notes: Table lists regression coefficients and shows standard errors in brackets. Temperature is in degree Celsius and soil moisture in mm in 1000 mm topsoil. The soil moisture is obtained from WBM at 6 arcmin output while precipitation and temperature are taken from PRISM at 2.5 arcmin. They are aggregated from grid cells to counties based on crop area weight. Yield data is acquired from the USDA. The constant term and coefficients on the interaction of each state and time trends are not reported.**





**Table A6. Corn yield estimation with bi-monthly soil moisture metrics**

|  | US Log CornYield | West Log CornYield | East Log CornYield |
|---|---|---|---|
| Degree days from 10˚C to 29˚C | .0003176*** (.0000774) | .0004543*** (.0000853) | .0002921*** (.0000838) |
| Degree days above 29˚C | -.0044571*** (.0006231) | -.0023373*** (.0004904) | -.0047849*** (.0006742) |
| Mean daily soil moisture content (mm), Apr-May | -.0029599* (.0015561) | .0045436** (.002061) | -.0034124** (.0015243) |
| Square of mean daily soil moisture content (mm), Apr-May | -9.800e-06 (.000022) | -.0000564 (.0000581) | -2.600e-06 (.0000216) |
| Mean daily soil moisture content (mm), Jun-Jul | .0141021*** (.0019928) | .0148123* (.0071408) | .013605*** (.0020404) |
| Square of mean daily soil moisture content (mm), Jun-Jul | -.0001589*** (.0000252) | -.0005616** (.0002422) | -.0001562*** (.0000258) |
| Mean daily soil moisture content (mm), Aug-Sep | .0030501* (.001805) | .007007 (.0049266) | .0026044 (.0018059) |
| Square of mean daily soil moisture content (mm), Aug-Sep | -.0000385 (.0000291) | -.000213 (.0002114) | -.0000351 (.0000294) |
| Observations | 69923 | 7829 | 62094 |
| R-squared | .4884616 | .2782172 | .515591 |
| Akaike's Crit | -23898.8 | -3040.6 | -22112.9 |
| Bayesian Crit | -23825.6 | -2984.8 | -22040.6 |

Standard errors are in parenthesis
*** p<0.01, ** p<0.05, * p<0.1

 **Notes: Table lists regression coefficients and shows standard errors in brackets. Temperature is in degree Celsius and soil moisture in mm in 1000 mm topsoil. The soil moisture is obtained from WBM at 6 arcmin output while precipitation and temperature are taken from PRISM at 2.5 arcmin. They are aggregated from grid cells to counties based on crop area weight. Yield data is acquired from the USDA. The constant term and coefficients on the interaction of each state and time trends are not reported.**





**Table A7. Estimating corn yields using ET and SMF from WBM**

|  | Log CornYield | Log CornYield | Log CornYield | Log CornYield | Log CornYield |
|---|---|---|---|---|---|
| Degree days from 10˚C to 29˚C | .0003422*** (.0000752) | .0003445*** (.0000741) | .0003193*** (.0000801) | .0003372*** (.0000751) | .0003426*** (.0000801) |
| Degree days above 29˚C | -.005298*** (.00069) | -.005343*** (.0006681) | -.005017*** (.00064) | -.004884*** (.0006367) | -.005115*** (.0006914) |
| Mean daily soil moisture fraction | .2533803** (.1107891) | .9821037*** (.2394119) |  |  |  |
| Sqr. mean soil moisture fraction | -.1030471 (.1166278) | -.777505*** (.2402404) |  |  |  |
| SD daily soil moisture fraction |  | -.509464*** (.1156073) |  |  |  |
| Mean daily ET* (mm) |  |  | .4901121*** (.0735423) | .6357687*** (.0985801) |  |
| Sqr. mean daily ET* |  |  | -.086206*** (.0234848) | -.118748*** (.0254433) |  |
| SD daily ET* |  |  |  | -.2516986** (.0997848) |  |
| Mean moisture content (mm)** |  |  |  |  | .0036395*** (.0006759) |
| Sqr. mean daily moisture content ** |  |  |  |  | -.000017*** (3.000e-06) |
| Observations | 69923 | 69923 | 69923 | 69923 | 69923 |
| R-squared | .4667911 | .4712361 | .4755177 | .4770727 | .4713225 |
| Akaike's Crit | -21005.7 | -21589.0 | -22159.5 | -22365.1 | -21602.5 |
| Bayesian Crit | -20969.0 | -21543.3 | -22122.9 | -22319.4 | -21565.9 |

Standard errors in parenthesis
*** $p<0.01$, ** $p<0.05$, * $p<0.1$

**Notes: Table lists regression coefficients and shows standard errors in brackets. Temperature is in degree Celsius and soil moisture in mm in 1000 mm topsoil. The soil moisture is obtained from WBM at 6 arcmin output while precipitation and temperature are taken from PRISM at 2.5 arcmin. They are aggregated from grid cells to counties based on crop area weight. Yield data is acquired from the USDA. The constant term and coefficients on the interaction of each state and time trends are not reported.**





**Table A8. Corn yield estimation with the interaction of heat and soil moisture**

|  | (US) log CornYield | (West) log CornYield | (East) log CornYield |
|---|---|---|---|
| Degree days from 10˚C to 29˚C | .0003083*** (.0000685) | .0004344*** (.0000847) | .0002963*** (.0000736) |
| dday29˚C & SM 75+ mm below normal (extreme deficit) | -.0082398*** (.0014372) | -.0074467* (.0035727) | -.0082928*** (.0014365) |
| dday29˚C & SM 25-75 mm below normal (deficit) | -.0062069*** (.0009793) | -.0033152* (.001627) | -.0061966*** (.0009797) |
| dday29˚C & SM 0-25 mm around normal (normal) | -.0037559*** (.0004045) | -.0024412*** (.0005053) | -.0041335*** (.0004376) |
| dday29˚C & SM 25-75 mm above normal (surplus) | -.0055709*** (.0012041) | -.004754* (.0024763) | -.005625*** (.0011677) |
| dday29˚C & SM 75+ mm above normal (extreme surplus) | -.0140295*** (.0019083) | .0095881 (.0128016) | -.0143573*** (.0018101) |
| Mean daily soil moisture content (mm) | .0026635*** (.0008153) | .0080027** (.0028858) | .0025636*** (.0008324) |
| Square of mean daily soil moisture content | -.0000161*** (2.600e-06) | -.0000844** (.0000326) | -.0000156*** (2.600e-06) |
| Observations | 69923 | 7829 | 62094 |
| R-squared | .4921263 | .2777862 | .5149811 |
| Akaike's Crit | -24401.6 | -3035.9 | -22034.8 |
| Bayesian Crit | -24328.3 | -2980.2 | -21962.5 |

Standard errors in parenthesis
*** $p<0.01$, ** $p<0.05$, * $p<0.1$

**Notes: Table lists regression coefficients and shows standard errors in brackets. Temperature is in degree Celsius and soil moisture in mm in 1000 mm topsoil. The soil moisture is obtained from WBM at 6 arcmin output while precipitation and temperature are taken from PRISM at 2.5 arcmin. They are aggregated from grid cells to counties based on crop area weight. Yield data is acquired from the USDA. The constant term and coefficients on the interaction of each state and time trends are not reported.**





**Table A9. Estimation while splitting the soil moisture indicators**

|  | (US) log CornYield | (West) log CornYield | (East) log CornYield |
|---|---|---|---|
| Degree days from 10°C to 29°C | .0003154*** (.0000689) | .0004451*** (.0000919) | .0002983*** (.000074) |
| Degree days above 29°C | -.004044*** (.0005384) | -.0020707*** (.0005793) | -.0044516*** (.0005981) |
| Index of normal soil moisture when T > T* | .0001199*** (.0000342) | .0001805 (.0001426) | .0001034*** (.0000358) |
| Index of extreme moisture surplus when T > T* | -.0000628*** (.0000151) | -.0001173 (.0001071) | -.0000586*** (.0000149) |
| Index of extreme moisture deficit when T > T* | .000092*** (.0000234) | -.0000526 (.0000978) | .0000817*** (.0000229) |
| Index of extreme moisture deficit when T < T* | .0000209*** (7.100e-06) | .0000287 (.0000337) | .0000223*** (7.000e-06) |
| Index of extreme moisture surplus when T < T* | -.0000326*** (3.200e-06) | -5.700e-06 (6.500e-06) | -.0000334*** (3.200e-06) |
| Index of normal soil moisture when T < T* | .000028** (.0000105) | .000063** (.0000249) | .0000247** (.0000102) |
| Observations | 69923 | 7829 | 62094 |
| R-squared | .5006312 | .2782242 | .5262193 |
| Akaike's Crit | -25582.4 | -3040.6 | -23490.5 |
| Bayesian Crit | -25509.2 | -2984.9 | -23418.2 |

Standard errors in parenthesis
*** p<0.01, ** p<0.05, * p<0.1

**Notes: Table lists regression coefficients and shows standard errors in brackets. Temperature is in degree Celsius and soil moisture in mm in 1000 mm topsoil. The soil moisture is obtained from WBM at 6 arcmin output while precipitation and temperature are taken from PRISM at 2.5 arcmin. They are aggregated from grid cells to counties based on crop area weight. Yield data is acquired from the USDA. The constant term and coefficients on the interaction of each state and time trends are not reported.**





(a) dynamics of soil moisture conditions

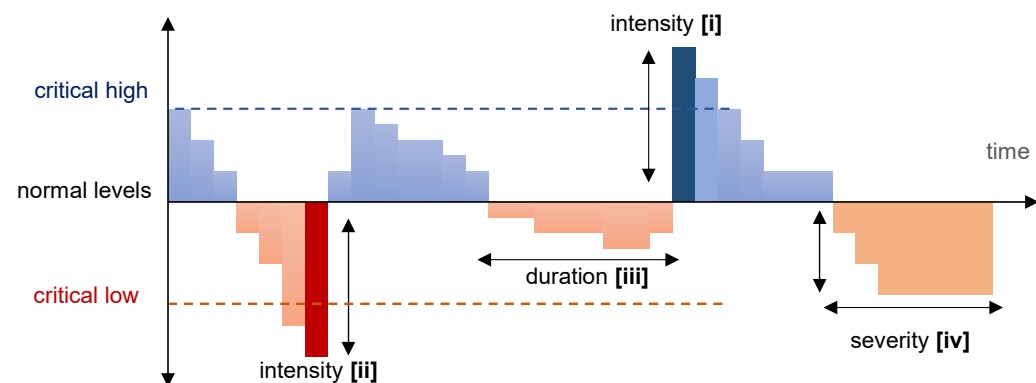


(b) soil moisture conditions and adaptations

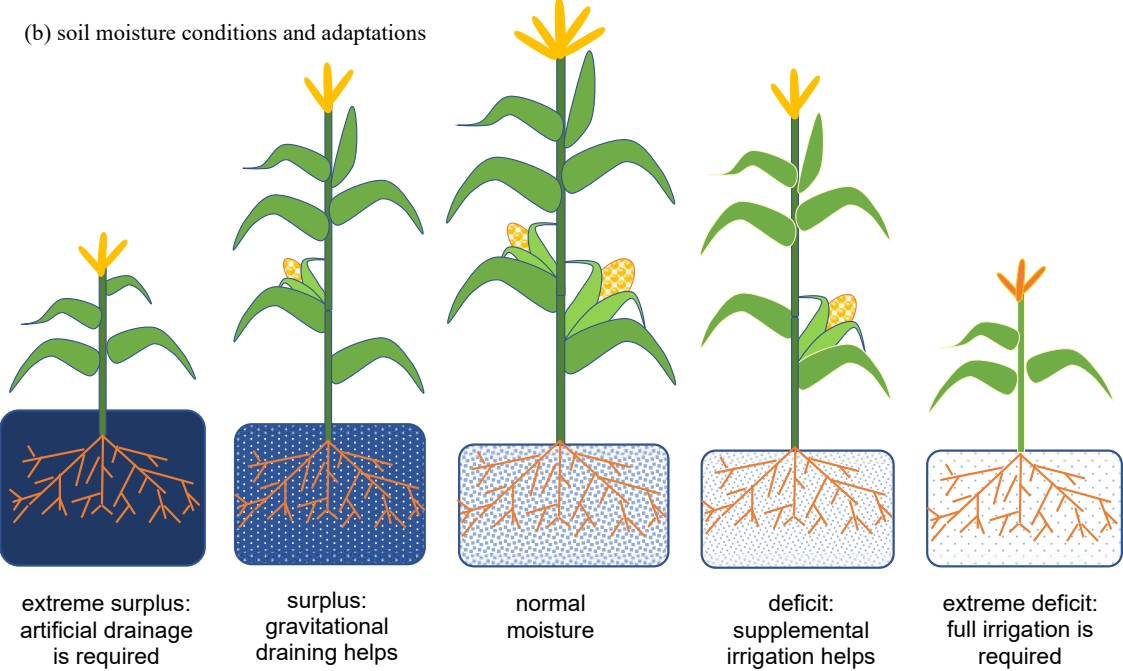

**Figure 1. (a) Soil moisture dynamics within a typical growing season. Some soil moisture conditions can be harmful to crops including excess wetness [i], moisture stress intensity[ii], duration of moisture stress [iii], and severity of soil moisture stress [iv]. (b) Adaptation mechanisms can reduce the damage to crops. As flood can cause severe damage to corn, artifitial drainage is required; excess water may slow down the growth; normal soil moisture makes optimum growth; water deficit can limit the growth, while supplemental irrigation can help; during an extreme water deficit, irrigation is necessary.**



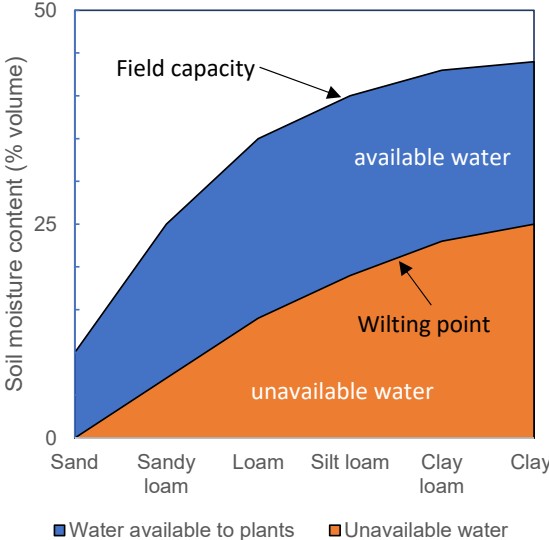

**Figure 2. Soil texture affects the wilting point, field water-holding capacity, and the moisture available to plants. This suggests that sandy soil has the lowest wilting point as well as low field capacity. As most of the water infiltrates, this leaves a little amount of moisture available to plants (Tsoar, 2005).**





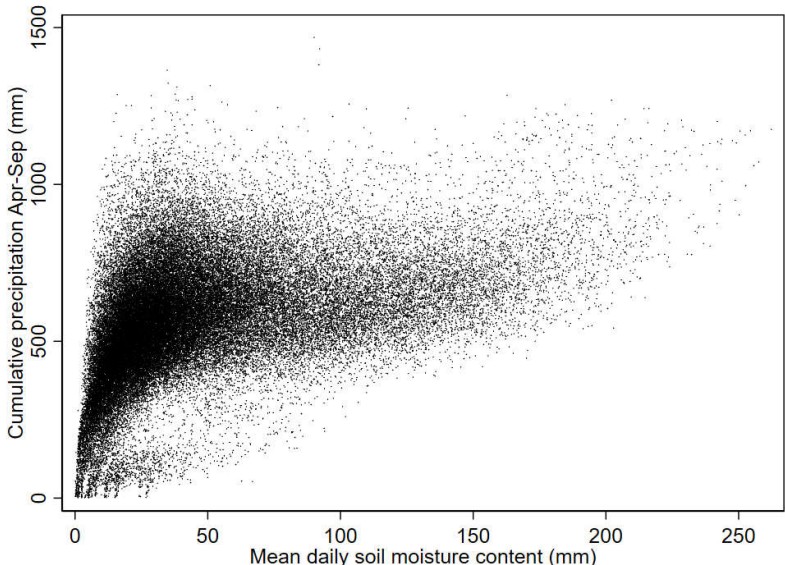

**Figure 3. WBM mean soil moisture versus PRISM cumulative precipitation for 1981-2015 by US counties.**




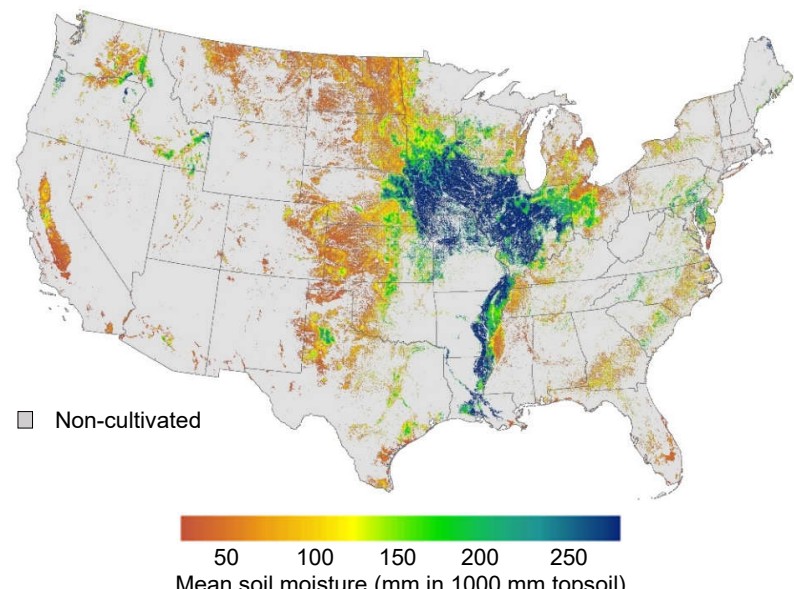

**Figure 4. Growing season mean soil moisture content (in mm in 1000 mm topsoil) as calculated based on daily root-zone soil moisture level from Apr-Sep for 1981-2015 at 2.5 x 2.5 arcmin grids excluding non-cultivated area. The soil moisture level is obtained from the Water Balance Model (WBM) and non-cultivated area information is from USDA National Cultivated Layer. This map illustrates the heterogeneity of simulated soil moisture over the Continental US and even within states.**



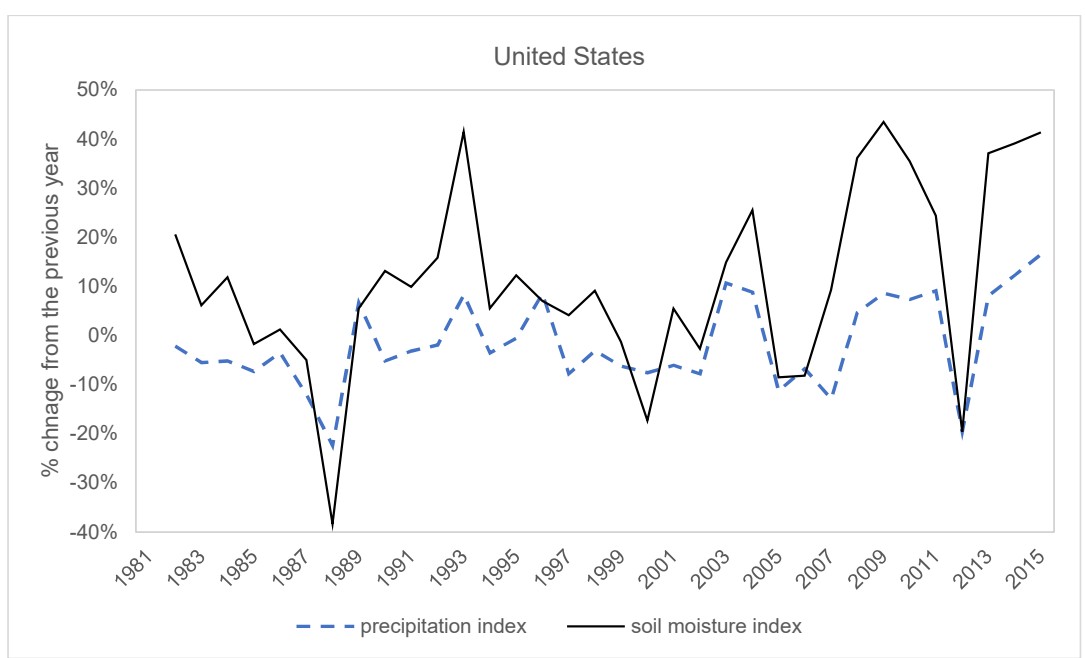

**Figure 5. Variations of average precipitation versus average soil moisture over corn areas in the United States. The precipitation is aggregated from PRISM and soil moisture is aggregated from WBM from 2.5 arcmin grid cells weighted by cropland area.**





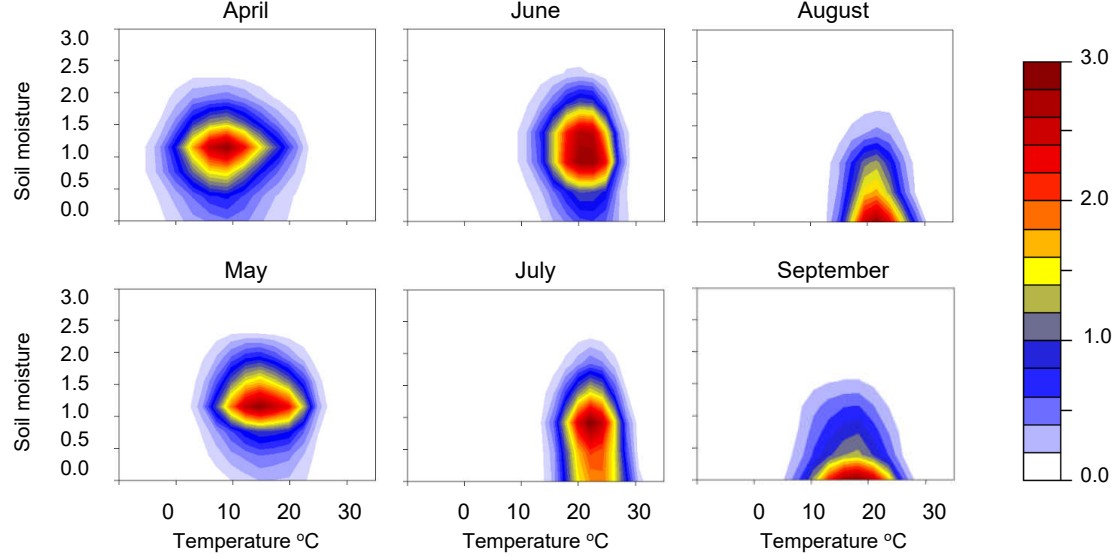


**Figure 6. The bivariate density of heat and soil moisture for 1981-2015 For all the grid cells in the US Corn Belt. The precipitation is aggregated from PRISM and soil moisture is aggregated from WBM based on 2.5 arcmin grid cells.**





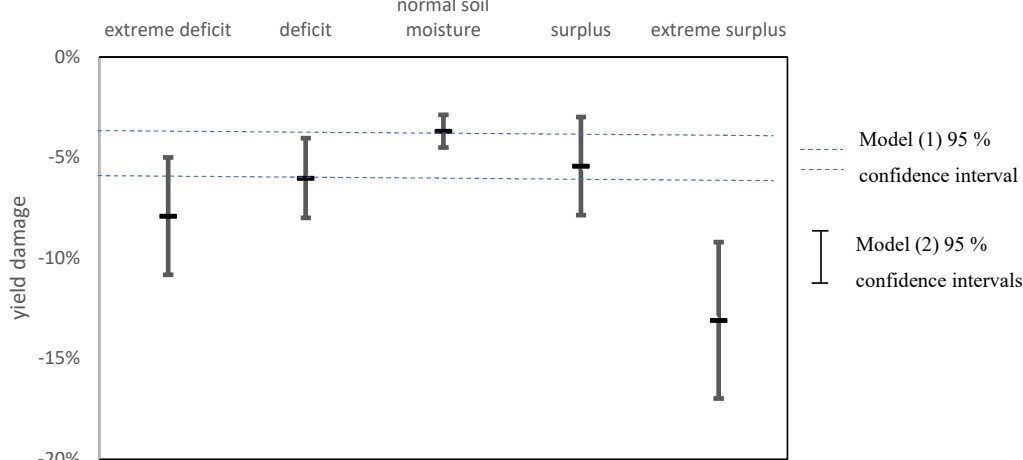

**Figure 7. Estimated damage to corn yield from an additional 10 degree-days above 29˚C and no change in seasonal mean soil moisture.**


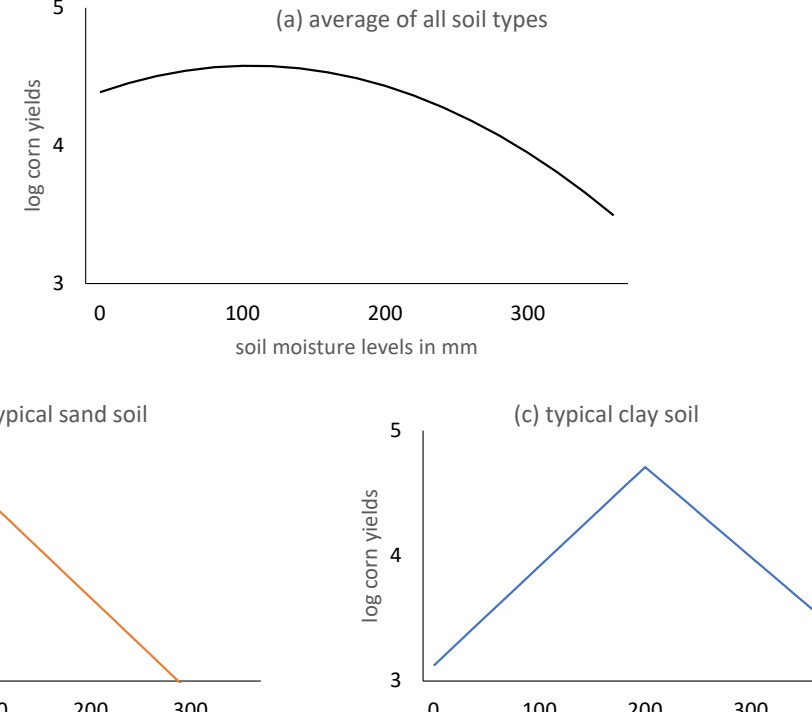

**Figure 8. Estimated impact of soil moisture on log corn yields. Including soil moisture in the regression and its square term, as in model 1-b, will give us a quadratic relationship between soil moisture and yields as in panel (a). A piece-wise linear parametrization, as in model 1-d, can provide location-specific piece-wise linear relationship based on soil moisture deviation from normal as in panels (b) and (c). This will cause the maximum of the response curve to be in lower soil moisture levels for sand and in higher soil moisture levels for clay soil texture.**




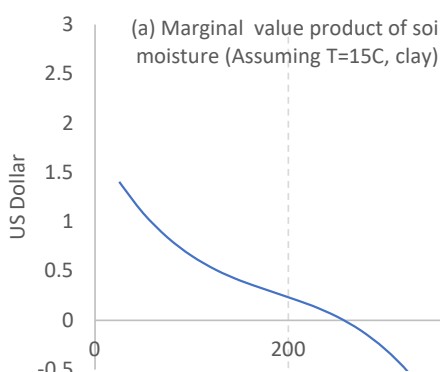
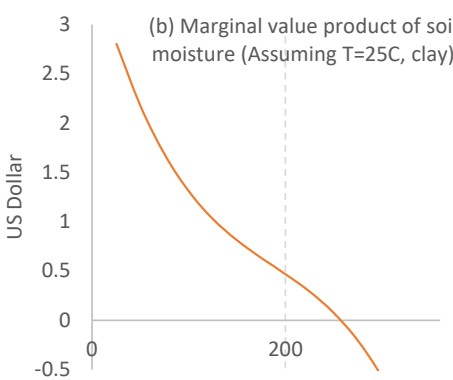

**Figure 9. The marginal value product of soil moisture in clay soil with normal soil moisture of 200 mm for hot days (average temperature = 25˚C) vs moderate days (average temperature = 15˚C).**






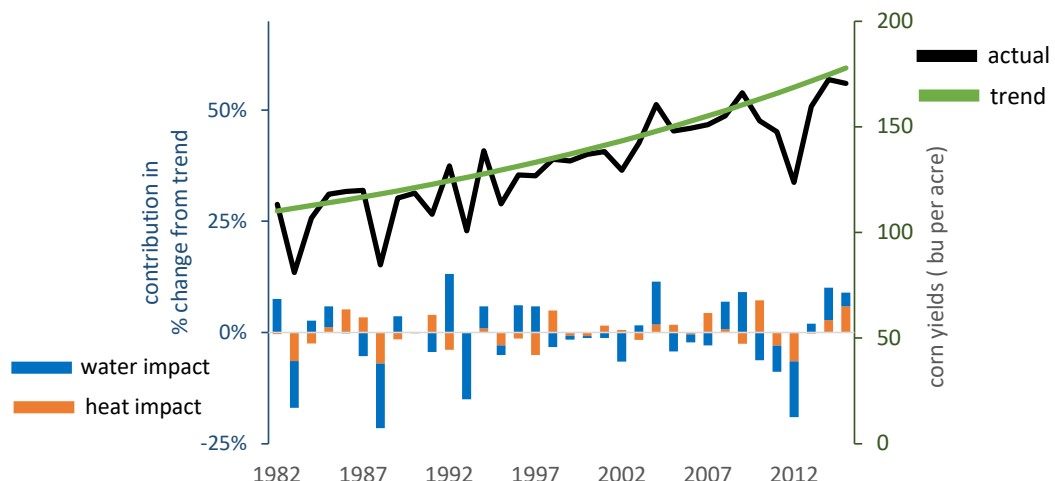

**Figure 10. The bars show the "contribution of water" and "contribution of heat" in variation of US corn yields (left axis). The lines illustrate actual yields and trend (right axis). The impact of other factors is not reported.**



(a) with full irrigation

(b) without irrigation

-70  -60  -50  -40  -30  -20  -10   0   10  20
% change in corn yields


**Figure 11. The impacts of climate change on corn yields with irrigation adaptation illustrated in panel (a) and without irrigation adaptation is shown in panel (b). The maps show the percentage change in corn yield due to climate change from 1976-2005 to 2036-2065 based on CMIP5 RCP 8.5 from NEX-GDDP climate product and estimated parameters in this study.**





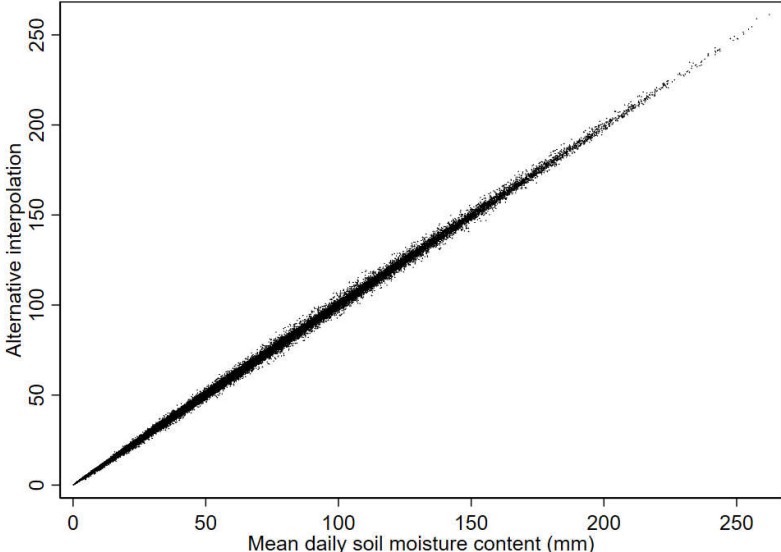


**Figure A1. County-level mean seasonal soil moisture based on bilinear interpolation versus alternative interpolation (nearest-neighbor) from WBM 6 arcmin grids to PRISM 2.5 arcmin resolution for the 1981-2015 period.**


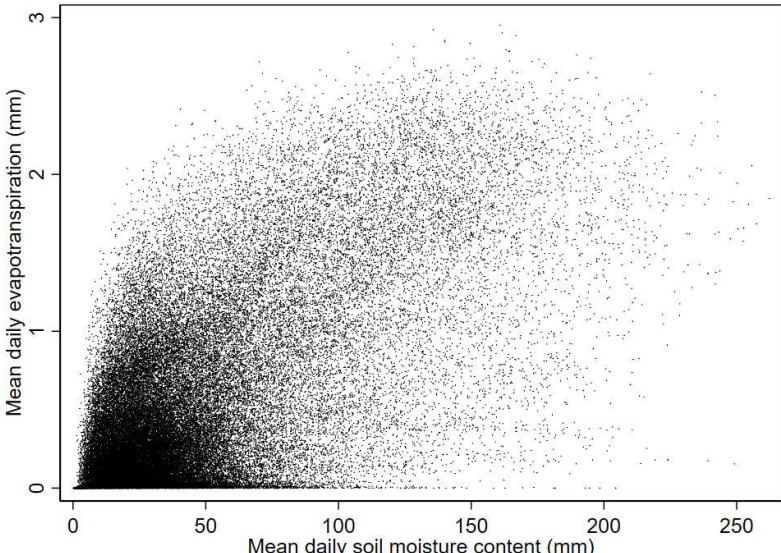

**Figure A2. County-level mean soil moisture versus mean ET aggregated from WBM for the 1981-2015 period.**




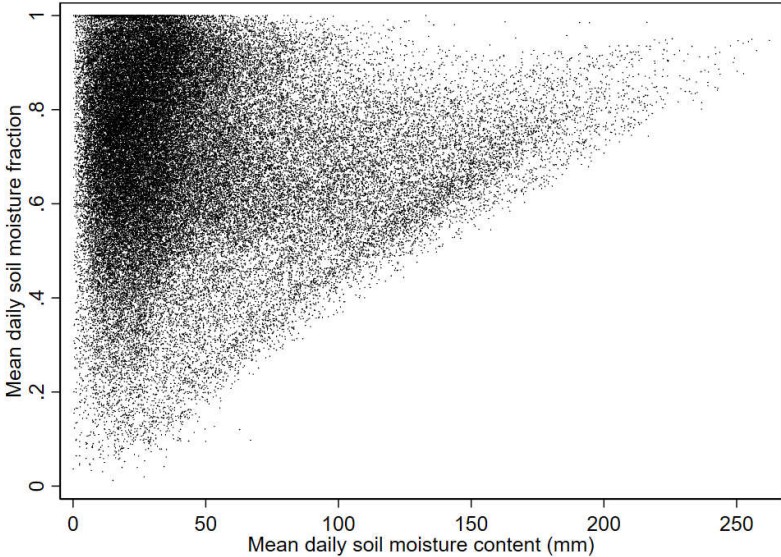

**Figure A3. County-level mean volumetric soil moisture content versus mean of soil moisture fraction aggregated from WBM for the 1981-2015 period.**



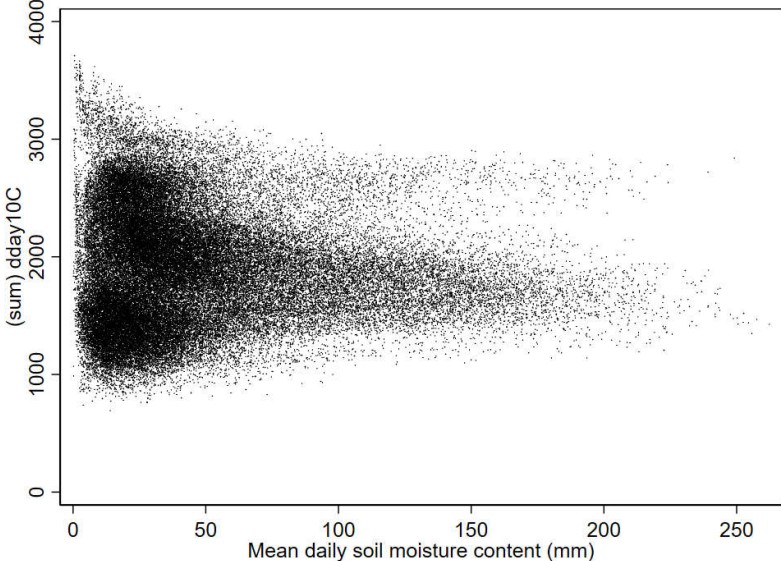


**Figure A4. County-level seasonal mean soil moisture versus seasonal heat index aggregated from WBM and PRISM for the 1981-2015 period.**