# Peer review of "Quantifying the Impacts of Compound Extremes on Agriculture and Irrigation Water Demand"

_Hydrology and Earth System Sciences, 2020_

## Referee Comment (RC1) · Anonymous Referee #1 · 24 Aug 2020

The paper provides a novel approach to quantify the compounding effects of soil moisture and heat stressors on crop yield in the US over a historical time period. The study investigates multiple statistical representations to try to tease out the importance of the interactions between heat stress and soil moisture conditions on crop yield, and takes advantage of a large scale hydrologic model to extract the necessary soil moisture data to build the various models. The paper looks technically sound, and the paper is a great contribution to the literature. However, the paper could almost be cut in half to get the key messages of the paper, and much of the text can be either moved to supplementary materials or completely omitted. For example, I would suggest moving the first 5 figures to supplementary materials. I struggled with the flow of the ideas and text, and there is quite a bit of redundancy, and unnecessarily verbose. I would

recommend major revisions, with most of the efforts on rearranging and streamlining the flow of the paper.

Specific comments:

Lines 1-2: I would suggest changing the title to something like "Quantifying the compounding effects of soil moisture and heat on crop yield" The paper does not talk about the impacts on irrigation water demand, and none of the figures show results looking at water demands.

Line 12: Are high-resolution and fine-scale intended to mean different things?

Introduction: The introduction section needs better arrangement for better flow of the ideas. I would recommend focusing on the importance of this work, what is the current state of the knowledge in this space, how this differs or builds on previous efforts, the key novelties it adds to the field, and the specific science/research questions it is trying to tackle. All of this is pretty much there, but it needs to flow better, and certainly results should be omitted from the introduction section to avoid redundancies.

Lines 22-32: this section is redundant with some of the content of the abstract and talks about the approach and key messages before even articulating the importance of the work. I would recommend omitting.

Lines 57-62: "We show that the coefficient . . ." Avoid throwing results in the middle of the introduction section to avoid redundancy. I would suggest omitting.

Line 79: it is a bit weird to talk about concerns before even talking about the approach.

Line 82: spell out Sec for consistency sake.

Line 83: Do you mean "background: Key factors impacting yield" or something along that line?

Line 85: "before starting our discussion" please rephrase.

Line 94: "we will briefly talk about" please rephrase

Lines 97-101: I would suggest omitting this paragraph. Water is discussed in section 2.3, and here the focus is on spatial aggregation.

Line 100: a sample of what? The sentence is somewhat vague.

Lines 105-110: "we construct our..." this reads like a methodology section and should be part of section 3.

Line 111: "another empirical challenge" This reads like you are talking about a different challenge than what was discussed in the above two paragraphs under 2.1. I would suggest separating this section as '2.2 Degree of temporal aggregation' and keeping the previous sub-section on the spatial aspect only.

Line 130: if you are going to end of each challenge with how this study tackles this challenge or differs from previous efforts, then I would suggest that this is done here as well, and at the end of each of the other challenges discussed in section 2.

Line 144: "To undertake..." please omit sentence. It does not add much.

Lines 145-147: Omit. I would suggest not throwing results at this stage. Plus, the reader does not know anything about WBM yet.

Line 166: "a fixed effect panel regression" I am not sure what that means. Please explain. Also, in the following sentence, what coefficients are you referring to? Please be specific.

Line 182: rephrase "as we prefer to take care of..."

Line 189: this section needs a concluding sentence to connect the dots.

Line 217: having read through section 2, it leaves the reader wondering what all of this has to do with compounding extremes. I wonder if section 2 can be shrunk or moved to a later section after describing the method section of the paper to improve the flow

of the paper.

Line 219: "we introduce two models." What kind of models? Please specify.

Lines 219-221: why design the two models in this manner? Explain the logic.

Line 226-227: "in summary," omit. You just started talking about the model here.

Line 229: "as reported by WBM" omit since the reader has not read about WBM yet unless you go with my recommendation to have section 3.3 moved to 3.1 as explained later.

Lines 227-233: these equations (1a-d) need to be shown here. They are core to the whole paper and deserve more attention in the paper.

Lines 225-234: are metrics, indicators, and water variables the name thing here?

Line 243: I would suggest making this sub-section (3.3 Data) as the first sub-section in the methods section for better flow. Sub-section 3.4 builds nicely on what's covered under the first two subsections, and the data comes in the middle and breaks the flow.

Line 250: "Daily interaction" how is this defined or calculated? Is this a term in equation 2? If so, then please state so.

Lines 261-265: omit this paragraph. It is identical to lines 249-255.

Line 231: Was there any validation work done on the soil moisture data? I am not necessarily asking for that to be shown here, and rather some citations of the previous validation work using WBM should suffice.

Line 304: are you using a single scan, or are you capturing the evolution of the cropland over the historical time period?

Line 319: I would suggest moving the materials here to be merged with subsections 3.1 and 3.2. For example, I would suggest moving lines 320 to 331 to appear in line 234. This would mean deleting the sentence "the estimation strategy is described in

section 3.4." Similarly, I would move lines 232 through 361 to line 242.

Line 363: "This section provides estimation results for different representations of Model (1)" well the authors discuss results from Model (2) as well (starting around line 389).

Line 384: so what does all of this mean? Which is the 'best' model formulation, on what basis, and how does this compare with previous findings?

Line 387: "the deficit and by 2300" – delete 'and'

Line 397: "The figure shows that Model (1) would..." It was not clear that the intent was to compare the two models (1 and 2) to get at the compounding aspect. Some articulation of that upfront would help the reader follow through.

Line 44: "from previous models" which models are you referring to? 1a,b,c,d, 2a?

Lines 409-422: I would suggest moving this to be part of the results section. And to change the title for section 5 to be simply "Discussion" and then to jump to 5.1 directly.

Lines 416-420: please expand on this section to explain what you found out from these additional analyses that are in the appendix. Currently, they come across as throw away sentences.

Lines 420-422: "Finally, we have provided . . ." I would omit these two sentences.

Line 432: "we recommend the use of soil . . ." I can't tell if this recommendation is based on the findings in this study, or simply an opinion based on past efforts/studies. Please clarify.

Line 454: "Model (2)" 2a or 2b?

Line 454: "while Model (3) predicts ..." do you mean model 1 here?

Lines 484-499: Subsection 5.4 comes as a surprise to the reader. It also reads more like a methods section. I would suggest dropping this subsection.

Lines 500-509: The first sentence is redundant. The subsection is relatively shallow as compared to previous subsections. Also, it is not clear if there is any conclusion that can be drawn from Figure 10. I wonder if this would fit better if moved to the results section instead of being a discussion subsection.

Lines 510-536: Subsection 5.6 is another big surprise to the reader. I was not expecting this as this was never baked in the framing of the paper in the initial sections. All the previous sections including the data subsection focused on the US. Though this extends the work globally, which begs the question of how the extrapolation was done. I would suggest omitting this from this paper and keeping it for a follow-on paper.

Figure 6: what are the units for the y-axis and for the color bar on the far right?

---

## Referee Comment (RC2) · Anonymous Referee #2 · 25 Aug 2020

Overall:

The paper is about a well-designed study aiming to elaborate individual and compound extreme event impacts on corn yields in the USA using statistical approach. The significance of extreme events on yield anomalies were studied using various indicators of soil moisture (representing water stress) as well. The outcomes of the paper can be insightful for further studies of predicting crop yield anomalies and assessing impacts of extreme weather conditions to crop yields. Consequently, the paper is worth for publishing with some revisions.

My major comments on the paper are:

1- The paper needs to be re-structured/re-written. First, it is too lengthy including text-

book information (e.g. Figure 1b, and Figure 2) which are not necessary for the reader (peer knowledge). Second, its structure is chaotic: the introduction chapter includes results and discussions points etc; it is like a short summary of the whole paper; the discussion section includes equations, methods, results and data sources. The authors claim to include results/conclusions which are too diverse and out of scope of the analysis (e.g. irrigation, farm soil management, marginal value, decision making as specified in the abstract). The framework of analysis do not support to make conclusions about these topics. The authors should revise their goals and associated conclusions accordingly. The paper is about compound vs individual extreme events on crop yield and comparison of different soil moisture indicators. Other conclusions not taken from this analysis can be excluded. Furthermore, the empirical concerns are relevant however too lengthy for readers. It can be reduced and can be removed to SI.

2- The authors claim that "marginal value of water" will be calculated and utilized in the paper. There is nothing about it in the method and result section (only shown in the discussion section – a short paragraph without any substantial info). I think having this goal of economic analysis is not relevant and beyond the scope the. It is better to exclude this part of the analysis so that the paper is coherent and consistent with its framework.

3- Discussion sections were boldly written (e.g. like for climate change discussion and farmer management). I recommend drawing conclusions only if it is supported by the data and analysis.

For more-detailed comments:

1) Abstract

- which crops were addressed in the article? Please specify. It is important to mention corn here. - "the value of water experiences a four-fold increase on hot days": not clear, what do the authors refer to by "value of water"? Is this volume? Value of water is generally associated with significance, importance, true cost etc. - This paper also

improves our understanding of the conditional marginal value (or damage)". Which way? And what is conditional marginal value? It is important to provide necessary descriptions in the text as well.

2) Introduction

- The first paragraph was written like a conclusion section (after line 26). It includes a short summary, reminding "an abstract". This part needs revision or can be completely excluded (or moved to discussion/conclusion sections). - Ln 33: there can be other factors affecting crop yield significantly such as soil, management, nutrients etc. - Ln 37-38: "Other metrics of extreme water conditions", please specify. - Ln 38-39: "Current statistical studies had limited success in statistically capturing the yield response to soil moisture metrics", please explain why. - Ln 43: "the impact of climate change on soil moisture". The paper is about individual extreme response of yield vs compound. It is not clear why the authors refer to CC studies. - Ln 46: "conditional marginal impact". Please explain what this means. - Ln 50: please explain "wet-heat stress" - Ln 55-60: this part is an outcome of the study. Please remove it to another section (e.g. discussion). - What are exactly marginal and conditional marginal impacts? It is better if definitions are given for readers. - Ln 64-79: this part is related to discussion/conclusion. I recommend deleting these parts or move to the other relevant sections. - Ln 77/78: the authors claim that they will show how the results can be used to economically quantify the marginal value of water, in the form of soil moisture, for corn production in the US under different hydroclimatic conditions. I couldn't see this in the rest of the paper. Please clarify.

3) Empirical concerns

- This section is mostly about discussion of the method and assumptions taken for the study. It can be presented as supplementary information, rather than in the main text. That can help reader to focus on the results of the paper and its wider implications. In its current form, it is too lengthy. - Equation 1: please describe what exactly each letter

in the equation refers to? For example please refer last variable in the equation as error and describe g(h) function? - Ln 126: "measure the value of water". Not clear what the authors refer to as "value of water". Please clarify. - Figure 1: this is nothing new, a known information– like a textbook. Excluding this figure does not change anything about the paper. I recommend not to include it. - Ln 134: "Many researchers have acknowledged the need for soil moisture data to predict the response of crop yields to variations in water availability." Please provide references to those researchers. - Ln 171: please provide references to those studies.

4) Method - This section is too long. Please shorten it and provide detailed information in SI. - Equation 2: please define each variable and function used in the equation. - Ln 230: "some indicators", please clarify which indicators. - Please provide numbers to the equations. - Ln 277: g(Ws), please define the parameter - Ln 290-295: this is a result of the analysis, not related to data/method or assumptions. - Figure 4,5 and 6 are outcomes of the model/analysis. They can be presented in the result section.

5) Results - Ln 363/364: "We will discuss the implications of these results in Sect. 5." The authors use lots of cross references between the sections as seen in here. This is not necessary, since discussion section means discussion of the results by definition. Please through the entire text and remove unnecessary cross-section references. - Table 2: note section is repetition of the previous sections, thus it is not necessary. - Ln 404: "This indicates that water is up to four times more valuable in hot weather." The authors can consider revising the sentence and be more explicit, "value of water" may mean several things. - Model (2-a) and Model (2-b) were mentioned here for the first time. Please describe the differences between these models in method/data section.

6) Discussion

- Ln 410/411: this is related to differences between model 1 & 2, right? Please clarify which model outcome supports (or all?) the statement. - Performance: does this mean best correlation between indicators of extreme events yield anomalies? Please

clarify. - First paragraph: what about model 1-c ? - Model 2 a-b were not defined in the previous parts of the paper. Please check consistency. - Ln 416-421: These are newly introduced topics. None of these research goals (including why to have them), methods and results were mentioned in the previous sections of the paper (e.g. new interaction model, why do you have that and this was never mentioned in the paper). It is like Appendix is another paper with its own results, methods and goals. Please revise the paper accordingly. - Ln 424-428: Is this an outcome supported by the results? If so, please indicate how. It is more like a general knowledge. - Ln 429-430: Please provide supporting data/result from the analysis. - Ln 433: what are the other metrics suggested in the literature? - Ln 434-438: Is this a conclusion related to compound vs individual extreme weather event analysis? Can we say the same if we use other metrics of water stress than soil moisture? - Ln 465-469: I question that the authors' research is critical for climate change studies. First, their analysis was based on historical data and says nothing about counterfactual analysis. This is not the first time impacts of a compound event was researched and like other studies this paper shows stronger impact of a compound event. It does not bring anything to climate change impact studies. - Ln 472: please clarify benefit of this collaboration. In which way it helps to solve the challenge. - Ln 479-ln 483: this recommendation is not related to the sub-section heading. The authors stated a discussion point which is out of scope of their analysis and not supported with the overall goal of the paper. Recommendations can be given to farmers etc; however their model/research is not aimed for decision -support guidance. Please remove this section of revise it. - Section 5.4: This section includes literature, method, data and equation related to an estimation. This is not a discussion section. Please previse it accordingly. This additional analysis doesn't bring anything to the value of the paper. I would recommend excluding this analysis from the paper in order to keep its coherence and consistency. - Ln 501: "We find that the average damage from excess heat has been up to four times more severe when combined with water stress" what is the damage, yield losses? - Line 517-525: the CC knowledge and analysis were not included in previous parts (method, data, results) section

of the paper. Please include info about this analysis in adequate sections. - Line 525-535: There is almost no economic analysis thus the paper does not contribute to CC economics. No policy analysis or research were provided either; also paper does not say/bring anything to regional resilience of agroecosystems, global food security, and as well as future climate impacts. These two paragraphs have to be re-written. These claims are bold and cannot be taken from the research as described in the paper.

---

## Author Comment (AC1) · 23 Sep 2020

**Comments and Responses to Anonymous Referee #1** *(Reviewer comments in italics)*

*Comment: The paper provides a novel approach to quantify the compounding effects of soil moisture and heat stressors on crop yield in the US over a historical time period. The study investigates multiple statistical representations to try to tease out the importance of the interactions between heat stress and soil moisture conditions on crop yield, and takes advantage of a large scale hydrologic model to extract the necessary soil moisture data to build the various models. The paper looks technically sound, and the paper is a great contribution to the literature.*

We would like to thank the referee for his/her helpful comments that helped to improve the manuscript. We have revised the paper accordingly and provided overall and specific answers below. Also, many thanks for the positive feedback on the technical details and the significance of the paper.

As the majority of the comments are around the organization of the paper, we have revised the flow of the paper and transitions within the sections. We have dropped the sections identified less relevant by the referees. This has resulted in a substantial re-ordering of the material presented, and these changes have substantially shortened the paper as requested by the reviewer. Now, the paper is focused on the main messages. The manuscript introduces the problem by stating the research gap as "current statistical models of crop yield prediction ignore the compound extreme". And we establish the discussion around the main finding that "statistical models ignoring compound hydroclimatic extremes will significantly underestimate the yield response to water in hot days while they will significantly overestimate the yield response to water in moderate days". The referee's comments also helped us identify the unclear terms and less critical ideas. They helped us to improve the cohesion of the writings by providing clarifying definitions for unfamiliar terms and by removing the ideas not critical for the argument. The background information has been moved to the Supplementary Materials. We have also clarified the methods, moved some parts of the appendix to the text, and moved some parts of the Methods section to the Supplementary. These are major changes:

Introduction: We have included some of the text from the section "Empirical concerns" to provide adequate background on the models and metrics of individual and compound hydroclimatic extremes for predicting corn yields. We limited the text on the state of the art in the statistical prediction of corn yields to highlight current shortcomings. We kept the text on the description of the objectives to give a clear view of the originality of the research. We have removed the sentences more relevant to the Results and Conclusion.

Empirical concerns: A shortened version of this section has been merged into "Methods" and "Introduction" sections as follows. The sentences regarding the Schlenker and Roberts (2009) model are moved to the Methods section making the base for our model with individual extremes. The sentences regarding spatial aggregation are removed, we only kept our method for spatial aggregation in the Methods section. The sentences regarding average versus extreme metrics of water availability are moved to the introduction as they show the shortcomings in the current literature and how we are going to address them in the paper. The sentences regarding "interaction of soil moisture and heat" are shortened, rephrased, and moved to the introduction as they are base for our arguments about compound extreme. We have also clarified the meaning of the statistical term "interaction" when it first

appeared in the manuscript. Finally, the sentences regarding measurement errors and endogeneity concerns are moved to Supplementary.

Methods: This section has some minor changes. We re-order the sub-sections introducing the data before the models. Also, technical terms are described including the "panel fixed effect" method, "daily interaction of heat and soil moisture", and "conditional marginal impact". Figures 1-3 are improved to support definitions and methods.

Results: The results from Model 1 (individual extremes) and Model 2 (compound extremes) have not changed. However, we added a couple of sentences to provide a comparison with previous studies. We added two critical subsections here. A new sub-section on "Model comparison" compares the performance of each model in predicting yields and to illustrate why we have estimated different models with different assumptions and different water metrics. It clearly shows the advantages of using a model with compound extremes. Also, a new sub-section on "Robustness checks" describes why we do these checks and what we learn. Figures 4-6 are moved to the Results section with more details.

Discussion: This section is substantially shortened. We dropped contents about methods and results. The section on "implications for climate studies" and the related text is dropped. The section on "implications for irrigation water demand" and the related text is dropped. Based on our findings we argue that "As we find that the coefficient on extreme heat is significantly different when considering soil moisture, it is possible that previous statistical studies have over- or under-estimated the yield impacts". The revised Discussion section is provided below.

In the following sections, we offer detailed responses to each comment.

*Comment: However, the paper could almost be cut in half to get the key messages of the paper, and much of the text can be either moved to supplementary materials or completely omitted. For example, I would suggest moving the first 5 figures to supplementary materials. I struggled with the flow of the ideas and text, and there is quite a bit of redundancy, and unnecessarily verbose. I would recommend major revisions, with most of the efforts on rearranging and streamlining the flow of the paper.*

Overall response: Thank you for these excellent suggestions. These comments helped us to improve the organization of the paper. To minimize redundancies and maximize audience engagement, we re-organized the manuscript. We omitted the less relevant parts in order to focus on the main message. This has resulted in a substantial re-ordering of the material presented, and substantially shortened the paper.

*Comment: "the paper could almost be cut in half… much of the text can be either moved to supplementary materials or completely omitted"*

Regarding the length of the paper, we have shortened the paper substantially from 52 pages (around 19,000 words) to 29 pages (around 10,000 words).

*Comment: "there is quite a bit of redundancy, and unnecessarily verbose"*

Thanks for this comment that helped us to improve the flow of the paper and the cohesion of the writings. We have revised the organization of the paper. The flow of the Introduction section has been

80  revised as you will see from the following responses. We have omitted the contents related to the conclusion, discussion, and summary from the Introduction. The Discussion section has been revised substantially as you will see below. We have omitted the equations, methods, and results type of content from it. In the revised version, we have focused on the main message. We have revised the flow of the paper focusing on the significance of compound extreme metrics and their advantage over the

85  individual extreme metrics.

*Comment: "I would suggest moving the first 5 figures to supplementary materials"*

Regarding the figures, we have dropped panel b from figure 1. We also moved figure 3 to the Supplementary. However, Figure 2 is revised to illustrate the critical concepts and definitions necessary for this study. Figures 2 is central to our proposed method for constructing the soil moisture variable.

90  We wanted to show the importance of introducing the metrics based on deviation from normal in figure 2; and the heterogeneity of mean soil moisture across space in figure 4. Figure 5 is illustrating critical results in rejection of the hypothesis that precipitation and soil moisture are the same metric for statistical studies. Below we illustrate the revised figures.

[Figure]

95

**Figure 1. Soil moisture dynamics within a typical growing season. Some soil moisture conditions can be harmful to crops including excess wetness [i], moisture stress intensity[ii], duration of moisture stress [iii], and severity of soil moisture stress [iv]. Normal level of soil moisture is defined as the historical average of volumetric soil moisture within the growing season.**

[Figure]

**Figure 2. Soil texture affects normal moisture levels. The sandy soil has the lowest normal level while the clay has the highest normal levels.**

*Specific comments:*

*Lines 1-2: I would suggest changing the title to something like "Quantifying the compounding effects of soil moisture and heat on crop yield" The paper does not talk about the impacts on irrigation water demand, and none of the figures show results looking at water demands.*

We agree that this section may disrupt the flow of the paper. To improve the flow of the paper and to focus on the main message, we decided to follow the reviewer's suggestion and eliminate the "irrigation demand" section.

*Line 12: Are high-resolution and fine-scale intended to mean different things?*

No. We did not mean different things by using these two separate terms. In the revised version, we only use "fine-scale" throughout the paper to avoid confusion.

*Introduction: The introduction section needs better arrangement for better flow of the ideas. I would recommend focusing on the importance of this work, what is the current state of the knowledge in this space, how this differs or builds on previous efforts, the key novelties it adds to the field, and the specific science/research questions it is trying to tackle. All of this is pretty much there, but it needs to flow better, and certainly results should be omitted from the introduction section to avoid redundancies.*

Many thanks for highlighting the relevance of the work. This section has been shortened and re-written according to these recommendations. We have also removed the results and summary contents from the Introduction section. We have included some of the text from the section "Empirical concerns" to provide adequate background on the models and metrics of individual and compound hydroclimatic extremes for predicting corn yields. We limit the text on the state of the art in the statistical prediction of corn yields to highlight current shortcomings. We also kept the text on the description of the objectives to give a clear view of the originality of the research.

*Lines 22-32: this section is redundant with some of the content of the abstract and talks about the approach and key messages before even articulating the importance of the work. I would recommend omitting.*

We have omitted these lines.

*Lines 57-62: "We show that the coefficient . . ." Avoid throwing results in the middle of the introduction section to avoid redundancy. I would suggest omitting.*

We have omitted these lines.

*Line 79: it is a bit weird to talk about concerns before even talking about the approach.*

This section has largely been moved, with key items moved to the Methods section and some are moved to the Introduction.

*Line 82: spell out Sec for consistency sake.*

According to the Manuscript Preparation Guideline, "the abbreviation 'Sect.' should be used when it appears in running text and should be followed by a number unless it comes at the beginning of a sentence". However, we make them consistent by putting them all at the beginning of the sentences here. We have also removed many of the Section references, as they are not needed.

*Line 83: Do you mean "background: Key factors impacting yield" or something along that line?*

This section has been moved to the Supplementary Materials and other relevant sections. This type of section is standard in the econometric literature from which the methods are mainly derived, but we agree it does not fit in the flow of the paper here.

*Line 85: "before starting our discussion" please rephrase.*

This section has been revised and moved to the Methods section.

155

*Line 94: "we will briefly talk about" please rephrase*

This section has been removed.

*Lines 97-101: I would suggest omitting this paragraph. Water is discussed in section 2.3, and here the*
160     *focus is on spatial aggregation.*

These lines have been omitted, with key details succinctly described in the methods section.

*Line 100: a sample of what? The sentence is somewhat vague.*

This paragraph is now eliminated.

165

*Lines 105-110: "we construct our. . ." this reads like a methodology section and should be part of section 3.*

We removed this part in the revised version. The data construction is explained in the Methds.

170     *Line 111: "another empirical challenge" This reads like you are talking about a different challenge than what was discussed in the above two paragraphs under 2.1. I would suggest separating this section as '2.2 Degree of temporal aggregation' and keeping the previous sub-section on the spatial aspect only.*

Omitted, with key details succinctly described in the methods section.

175     *Line 130: if you are going to end of each challenge with how this study tackles this challenge or differs from previous efforts, then I would suggest that this is done here as well, and at the end of each of the other challenges discussed in section 2.*

Omitted, with key details succinctly described in the methods section.

180     *Line 144: "To undertake. . ." please omit sentence. It does not add much.*

This sentence is omitted in the revised version.

*Lines 145-147: Omit. I would suggest not throwing results at this stage. Plus, the reader does not know anything about WBM yet.*

185     These lines are omitted in the revised version.

*Line 166: "a fixed effect panel regression" I am not sure what that means. Please explain. Also, in the following sentence, what coefficients are you referring to? Please be specific.*

Thanks for raising the need for clarification about this method. We added a brief description of the fixed-effect panel regression. Also, we removed this term in any text before the methods section. The following sentence is added in the description of variables in the Model (1)

190

"The fixed effect variable (also termed the unobserved individual effect) allows us to control for other biophysical or economic characteristics of each location which are not varying over time and can potentially explain the yield differences between counties."

Also, we added the following in the estimation strategy section:

195

"A panel fixed-effect approach is a statistical method for analyzing two-dimensional (e.g. time and location) panel data. This method is helpful for analyzing data that is collected repeatedly for the same locations over time with a relatively short time span (Wooldridge, 2016). As our data set contains information for counties over time, a panel data analysis is appropriate. In addition, a fixed-effect model is appropriate as there are unique biophysical and economic

200

attributes of counties that can explain yield differences across counties and are not changing over time. When we conduct a statistical test (Hausman test), it rejects the random effects model in favor of the fixed effect models we use."

References:

Wooldridge, Jeffrey M. *Introductory Econometrics: A modern approach*. Nelson Education, 2016.

205

*Line 182: rephrase "as we prefer to take care of. . ."*

This sentence is omitted, reworded in the methods.

*Line 189: this section needs a concluding sentence to connect the dots.*

210    The section has been dropped from the revised manuscript.

*Line 217: having read through section 2, it leaves the reader wondering what all of this has to do with compounding extremes. I wonder if section 2 can be shrunk or moved to a later section after describing the method section of the paper to improve the flow of the paper.*

215    Thanks for your suggestion. To improve the flow of the paper, we have shortened the content of this section and moved much of the material to the Supplementary, Methods, or other relevant sections. Here are some of the major changes:

Line 84-92: shortened and moved to the Methods.

Line 93-116: omitted.

220 Line 117-126: shortened and moved to the Introduction.

Line 127-130: moved to the Methods.

Line 131-147: shortened and moved to the Introduction.

Line 148-163: shortened and moved to the Introduction.

Line 164-171: shortened and moved to the Introduction.

225 Line 172-177: shortened and moved to the Methods.

Line 178-189: shortened and moved to the Introduction.

Line 190-217: omitted.

*Line 219: "we introduce two models." What kind of models? Please specify.*

230 Specified now as "statistical models"

*Lines 219-221: why design the two models in this manner? Explain the logic.*

A brief introduction to why these two models are used has been added to the beginning of the Methods section. Here is the related text:

235 "Model 1 assumes the impacts of heat and water on corn yields are separable. This model considers metrics of individual extremes (heat stress and water availability). … Within this framework, we investigate which indicator of individual extremes is a better predictor of corn yields. Relaxing the separability assumption, model 2 assumes the yield impacts of heat and water are mutually interdependent. Model 2 considers indicators of compound extremes."

240

*Line 226-227: "in summary," omit. You just started talking about the model here.*

Omitted.

*Line 229: "as reported by WBM" omit since the reader has not read about WBM yet unless you go with*
245 *my recommendation to have section 3.3 moved to 3.1 as explained later.*

The sentence is omitted.  We have also moved section 3.3 to 3.1 following your suggestion.

*Lines 227-233: these equations (1a-d) need to be shown here. They are core to the whole paper and deserve more attention in the paper.*

250 We have discussed the models in the estimation strategy. In the revised version, the following is added:

"Considering the exposure to each temperature interval to capture the marginal impact of heat and water on crop yields, we estimate the following for model (1-a):

$$y_{it} = \alpha D_{it}^{10-29} + \beta D_{it}^{29} + \delta_a P_{it} + \delta_a' P_{it}^2 + \lambda_s t + \lambda_s' t^2 + c_i + \varepsilon_{it} \tag{1}$$

where $i$ is an index for counties, $t$ is the index of time, $s$ is the index for states, $y_{it}$ is the log corn yields, $D_{it}$ represents growing degree day variables, $P$ shows cumulative precipitation over the growing season, $t$ shows the time trend variable ($t$ = year – 1950), $c_i$ is a time-invariant county fixed effect, $\varepsilon$ is the residual, and $\alpha, \beta, \delta, \lambda$ are the regression parameters showing the marginal impacts. The subscript $a$ is used to show the water coefficients ($\delta$) are related to metrics in Model (1-a).

To evaluate the importance of soil moisture metrics in Model (1-b), we estimate the following:

$$y_{it} = \alpha D_{it}^{10-29} + \beta D_{it}^{29} + \delta_b M_{it} + \delta_b' M_{it}^2 + \lambda_s t + \lambda_s' t^2 + c_i + \varepsilon_{it} \tag{2}$$

where the variables are defined as Model (1-a) except for the water availability metric. Here $M$ shows the seasonal mean soil moisture index calculated as average daily root zone soil moisture from the first day of April to the end of September. The subscript $b$ is used for $\delta$ to distinguish the water coefficients in Model (1-b).

For Model (1-c) we estimate the following model:

$$y_{it} = \alpha D_{it}^{10-29} + \beta D_{it}^{29} + \delta_c N_{it}^{def} + \delta_c' N_{it}^{sur} + \lambda_s t + \lambda_s' t^2 + c_i + \varepsilon_{it} \tag{3}$$

where we replace seasonal mean or cumulative metrics with two new metrics to control the impacts of water extremes on corn yields. Here, $N^{def}$ is the number of days that soil moisture is under 25 mm below normal levels (deficit); and $N^{sur}$ is the number of days that soil moisture is higher than 25 mm above normal levels. The rest of the variables are defined as Model (1-a). The subscript $c$ shows $\delta_c$ is specific to Model (1-c).

Finally, we estimate the following equation for Model (1-d):

$$y_{it} = \alpha D_{it}^{10-29} + \beta D_{it}^{29} + \delta_d M_{it}^{pos} + \delta_d' M_{it}^{neg} + \lambda_s t + \lambda_s' t^2 + c_i + \varepsilon_{it} \tag{4}$$

where $M^{pos}$ is a cumulative measure of positive soil moisture deviations compared to the normal levels (equivalent to A+B+C in Figure 1). And $M^{neg}$ is the cumulative measure of negative soil moisture deviations compared to the normal levels (equivalent to D+E+F in Figure 1). The subscript $d$ distinguished estimated $\delta$ from previous models."

*Lines 225-234: are metrics, indicators, and water variables the name thing here?*

In the revised paper we only use "water metric" when writing specifically about methods used in this paper.

*Line 243: I would suggest making this sub-section (3.3 Data) as the first sub-section in the methods section for better flow. Sub-section 3.4 builds nicely on what's covered under the first two subsections, and the data comes in the middle and breaks the flow.*

We have moved Data to Section 2.1 (first methods section)

*Line 250: "Daily interaction" how is this defined or calculated? Is this a term in equation 2? If so, then please state so.*

295    We added more details on the daily interaction. Here is the text in the estimation strategy discussion:

"For Model (2), we consider the daily interaction of heat and soil moisture as the compound metric. The interaction term is defined when the marginal impact of an explanatory variable depends on the magnitude of yet another explanatory variable (Wooldridge, 2016). Here, the marginal impact of heat on yield depends on water availability; also, the marginal impact of
300    water on yield depends on heat. This is called conditional marginal impact."

References:

Wooldridge, Jeffrey M. *Introductory Econometrics: A modern approach*. Nelson Education, 2016.

*Lines 261-265: omit this paragraph. It is identical to lines 249-255.*

305    Omitted.

*Line 231: Was there any validation work done on the soil moisture data? I am not necessarily asking for that to be shown here, and rather some citations of the previous validation work using WBM should suffice.*

310    Previous work that uses WBM in an agricultural context is provided in the following references (Grogan, 2016; Grogan et al., 2017; Wisser et al., 2008, 2010):

Grogan, D.: Global and regional assessments of unsustainable groundwater use in irrigated agriculture, Doctoral Dissertations [online] Available from: https://scholars.unh.edu/dissertation/2, 2016.

Grogan, D. S., Wisser, D., Prusevich, A., Lammers, R. B. and Frolking, S.: The use and re-use of
315    unsustainable groundwater for irrigation: a global budget, Environ. Res. Lett., 12(3), 034017, doi:10.1088/1748-9326/aa5fb2, 2017.

Wisser, D., Frolking, S., Douglas, E. M., Fekete, B. M., Vörösmarty, C. J. and Schumann, A. H.: Global irrigation water demand: Variability and uncertainties arising from agricultural and climate data sets, Geophysical Research Letters, 35(24), doi:10.1029/2008GL035296, 2008.

320    Wisser, D., Fekete, B. M., Vörösmarty, C. J. and Schumann, A. H.: Reconstructing 20th century global hydrography: a contribution to the Global Terrestrial Network-Hydrology (GTN-H), Hydrology and Earth System Sciences, 14(1), 1–24, 2010.

*Line 304: are you using a single scan, or are you capturing the evolution of the cropland over the*
325    *historical time period?*

The cropland data product, the Crop Data Layer (CDL, USDA NASS, 2017), is an annual time series of cropland area.  This captures the evolution over time. We have revised the sentence to the following:

> "We employed the Crop Data Layer from the US Department of Agriculture  to exclude grid cells with no cropland and to aggregate the grid cell information to the county level (Boryan et al., 2012; USDA-NASS, 2017)."

Boryan, C., Yang, Z. and Di, L.: Deriving 2011 cultivated land cover data sets using usda national agricultural statistics service historic cropland data layers, in 2012 IEEE International Geoscience and Remote Sensing Symposium, pp. 6297–6300, IEEE., 2012.

USDA-NASS: USDA-National Agricultural Statistics Service, Cropland Data Layer, United States Department of Agriculture, National Agricultural Statistics Service, Marketing and Information Services Office, Washington, DC [online] Available from: http//nassgeodata. gmu. edu/Crop-Scape, 2017.

*Line 319: I would suggest moving the materials here to be merged with subsections 3.1 and 3.2. For example, I would suggest moving lines 320 to 331 to appear in line 234. This would mean deleting the sentence "the estimation strategy is described in section 3.4." Similarly, I would move lines 232 through 361 to line 242.*

The methods section has been substantially reorganized, including changing the order of how the data, model equations, and estimation strategy are described. We have also omitted cross-references to sections.

*Line 363: "This section provides estimation results for different representations of Model (1)" well the authors discuss results from Model (2) as well (starting around line 389).*

This sentence has been removed. The results section is re-organized to focus on the main results and to improve the flow of the paper. Here is the new order:

> 3.1. Model (1): predicting yield responses to individual extremes
>
> 3.2 Model (2): predicting yield responses to compound extremes
>
> 3.3 Model comparison
>
> 3.4 Decomposing the variation in US corn yields
>
> 3.5 Robustness checks

*Line 384: so what does all of this mean? Which is the 'best' model formulation, on what basis, and how does this compare with previous findings?*

We have added a section on the model comparison, and section titles have been added for clarity. The performances of the models are compared based on AIC and BIC. While R-squared is not necessarily the best measure for model comparison, we have reported it for interested readers.

"A comparison of model performance metrics is given in Table 5, along with a description of the water metric and the extreme metric used in each model. We find that for Models 1b-d and Models 2a-d the coefficients on the soil moisture metrics are significant and with expected signs. Comparing the models' performance suggests that Model (1-b), with mean soil moisture, performs better than the Model (1-a), with cumulative precipitation. Also, Model (1-d), with the extreme soil moisture metrics, outperforms all previous models (with cumulative precipitation or with mean soil moisture). The best corn yield predictor is from Models (2-a) and (2-b), considering compound extremes through the daily interaction of heat and soil moisture."

*Line 387: "the deficit and by 2300" – delete 'and'*

Deleted.

*Line 397: "The figure shows that Model (1) would. . ." It was not clear that the intent was to compare the two models (1 and 2) to get at the compounding aspect. Some articulation of that upfront would help the reader follow through.*

Many thanks for your comment which helped us focus on the central finding of the paper. As this is a significant point, we have talked about it at the beginning of the Results section. We have added the following:

"Here we describe the regression results from each individual model, and compare their performance to identify which metrics are important to include in the statistical estimate of corn yields. The central finding is that measures of soil moisture extremes are statistically significant, and models including intensity, duration, and severity metrics (as illustrated in Fig. 1) better capture both mean and year-to-year variation in U.S. corn yields.  This point is illustrated in Figure 7, which compares Model 1 (a-d range) to Model 2a: each model estimates the percentage change in corn yields assuming an additional 10 degree-days above 29˚C and no change in mean soil moisture. The figure shows that Model (1) would significantly underestimate the damage for conditions with extreme water surplus or extreme water deficit."

*Line 44: "from previous models" which models are you referring to? 1a,b,c,d, 2a?*

Thanks for pointing to this issue. We have clarified the sentence as:

"This is not significantly different from previous models (1-a, 1-b, 1-c, 1-d, and 2-a)."

*Lines 409-422: I would suggest moving this to be part of the results section. And to change the title for section 5 to be simply "Discussion" and then to jump to 5.1 directly.*

Thanks for your suggestion. This section is shortened and moved to the "Model comparison" subsection of the Results.

*Lines 416-420: please expand on this section to explain what you found out from these additional analyses that are in the appendix. Currently, they come across as throw away sentences.*

400    We have moved these sentences to the "Robustness check" as a subsection of the Results.

"The Supplementary Materials provide several robustness checks. The goal is to investigate whether different assumptions can improve the predictive power of Model (1) such that it outperforms Model (2). We answer three questions. First, are the estimation results from Model (1) different from those using alternative water metrics from WBM output? Second, are the

405    estimates in Model (1) different from those obtained using a model considering growth stages? And third, do the main findings change if we alter the geographical scope of the study?

For the first robustness question, alternative water metrics, we re-estimate Model (1) using daily evapotranspiration (which is related to the water requirements of plants) and soil moisture fraction (soil moisture content divided by field capacity). Overall, the findings remain robust to

410    alternative soil moisture metrics from WBM including the mean of soil moisture fraction, the seasonal mean of evapotranspiration as well as within season standard deviation of them. We also look at the results using an alternative interpolation of WBM data to PRISM resolution (nearest neighbor versus bilinear interpolations). We reject the null hypothesis that the coefficient on yield response to heat is different between these two metrics. Also, we reject the

415    null hypothesis that the prediction power across these models is higher than Model (2).

To test the second robustness question, time separability, we re-estimate Model (1-b) for two-month intervals (Apr-May, Jun-Jul, Aug-Sep), and the findings remain robust. We find that considering bi-monthly variables does not change the yield response to heat. Although this alternative formulation does improve the predictive power of Model (1-b) a little bit, the

420    performance is not better than the original Models (2-a) and (2-b) with compound extremes.

To test the sensitivity of our findings to the geographical area, we re-estimate the models for the Eastern US and the Western US. We find that the estimated coefficients of Models (1-a) and (1-b) are not robust to the geographical choice, while those of Model (2) remain robust."

425    *Lines 420-422: "Finally, we have provided . . ." I would omit these two sentences.*

The sentence is omitted, and the whole section is re-organized.

*Line 432: "we recommend the use of soil . . ." I can't tell if this recommendation is based on the findings in this study, or simply an opinion based on past efforts/studies. Please clarify.*

430    We have shortened the discussion section and focused on the main messages and central findings. This is the revised Discussion:

In this paper, we have identified new water availability metrics that improve the predictive power of statistical corn yield models. While predictive power is an important outcome of this

435  analysis, the insights gained from incrementally adding higher temporal-resolution metrics of water extremes to the models are also valuable for understanding the drivers of corn yield variability, and for revealing the resolution of water availability data required to capture future extremes under climate change scenarios.  Statistical crop models have been used to both elucidate drivers of crop yield trends and variability, and to evaluate potential climate change impacts on crop production in the future (e.g., Lobell and Burke, 2010; Diffenbaugh et al. 2012).

440  However, these models typically use seasonally averaged water availability metrics (e.g., total growing season precipitation), and utilize precipitation more often than soil moisture. Generally, if the location of the study does not expect a significant change in the within-season distribution of the soil moisture, a mean soil moisture index will work. However, if there is an expected change in this distribution, using the mean variable will create biased yield projections. Because

445  climate models project significant changes in the frequency and intensity of both extreme precipitation and temperature (Myhre et al., 2019; Zscheischler et al., 2018; Manning et al., 2019; Bevacqua et al., 2019; Poschlod et al., 2020; Potopová et al., 2020; Wehner, 2019), the results presented here show that the mean metrics of water availability – especially mean precipitation - are not sufficient to capture the impacts on yields. It is necessary to consider the

450  metrics of extreme events as illustrated in Figure 1. As we find that the coefficient on extreme heat is significantly different when considering soil moisture, it is possible that previous climate impact studies have over- or under-estimated the yield impacts. Further, farm management practices can alter soil moisture – and therefore yields – independent of precipitation. Supplemental irrigation, as well as no-till farming, cover cropping, and soil conservation, can

455  increase soil moisture.  These adaptations may occur in places predicted to face higher mean precipitation coupled with more extreme water events. The results of these management practices cannot be captured by statistical models looking at precipitation metrics alone. Such precipitation-based studies could potentially lead to over-estimation of yield damages under future climate extremes by not accounting for human adaptations designed to conserve soil

460  moisture.

*Line 454: "Model (2)" 2a or 2b?*

This part is omitted in the revised version.

465  *Line 454: "while Model (3) predicts ..." do you mean model 1 here?*

This part is omitted in the revised version.

*Lines 484-499: Subsection 5.4 comes as a surprise to the reader. It also reads more like a methods section. I would suggest dropping this subsection.*

470  Thanks for your comment. To improve the flow of the paper, we followed your suggestion. This part is dropped in the revised version.

*Lines 500-509: The first sentence is redundant. The subsection is relatively shallow as compared to previous subsections. Also, it is not clear if there is any conclusion that can be drawn from Figure 10. I wonder if this would fit better if moved to the results section instead of being a discussion subsection.*

The redundant section has been removed, and the remainder has been moved to the results. We have also clarified the methodology.

> "To show the significance of weather variation for crop yields, we estimated the historical impacts of heat and water using Model (2-a). The trend is estimated assuming no variation in heat and water availability. Then, we predicted the impact of heat on yields considering observed variation in heat and assuming normal soil moisture. Finally, we predicted the yield considering observed variation in heat and simulated variation in soil moisture. The residual is not reported."

*Lines 510-536: Subsection 5.6 is another big surprise to the reader. I was not expecting this as this was never baked in the framing of the paper in the initial sections. All the previous sections including the data subsection focused on the US. Though this extends the work globally, which begs the question of how the extrapolation was done. I would suggest omitting this from this paper and keeping it for a follow-on paper.*

Thanks for your suggestion. This section is omitted in the revised section.

*Figure 6: what are the units for the y-axis and for the color bar on the far right?*

Thanks for pointing to the missing units. It is the ratio of soil moisture to normal soil moisture. We have corrected this in the revised version.

---

## Author Comment (AC2) · 23 Sep 2020

**Comments and Responses to Anonymous Referee #2** *(Reviewer comments in italics)*

**Overall:** *The paper is about a well-designed study aiming to elaborate individual and compound extreme event impacts on corn yields in the USA using statistical approach. The significance of extreme events on yield anomalies were studied using various indicators of soil moisture (representing water stress) as well. The outcomes of the paper can be insightful for further studies of predicting crop yield anomalies and assessing impacts of extreme weather conditions to crop yields. Consequently, the paper is worth for publishing with some revisions.*

We would like to thank the referee for his/her helpful comments that helped to improve the manuscript. We have revised the paper accordingly and provided overall and specific answers below. Also, many thanks for the positive feedback on the technical details and the significance of the paper.

As the majority of the comments are around the organization of the paper, we have revised the flow of the paper and transitions within the sections. We have dropped the sections identified less relevant by the referees. This has resulted in a substantial re-ordering of the material presented, and these changes have substantially shortened the paper as requested by the reviewer. Now, the paper is focused on the main messages. The manuscript introduces the problem by stating the research gap as "current statistical models of crop yield prediction ignore the compound extreme". And we establish the discussion around the main finding that "statistical models ignoring compound hydroclimatic extremes will significantly underestimate the yield response to water in hot days while they will significantly overestimate the yield response to water in moderate days". The referee's comments also helped us identify the unclear terms and less critical ideas. They helped us to improve the cohesion of the writings by providing clarifying definitions for unfamiliar terms and by removing the ideas not critical for the argument. The background information has been moved to the Supplementary Materials. We have also clarified the methods, moved some parts of the appendix to the text, and moved some parts of the Methods section to the Supplementary. These are major changes:

Introduction: We have included some of the text from the section "Empirical concerns" to provide adequate background on the models and metrics of individual and compound hydroclimatic extremes for predicting corn yields. We limited the text on the state of the art in the statistical prediction of corn yields to highlight current shortcomings. We kept the text on the description of the objectives to give a clear view of the originality of the research. We have removed the sentences more relevant to the Results and Conclusion.

Empirical concerns: A shortened version of this section has been merged into "Methods" and "Introduction" sections as follows. The sentences regarding the Schlenker and Roberts (2009) model are moved to the Methods section making the base for our model with individual extremes. The sentences regarding spatial aggregation are removed, we only kept our method for spatial aggregation in the Methods section. The sentences regarding average versus extreme metrics of water availability are moved to the introduction as they show the shortcomings in the current literature and how we are going to address them in the paper. The sentences regarding "interaction of soil moisture and heat" are shortened, rephrased, and moved to the introduction as they are base for our arguments about compound extreme. We have also clarified the meaning of the statistical term "interaction" when it first appeared in the manuscript. Finally, the sentences regarding measurement errors and endogeneity concerns are moved to Supplementary.

Methods: This section has some minor changes. We re-order the sub-sections introducing the data before the models. Also, technical terms are described including the "panel fixed effect" method, "daily interaction of heat and soil moisture", and "conditional marginal impact". Figures 1-3 are improved to support definitions and methods.

Results: The results from Model 1 (individual extremes) and Model 2 (compound extremes) have not changed. However, we added a couple of sentences to provide a comparison with previous studies. We added two critical subsections here. A new sub-section on "Model comparison" compares the performance of each model in predicting yields and to illustrate why we have estimated different models with different assumptions and different water metrics. It clearly shows the advantages of using a model with compound extremes. Also, a new sub-section on "Robustness checks" describes why we do these checks and what we learn. Figures 4-6 are moved to the Results section with more details.

Discussion: This section is substantially shortened. We dropped contents about methods and results. The section on "implications for climate studies" and the related text is dropped. The section on "implications for irrigation water demand" and the related text is dropped. Based on our findings we argue that "As we find that the coefficient on extreme heat is significantly different when considering soil moisture, it is possible that previous statistical studies have over- or under-estimated the yield impacts". The revised Discussion section is provided below.

In the following sections, we offer detailed responses to each comment.

*My major comments on the paper are:*

*1- The paper needs to be re-structured/re-written. First, it is too lengthy including textbook information (e.g. Figure 1b, and Figure 2) which are not necessary for the reader (peer knowledge). Second, its structure is chaotic: the introduction chapter includes results and discussions points etc; it is like a short summary of the whole paper; the discussion section includes equations, methods, results and data sources. The authors claim*

*to include results/conclusions which are too diverse and out of scope of the analysis (e.g. irrigation, farm soil management, marginal value, decision making as specified in the abstract). The framework of analysis do not support to make conclusions about these topics. The authors should revise their goals and associated conclusions accordingly. The paper is about compound vs individual extreme events on crop yield and comparison of different soil moisture indicators. Other conclusions not taken from this analysis can be*

*excluded. Furthermore, the empirical concerns are relevant however too lengthy for readers. It can be reduced and can be removed to SI.*

Overall response: Thanks for these excellent suggestions. These comments helped us to improve the organization of the paper. To minimize redundancies and maximize the audience engagement, we re-organized the manuscript. We omitted the less relevant parts in order to focus on the main message. This has resulted in a substantial re-ordering of the material presented, and substantially shortened the paper.

*Comment: "it is too lengthy"*

Response: Regarding the length of the paper, we have shortened the paper substantially from 52 pages (around 19,000 words) to 29 pages (around 10,000 words).

*Comment: "including textbook information (e.g. Figure 1b, and Figure 2) which are not necessary for the reader (peer knowledge)"*

Response: Regarding the textbook information, we have dropped panel b from figure 1. Figure 2 and 3 are revised to illustrate the critical concepts and definitions necessary for this study.

Comment: *"the introduction chapter includes results and discussions points etc; it is like a short summary of the whole paper"*

Response: The flow of the Introduction section has been revised as you will see from the following responses. We have omitted the contents related to conclusion, discussion and summary from the Introduction. The first paragraph and the last paragraph are omitted too.

Comment: *"the discussion section includes equations, methods, results and data sources"*

Response: The Discussion section has been revised substantially as you will see below. We have omitted the equations, methods, and results type of content from it.

Comment: *"The authors claim to include results/conclusions which are too diverse and out of scope of the analysis (e.g. irrigation, farm soil management, marginal value, decision making as specified in the abstract). The framework of analysis do not support to make conclusions about these topics. … Other conclusions not taken from this analysis can be excluded."*

Response: We agree that some of the discussions required further details and their relevance to the main
message were not well-defined. Hence, we have focused on the main message and omitted the discussions about marginal value, farm soil management, supplemental irrigation. Below we have included the shortened and revised Discussion section.

Comment: *"The authors should revise their goals and associated conclusions accordingly. The paper is about
compound vs individual extreme events on crop yield and comparison of different soil moisture indicators."*

Response: Thanks for this very helpful comment. We have revised the flow of the paper focusing on the significance of compound extreme metrics and their advantage over the individual extreme metrics.

Comment: *"Furthermore, the empirical concerns are relevant however too lengthy for readers. It can be
reduced and can be removed to SI".*

Response: Thanks for highlighting the relevance of this material. The content of this section is shortened and moved to SI and other relevant sections. Below, we will describe the changes in more details.

*2- The authors claim that "marginal value of water" will be calculated and utilized in the paper. There is*
*nothing about it in the method and result section (only shown in the discussion section – a short paragraph without any substantial info). I think having this goal of economic analysis is not relevant and beyond the scope the. It is better to exclude this part of the analysis so that the paper is coherent and consistent with its framework.*

It is true that the paper does not provide details on the implications for irrigation water demand. While the
paper could potentially talk about economic and agronomic water demand, it only briefly discussed the economic demand. To improve the flow of the paper and to focus on the main message, we decided to cut the "irrigation demand" section.

*3- Discussion sections were boldly written (e.g. like for climate change discussion and farmer management). I*
*recommend drawing conclusions only if it is supported by the data and analysis.*

Thanks for this comment that helped us focus on the critical findings. We omitted the climate change implications. We have omitted the contents are not critical to our main message. Also, we have revised the conclusion and discussion to only draw the conclusions supported by our analysis. This is the revised Discussion:

"In this paper, we have identified new water availability metrics that improve the predictive power of statistical corn yield models. While predictive power is an important outcome of this analysis, the insights gained from incrementally adding higher temporal-resolution metrics of water extremes to the models are also valuable for understanding the drivers of corn yield variability, and for revealing the resolution of water availability data required to capture future extremes under climate change
scenarios. Statistical crop models have been used to both elucidate drivers of crop yield trends and variability, and to evaluate potential climate change impacts on crop production in the future (Diffenbaugh et al., 2012; Lobell and Burke, 2010) . However, these models typically use seasonally averaged water availability metrics (e.g., total growing season precipitation), and utilize precipitation more often than soil moisture. Generally, if the location of the study does not expect a significant
change in the within-season distribution of the soil moisture, a mean soil moisture index will work. However, if there is an expected change in this distribution, using the mean variable will create biased yield projections. Because climate models project significant changes in the frequency and intensity of both extreme precipitation and temperature (Zscheischler et al., 2018; Manning et al., 2019; Bevacqua et al., 2019; Poschlod et al., 2020; Potopová et al., 2020; Wehner, 2019), the results
presented here show that the mean metrics of water availability – especially mean precipitation - are not sufficient to capture the impacts on yields. It is necessary to consider the metrics of extreme events as illustrated in Figure 1. As we find that the coefficient on extreme heat is significantly different when considering soil moisture, it is possible that previous climate impact studies have over- or under-estimated the yield impacts. Further, farm management practices can alter soil
moisture – and therefore yields – independent of precipitation. Supplemental irrigation, as well as no-till farming, cover cropping, and soil conservation can increase soil moisture. These adaptations may occur in places predicted to face higher mean precipitation coupled with more extreme water events. The results of these management practices cannot be captured by statistical models looking at precipitation metrics alone. Such precipitation-based studies could potentially lead to over-

155   estimation of yield damages under future climate extremes by not accounting for human adaptations designed to conserve soil moisture."

References:

Bevacqua, E., Maraun, D., Vousdoukas, M. I., Voukouvalas, E., Vrac, M., Mentaschi, L. and Widmann, M.: Higher probability of compound flooding from precipitation and storm surge in Europe under anthropogenic
160 climate change, Science Advances, 5(9), eaaw5531, doi:10.1126/sciadv.aaw5531, 2019.

Diffenbaugh, N. S., Hertel, T. W., Scherer, M. and Verma, M.: Response of corn markets to climate volatility under alternative energy futures, Nature Climate Change, 2(7), 514–518, doi:10.1038/nclimate1491, 2012.

Lobell, D. B. and Burke, M. B.: On the use of statistical models to predict crop yield responses to climate change, Agricultural and Forest Meteorology, 150(11), 1443–1452, doi:10.1016/j.agrformet.2010.07.008,
165 2010.

Manning, C., Widmann, M., Bevacqua, E., Loon, A. F. V., Maraun, D. and Vrac, M.: Increased probability of compound long-duration dry and hot events in Europe during summer (1950–2013), Environ. Res. Lett., 14(9), 094006, doi:10.1088/1748-9326/ab23bf, 2019.

Myhre, G., Alterskjær, K., Stjern, C. W., Hodnebrog, Ø., Marelle, L., Samset, B. H., Sillmann, J., Schaller, N.,
170 Fischer, E., Schulz, M. and Stohl, A.: Frequency of extreme precipitation increases extensively with event rareness under global warming, Scientific Reports, 9(1), 1–10, doi:10.1038/s41598-019-52277-4, 2019.

Poschlod, B., Zscheischler, J., Sillmann, J., Wood, R. R. and Ludwig, R.: Climate change effects on hydrometeorological compound events over southern Norway, Weather and Climate Extremes, 28, 100253, doi:10.1016/j.wace.2020.100253, 2020.

175 Potopová, V., Trnka, M., Hamouz, P., Soukup, J. and Castraveț, T.: Statistical modelling of drought-related yield losses using soil moisture-vegetation remote sensing and multiscalar indices in the south-eastern Europe, Agricultural Water Management, 236, 106168, doi:10.1016/j.agwat.2020.106168, 2020.

Wehner, M.: Estimating the probability of multi-variate extreme weather events, in Workshop on Correlated Extremes, Columbia University., 2019.

180 Zscheischler, J., Westra, S., Van Den Hurk, B. J., Seneviratne, S. I., Ward, P. J., Pitman, A., AghaKouchak, A., Bresch, D. N., Leonard, M. and Wahl, T.: Future climate risk from compound events, Nature Climate Change, 8(6), 469–477, 2018.

***For more-detailed comments:***

185 *1) Abstract*

*- which crops were addressed in the article? Please specify. It is important to mention corn here.*

The paper is focused on corn in the US, we have added this in the revised abstract.

*- "the value of water experiences a four-fold increase on hot days": not clear, what do the authors refer to by "value of water"? Is this volume? Value of water is generally associated with significance, importance, true cost etc.*

This sentence is omitted from the abstract. This term was used to refer to economic value, but the related section and discussions are removed from the revised paper.

*- This paper also improves our understanding of the conditional marginal value (or damage)". Which way? And what is conditional marginal value? It is important to provide necessary descriptions in the text as well.*

This sentence is related to a section which is omitted from the revised paper. However, the concept of conditional marginal value has been defined in the paper. This is added in the text:

"Marginal impact and conditional marginal impact are two statistical concepts equivalent to partial derivatives in mathematics. When the partial derivative of one variable does not depend on other variables, we use the term "marginal impact". When it depends on other variables, we use "conditional marginal impact". A conditional marginal impact shows the impact of a compound extreme. A non-conditional marginal impact can show the impact of individual extremes."

*2) Introduction*

*- The first paragraph was written like a conclusion section (after line 26). It includes a short summary, reminding "an abstract". This part needs revision or can be completely excluded (or moved to discussion/conclusion sections).*

This paragraph is excluded in the revised paper.

*- Ln 33: there can be other factors affecting crop yield significantly such as soil, management, nutrients etc.*

This is completely right. The word "variation" was missing. We revised the sentence to the following:

"In agricultural production, water and heat extremes are key determinants of yield variations".

*- Ln 37-38: "Other metrics of extreme water conditions", please specify.*

We revised the sentence as:

"While soil moisture is a more appropriate measure of water availability for crops, extreme water indicators based on soil moisture have been only minimally explored".

*- Ln 38-39: "Current statistical studies had limited success in statistically capturing the yield response to soil moisture metrics", please explain why.*

We added the following explanation:

> "There are several potential reasons for the limited success of previous statistical studies in capturing yield response to soil moisture. Direct measures of soil water availability include complex biophysical and hydrological processes that are difficult to capture in a rather simple statistical model. On the other hand, seasonal mean soil moisture is highly correlated to seasonal precipitation. Thus, including an average of soil water content may not add value to a statistical model."

*- Ln 43: "the impact of climate change on soil moisture". The paper is about individual extreme response of yield vs compound. It is not clear why the authors refer to CC studies.*

This is omitted. The climate change section is dropped now, so this sentence is no longer relevant.

*- Ln 46: "conditional marginal impact". Please explain what this means.*

See explanation above.

*- Ln 50: please explain "wet-heat stress"*

Wet heat stress or moist heat stress are the terms have been used in different disciplines to talk about hot and humid or moist conditions (Buzan and Huber, 2020). Soil water can exacerbate the heat stress under conditions of high humidity. This is not a prevalent condition. However, it can arise in the context of complex meteorological, hydrological, and agronomic interactions. In the US Midwest, a combination of heatwave and corn sweat can create "moist heat stress" which is dangerous for people, animals, and plants.

Reference:

Buzan, J. R. and Huber, M.: Moist heat stress on a hotter Earth, Annual Review of Earth and Planetary Sciences, 48, 2020.

*- Ln 55-60: this part is an outcome of the study. Please remove it to another section (e.g. discussion).*

This part has been shortened and moved to Methods and Results.

*- What are exactly marginal and conditional marginal impacts? It is better if definitions are given for readers.*

See explanation above.

*- Ln 64-79: this part is related to discussion/conclusion. I recommend deleting these parts or move to the other relevant sections.*

In order to shorten the length of the paper, this part has been removed.

*- Ln 77/78: the authors claim that they will show how the results can be used to economically quantify the marginal value of water, in the form of soil moisture, for corn production in the US under different hydroclimatic conditions. I couldn't see this in the rest of the paper. Please clarify.*

This topic has now been omitted as it is tangential to the main theme of this paper.

*3) Empirical concerns*

*- This section is mostly about discussion of the method and assumptions taken for the study. It can be presented as supplementary information, rather than in the main text. That can help reader to focus on the results of the paper and its wider implications. In its current form, it is too lengthy.*

Thanks for your suggestion. To improve the flow of the paper, we have shortened the content of this section and moved them to the Supplementary, Methods, or other relevant sections. Here are some of the major changes:

Line 84-92: shortened and moved to the Methods.

Line 93-116: omitted.

Line 117-126: shortened and moved to the Introduction.

Line 127-130: moved to the Methods.

Line 131-147: shortened and moved to the Introduction.

Line 148-163: shortened and moved to the Introduction.

Line 164-171: shortened and moved to the Introduction.

Line 172-177: shortened and moved to the Methods.

Line 178-189: shortened and moved to the Introduction.

Line 190-217: omitted.

*- Equation 1: please describe what exactly each letter in the equation refers to? For example please refer last*
*variable in the equation as error and describe g(h) function?*

Thanks for catching this. We have added the description for the missing variables. Here, g(h) is a general function showing the yield growth as function of heat.

*- Ln 126: "measure the value of water". Not clear what the authors refer to as "value of water". Please clarify.*

This part is omitted in the revised version.

*- Figure 1: this is nothing new, a known information– like a textbook. Excluding this figure does not change anything about the paper. I recommend not to include it.*

We have dropped panel b of the Figure 1. We believe that Figure 1-a illustrates the concepts that are central to the Methods. While illustration itself might look like textbook information, it helps us to define the metrics of soil moisture extremes. To distinguish this from a common-knowledge figure, we have modified it as follows:

[Figure]

**Figure 1. Soil moisture dynamics within a typical growing season. Some soil moisture conditions can be harmful to crops including excess wetness [i], moisture stress intensity[ii], duration of moisture stress [iii], and severity of soil moisture stress [iv]. Normal level of soil moisture is defined as the historical average of volumetric soil moisture within the growing season.**

*- Ln 134: "Many researchers have acknowledged the need for soil moisture data to predict the response of crop yields to variations in water availability." Please provide references to those researchers.*

This sentence has been omitted in the revised version.

*- Ln 171: please provide references to those studies.*

This sentence has been rephrased and moved to the introduction:

"It has become a standard practice either to focus on a limited geographical area (Rizzo et al., 2018; Wang et al., 2017) or to employ a proxy variable like precipitation, evapotranspiration, or vapor pressure deficit estimates (Comas et al., 2019; Roberts et al., 2013)."

References:

Comas, L. H., Trout, T. J., DeJonge, K. C., Zhang, H. and Gleason, S. M.: Water productivity under strategic
growth stage-based deficit irrigation in maize, Agricultural Water Management, 212, 433–440, doi:10.1016/j.agwat.2018.07.015, 2019.

Rizzo, G., Edreira, J. I. R., Archontoulis, S. V., Yang, H. S. and Grassini, P.: Do shallow water tables contribute to high and stable maize yields in the US Corn Belt?, Global Food Security, 18, 27–34, doi:10.1016/j.gfs.2018.07.002, 2018.

Roberts, M. J., Schlenker, W. and Eyer, J.: Agronomic Weather Measures in Econometric Models of Crop Yield with Implications for Climate Change, Am J Agric Econ, 95(2), 236–243, doi:10.1093/ajae/aas047, 2013.

Wang, R., Bowling, L. C., Cherkauer, K. A., Cibin, R., Her, Y. and Chaubey, I.: Biophysical and hydrological effects of future climate change including trends in CO2, in the St. Joseph River watershed, Eastern Corn Belt, Agricultural Water Management, 180, 280–296, doi:10.1016/j.agwat.2016.09.017, 2017.

*4) Method*

*- This section is too long. Please shorten it and provide detailed information in SI.*

We have substantially revised the organization and transitions within the Methods section. The section is re-organized to focus on the critical parts of the methods and to improve the flow of the paper. Here is the new
order:

     2.1. Data

     2.2 Model (1): individual extremes

     2.3 Model (2): compound extremes

     2.4 Estimation strategy

*- Equation 2: please define each variable and function used in the equation.*

Thanks for pointing to the missing definitions. We have corrected it.

     "where $y_{it}$ is the crop yield, $g(h, m)$ is the yield response function to each combination of soil moisture level, $m$, and temperature (heat), $h$; $\varphi(h, m)$ is the distribution of soil moisture and heat; $\overline{m}$
and $\underline{m}$ are maximum and minimum soil moisture; $\overline{h}$ and $\underline{h}$ are maximum and minimum temperature; and $c_i$ is a time-invariant county fixed effect. Here, we do not separate the impact of heat from water. In other words, the marginal impact of heat depends on water; and the marginal impact of water depends on heat."

*- Ln 230: "some indicators", please clarify which indicators.*

We have clarified this as:

> "In Model (1-c), we consider the number of days that soil moisture is either too high or too low. The model with these metrics of soil moisture extremes further improves the fit, revealing a negative marginal relationship associated with the number of days with low/high soil moisture."

*- Please provide numbers to the equations.*

Thanks for your comment. We added equation numbers in the revised version.

*- Ln 277: g(Ws), please define the parameter*

This description has been added to the text: "and $g(W_s)$ is 1 for all crops, while it is an exponential function of soil moisture depth for non-crop soil areas."

*- Ln 290-295: this is a result of the analysis, not related to data/method or assumptions.*

Thanks for your comment. We have moved this figure to the Supplementary Material. This information is
important to ensuring that soil moisture is a different metric than precipitation. This information is added, and the statement re-contextualized and rephrased.

> "In a statistical study, a natural first step is to look at the correlation between these variables. To show that mean soil moisture is a different metric than mean precipitation, we have plotted the annual mean soil moisture versus annual cumulative precipitation in Fig. S1. This figure is a scatter
> plot for US counties for the growing season from 1981 to 2015. The simple correlation coefficient between them is 0.44. This rejects the hypothesis that soil moisture is highly correlated with precipitation. As mean precipitation has a linear relationship with cumulative precipitation, the results show that mean soil moisture is a different metric than cumulative or mean precipitation."

*- Figure 4,5 and 6 are outcomes of the model/analysis. They can be presented in the result section.*

These figures have moved to the results section. We have also added more explanations about the figures and their messages.

> "The overall simulation results from WBM are illustrated in Fig. 4-6, showing the gridded historical mean for the cultivated continental US, average annual variations for the cultivated continental US,
> and bivariate distribution of soil moisture and heat for the corn growing grid cells. To illustrate the spatial heterogeneity, Fig. 4 shows the growing season mean soil moisture content (in mm in 1000 mm topsoil) as calculated based on daily root-zone soil moisture level from Apr-Sep for 1981-2015 at 2.5 x 2.5 arcmin grids excluding non-cultivated area. Average growing season soil moisture is heterogeneous across the Continental US, with distinct regional patterns (see Fig. 4). For the corn belt, the soil moisture level is relatively high compared to other regions. The mean of volumetric soil moisture ranges from below 50 mm in southern California to above 250 mm in the Corn Belt and around Mississippi.

To compare the variation of simulated soil moisture and precipitation, Fig 4 illustrates the weighted average soil moisture and precipitation over the cultivated US for 1981-2015. In general, variation in
soil moisture average is higher than in that of precipitation (Fig. 5), showing how this new water metric is different from previous approaches. One interesting finding is that for some years the mean precipitation and the mean soil moisture move in opposite directions. For example, in 1990 the mean precipitation declined by around 5% while mean soil moisture increased by around 13%.

To show the dynamics of soil moisture and heat, Fig. 6 shows their bivariate distribution by month
based on daily information for all the cultivated grid cells in the US Corn Belt for 1981-2015. Heat and soil moisture combinations vary through the growing season (Fig. 6) The data shows significant month-to-month variation, with the second half of the season facing hotter and dryer days. Also, July has the highest variation in soil moisture deviation with high probability of compound extremes as the distribution moves toward the lower right. "

*5) Results*

*- Ln 363/364: "We will discuss the implications of these results in Sect. 5." The authors use lots of cross references between the sections as seen in here. This is not necessary, since discussion section means*
*discussion of the results by definition. Please through the entire text and remove unnecessary cross-section references.*

Good point. By cutting the length of the manuscript an improved flow of the paper, there is no need to these references. Thus, the superfluous section cross-references have been removed.

*- Table 2: note section is repetition of the previous sections, thus it is not necessary.*

The table notes have been removed or shortened for all the Tables.

*- Ln 404: "This indicates that water is up to four times more valuable in hot weather." The authors can consider revising the sentence and be more explicit, "value of water" may mean several things.*
As we omitted the value of water section, we have revised this as follows:

"The estimated parameters show the yield response to changes in soil water content. Comparing the parameter values can show the difference in yield response to soil moisture in hot weather and moderate weather…. This indicates that the average yield response to water is up to four times higher in hot weather."

*- Model (2-a) and Model (2-b) were mentioned here for the first time. Please describe the differences between these models in method/data section.*

The Methods section is revised to consider this comment. We have introduced the models in the relevant subsections on the Methods section. Here is the new order:

3.1. Model (1): predicting yield responses to individual extremes

3.2 Model (2): predicting yield responses to compound extremes

3.3 Model comparison

3.4 Decomposing the variation in US corn yields

3.5 Robustness checks

*6) Discussion*

*- Ln 410/411: this is related to differences between model 1 & 2, right? Please clarify which model outcome supports (or all?) the statement.*

These lines are omitted. The clarification has been added in subsection 3.2 "Model (2): predicting yield response to compound extremes".

*- Performance: does this mean best correlation between indicators of extreme events yield anomalies? Please clarify.*

For comparing the models, we have Looked at statistical criteria. We have added Table 5 to compare the performance metrics of the models.

**Table 5: Performance metrics for Models 1(a-d) and 2(a-d).**

| Model | Water metric | Extreme metric | R-squared | AIC (Akaike's information criterion) | BIC (Bayesian information criterion) |
|---|---|---|---|---|---|
| 1-a | Avg. precipitation | Precipitation sqr | 0.469 | -21,238 | -21,201 |
| 1-b | Avg. soil moisture | Soil moisture sqr | 0.471 | -21,612 | -21,576 |
| 1-c | Avg. soil moisture | Number of days with low/high soil moisture | 0.480 | -22,697 | -22,660 |
| 1-d | Avg. soil moisture | Avg soil moisture deficit/surplus | 0.491 | -24,303 | -24,267 |
| 2-a | Avg. soil moisture | T binned by extreme deficit/surplus | 0.492 | -24,402 | -24,328 |
| 2-b | normal soil moisture x T | extreme deficit/surplus x T | 0.501 | -25,582 | -25,509 |

*- First paragraph: what about model 1-c ?*

This section is omitted. We have presented the results from model 1-c in subsection 3.1 " Model (1): predicting yield response to individual extremes".

"Regarding Model (1-c), the coefficient on the number of days with low moisture is also significant and negative. Our estimation sample shows on average 26 days of high soil moisture and 27 days of low soil moisture. The implication is that eliminating 25 days of high soil moisture and 25 days of low soil moisture can improve the corn yields by up to 12.6%."

*- Model 2 a-b were not defined in the previous parts of the paper. Please check consistency.*

The Methods section is revised to consider this issue. As mentioned above, we have introduced the models in the relevant subsections on the Methods section.

     "First, we construct a binning estimator based on daily interaction on heat and soil moisture in model (2-a). …. We estimate a coefficient for each combination of excess heat and soil moisture; i.e., we
estimate a model with metrics of degree days while controlling for soil moisture. The model provides the conditional marginal impact of excess heat as:

$$y_{it} = \alpha D_{it}^{10-29} + \left\{ \sum_m \beta_m D_{mit}^{29} \right\} + \delta M_{it} + \delta' M_{it}^2 + \lambda_s t + \lambda_s' t^2 + c_i + \varepsilon_{it} \qquad \text{(2-a)}$$

where $i$ is the county index, $t$ is the time index, $m$ is an index of soil moisture condition (high, low, normal), $s$ is an index for states, $y$ is average corn yields, $D$ represents conditional growing degree day
variables, $M$ shows the seasonal mean soil moisture content, $T$ stands for the time trend variable, $c_i$ is a time-invariant county fixed effect. Here, $\beta$ is indexed by $m$; i.e., the marginal impact of heat is conditional to soil moisture conditions. $\alpha, \beta, \delta, \lambda$ are the regression parameters showing the marginal impacts.

Second, we estimate a model with metrics of soil moisture while controlling for temperature in model (2-b). We define an index of soil moisture when the temperature is above the threshold and an index of soil moisture when the temperature is below the threshold. In this model, the soil moisture is separated by a temperature threshold $H^*$.

$$y_{it} = \alpha D_{it}^{10-29} + \beta D_{it}^{29} + \left\{ \sum_m \delta_m M_{mit}\big|_{H<H^*} + \delta_m' M_{mit}\big|_{H>H^*} \right\} + \lambda_s t + \lambda_s' t^2 + c_i + \varepsilon_{it} \qquad \text{(2-b)}$$

where $i$ is the county index, $t$ is the time index, $m$ is an index of soil moisture condition, $s$ is an index for states, $y$ shows average corn yields, $D$ represents growing degree day variables, $M$ shows conditional seasonal mean soil moisture, $t$ stands for the time trend variable, $H$ is the average daily temperature, $H^*$ is the temperature threshold, and $c_i$ is a time-invariant county fixed effect. Here, we define $\delta$ and $\delta'$ to test whether the marginal impact of soil moisture depends on heat. The soil moisture metrics are calculated from daily gridded data and aggregated to county and growing season. This includes the index of normal soil moisture (*SM* 0-25+ mm around normal) when $H > H^*$, the index of normal soil moisture when $H < H^*$, the index of moisture deficit (*SM* 25+ mm below normal) when $H > H^*$, index of moisture deficit when $H < H^*$, the index of moisture surplus (*SM* 25+ mm above normal) when $H > H^*$, and the index of moisture surplus when $H < H^*$. *α, β, δ, λ* are the
regression parameters showing the marginal impacts. "

*- Ln 416-421: These are newly introduced topics. None of these research goals (including why to have them), methods and results were mentioned in the previous sections of the paper (e.g. new interaction model, why do you have that and this was never mentioned in the paper). It is like Appendix is another paper with its own*
*results, methods and goals. Please revise the paper accordingly.*

We have substantially shortened and revised the Discussions and Appendix sections. The paper has been revised to focus on the main contribution and major messages. Thus, we dropped Model 2-c and 2-d as well as the discussions on "Implications for irrigation water demand and subsurface drainage" and "Implications for climate studies."

*- Ln 424-428: Is this an outcome supported by the results? If so, please indicate how. It is more like a general knowledge.*

This paragraph is omitted. We have revised the discussion section around the advantages of using the metrics of individual and compound extremes.

*- Ln 429-430: Please provide supporting data/result from the analysis.*

We have removed this paragraph as it requires further investigations which are not related to the main message of the paper.

*- Ln 433: what are the other metrics suggested in the literature?*

This section is omitted in the revised version.

*- Ln 434-438: Is this a conclusion related to compound vs individual extreme weather event analysis? Can we say the same if we use other metrics of water stress than soil moisture?*

Thanks for your helpful question. This section is omitted in the revised version. However, we used your suggestion in revising the paper. We focused on comparing models with individual extremes and models with compound extremes. This has improved the flow of the paper and highlighted the significance of this study.

*- Ln 465-469: I question that the authors' research is critical for climate change studies. First, their analysis was based on historical data and says nothing about counterfactual analysis. This is not the first time impacts of a compound event was researched and like other studies this paper shows stronger impact of a compound event. It does not bring anything to climate change impact studies.*

We have omitted this subsection in the revised version and briefly talked about it in the revised manuscript. However, we believe that the findings are critical for climate impact studies for several reasons. First, the current literature follows methods like Schlenker and Roberts (2009) by modelling yield response functions looking only at average water conditions. They ignore individual and compound extremes related to water. As we find that the coefficient on heat stress variable is significantly different when considering soil moisture and compound extremes, it is possible that previous climate impact studies have over- or under-estimated the yield impacts of climate change. Second, we are introducing simple but operational metrics of individual and compound extremes that can be constructed using hydroclimatic models for the future. These metrics can improve the prediction of crop yields. We are not aware of any other study suggesting such a simple yet powerful prediction framework.

*- Ln 472: please clarify benefit of this collaboration. In which way it helps to solve the challenge.*

We believe that collaboration between hydrologists, climate scientists, and statisticians can improve data generating processes and leads to better models and metrics to help better decisions among people and policymakers. Here is the revised text:

"Applying this framework to climate impact studies will face a key challenge —namely projecting the future compound extremes with the high temporal resolution of Model 2. It requires collaboration between hydrologists, climate scientists, and statisticians (Zscheischler et al., 2020). For future yield projections, we need reliable future projections of daily temperature (maximum and minimum) and soil moisture. Unfortunately, to the best of our knowledge, available data sets including predictions of future soil moisture have a relatively coarse spatial and temporal resolution, and rely on climate model projections with known difficulties representing daily temporal resolution events (Hempel et al., 2013). Further research is required to improve the ability of climate models and impact models in projecting the bivariate distribution of heat-moisture (Sarhadi et al., 2018)."

References:

Zscheischler, J., van den Hurk, B., Ward, P. J. and Westra, S.: Multivariate extremes and compound events, in Climate Extremes and Their Implications for Impact and Risk Assessment, pp. 59–76, Elsevier., 2020.

Sarhadi, A., Ausín, M. C., Wiper, M. P., Touma, D. and Diffenbaugh, N. S.: Multidimensional risk in a nonstationary climate: Joint probability of increasingly severe warm and dry conditions, Science Advances, 4(11), eaau3487, doi:10.1126/sciadv.aau3487, 2018.

*- Ln 479-ln 483: this recommendation is not related to the sub-section heading. The authors stated a discussion point which is out of scope of their analysis and not supported with the overall goal of the paper. Recommendations can be given to farmers etc; however their model/research is not aimed for decision -support guidance. Please remove this section of revise it.*

Thanks for your comment. We have omitted this part.

*- Section 5.4: This section includes literature, method, data and equation related to an estimation. This is not a discussion section. Please previse it accordingly. This additional analysis doesn't bring anything to the value of the paper. I would recommend excluding this analysis from the paper in order to keep its coherence and consistency.*

Thanks for your comments which helped to improve the flow of the paper. We have omitted this subsection.

*- Ln 501: "We find that the average damage from excess heat has been up to four times more severe when combined with water stress" what is the damage, yield losses?*

Thanks for your comment. Originally benefits and damages were considered from an economics point of view. In the revised version, we removed the economic analysis of the value of soil moisture. Now we have revised and clarified the sentence as:

> "Finally, the marginal impact of heat index on crop yields depends on the soil moisture level. We show the average yield damage from heat stress is up to four times more severe when combined with water stress; and therefore the value of water in maintaining crop yield is up to four times larger on hot days."

*- Line 517-525: the CC knowledge and analysis were not included in previous parts (method, data, results) section of the paper. Please include info about this analysis in adequate sections.*

To improve the flow of the paper and reduce the redundancy, the climate change material is omitted.

*- Line 525- 535: There is almost no economic analysis thus the paper does not contribute to CC economics. No policy analysis or research were provided either; also paper does not say/bring anything to regional resilience of agroecosystems, global food security, and as well as future climate impacts. These two paragraphs have to be re-written. These claims are bold and cannot be taken from the research as described in the paper.*

Thanks for your comment. As we have dropped the subsection, theses paragraphs are also omitted.

---

## Author Response (AR1)

**Comments and Responses to the Editor:**

Authors' comment:

5   We would like to thank the Editor for his helpful comments and considerations. We have revised the paper according to the referees' suggestions and provided overall and specific answers separately. Regarding Fig. 2, we agree that it provides standard textbook material. We have revised it to show the importance of introducing the metrics based on deviation from normal. The revised figure is moved to the Supplementary material. The followings are:

10      1- Comments and Responses to Anonymous Referee #1 (pages RC1-1 to RC1-15)
        2- Comments and Responses to Anonymous Referee #2 (pages RC2-1 to RC2-17)
        3- Track-change version of the manuscript (pages 1-66)
*Comment: The paper provides a novel approach to quantify the compounding effects of soil moisture and heat stressors on crop yield in the US over a historical time period. The study investigates multiple statistical representations to try to tease out the importance of the interactions between heat stress and soil moisture conditions on crop yield, and takes advantage of a large scale hydrologic model to extract the necessary soil moisture data to build the various models. The paper looks technically sound, and the paper is a great contribution to the literature.*

We would like to thank the referee for his/her helpful comments that helped to improve the manuscript. We have revised the paper accordingly and provided overall and specific answers below. Also, many thanks for the positive feedback on the technical details and the significance of the paper.

As the majority of the comments are around the organization of the paper, we have revised the flow of the paper and transitions within the sections. We have dropped the sections identified less relevant by the referees. This has resulted in a substantial re-ordering of the material presented, and these changes have substantially shortened the paper as requested by the reviewer. Now, the paper is focused on the main messages. The manuscript introduces the problem by stating the research gap as "current statistical models of crop yield prediction ignore the compound extreme". And we establish the discussion around the main finding that "statistical models ignoring compound hydroclimatic extremes will significantly underestimate the yield response to water in hot days while they will significantly overestimate the yield response to water in moderate days". The referee's comments also helped us identify the unclear terms and less critical ideas. They helped us to improve the cohesion of the writings by providing clarifying definitions for unfamiliar terms and by removing the ideas not critical for the argument. The background information has been moved to the Supplementary Materials. We have also clarified the methods, moved some parts of the appendix to the text, and moved some parts of the Methods section to the Supplementary. These are major changes:

Introduction: We have included some of the text from the section "Empirical concerns" to provide adequate background on the models and metrics of individual and compound hydroclimatic extremes for predicting corn yields. We limited the text on the state of the art in the statistical prediction of corn yields to highlight current shortcomings. We kept the text on the description of the objectives to give a clear view of the originality of the research. We have removed the sentences more relevant to the Results and Conclusion.

Empirical concerns: A shortened version of this section has been merged into "Methods" and "Introduction" sections as follows. The sentences regarding the Schlenker and Roberts (2009) model are moved to the Methods section making the base for our model with individual extremes. The sentences regarding spatial aggregation are removed, we only kept our method for spatial aggregation in the Methods section. The sentences regarding average versus extreme metrics of water availability are moved to the introduction as they show the shortcomings in the current literature and how we are going to address them in the paper. The sentences regarding "interaction of soil moisture and heat" are shortened, rephrased, and moved to the introduction as they are base for our arguments about compound extreme. We have also clarified the meaning of the statistical term "interaction" when it first

appeared in the manuscript. Finally, the sentences regarding measurement errors and endogeneity concerns are moved to Supplementary.

Methods: This section has some minor changes. We re-order the sub-sections introducing the data before the models. Also, technical terms are described including the "panel fixed effect" method, "daily interaction of heat and soil moisture", and "conditional marginal impact". Figures 1-3 are improved to support definitions and methods.

Results: The results from Model 1 (individual extremes) and Model 2 (compound extremes) have not changed. However, we added a couple of sentences to provide a comparison with previous studies. We added two critical subsections here. A new sub-section on "Model comparison" compares the performance of each model in predicting yields and to illustrate why we have estimated different models with different assumptions and different water metrics. It clearly shows the advantages of using a model with compound extremes. Also, a new sub-section on "Robustness checks" describes why we do these checks and what we learn. Figures 4-6 are moved to the Results section with more details.

Discussion: This section is substantially shortened.  We dropped contents about methods and results. The section on "implications for climate studies" and the related text is dropped. The section on "implications for irrigation water demand" and the related text is dropped. Based on our findings we argue that "As we find that the coefficient on extreme heat is significantly different when considering soil moisture, it is possible that previous statistical studies have over- or under-estimated the yield impacts". The revised Discussion section is provided below.

In the following sections, we offer detailed responses to each comment.

*Comment: However, the paper could almost be cut in half to get the key messages of the paper, and much of the text can be either moved to supplementary materials or completely omitted. For example, I would suggest moving the first 5 figures to supplementary materials. I struggled with the flow of the ideas and text, and there is quite a bit of redundancy, and unnecessarily verbose. I would recommend major revisions, with most of the efforts on rearranging and streamlining the flow of the paper.*

Overall response: Thank you for these excellent suggestions. These comments helped us to improve the organization of the paper. To minimize redundancies and maximize audience engagement, we re-organized the manuscript. We omitted the less relevant parts in order to focus on the main message. This has resulted in a substantial re-ordering of the material presented, and substantially shortened the paper.

*Comment: "the paper could almost be cut in half… much of the text can be either moved to supplementary materials or completely omitted"*

Regarding the length of the paper, we have shortened the paper substantially from 52 pages (around 19,000 words) to 28 pages (around 10,000 words). Around 3,000 words are moved to the supplementary.

*Comment: "there is quite a bit of redundancy, and unnecessarily verbose"*

80  Thanks for this comment that helped us to improve the flow of the paper and the cohesion of the writings. We have revised the organization of the paper. The flow of the Introduction section has been revised as you will see from the following responses. We have omitted the contents related to the conclusion, discussion, and summary from the Introduction. The Discussion section has been revised substantially as you will see below. We have omitted the equations, methods, and results type of content from it. In the revised version, we have focused on the main message. We have revised the flow

85  of the paper focusing on the significance of compound extreme metrics and their advantage over the individual extreme metrics.

*Comment: "I would suggest moving the first 5 figures to supplementary materials"*

Regarding the figures, we have dropped panel b from figure 1. We also moved figures 3 to the Supplementary. Figure 2 is also moved to the supplementary with revisions to illustrate the critical

90  concepts and definitions necessary for this study. Figures 4 and 5 are important. We wanted to show the heterogeneity of mean soil moisture across space in figure 4. Figure 5 is illustrating critical results in rejection of the hypothesis that precipitation and soil moisture are the same metric for statistical studies. Below we illustrate the revised figures.

[Figure]

**Figure 1. Soil moisture dynamics within a typical growing season. Some soil moisture conditions can be harmful to crops including excess wetness [i], moisture stress intensity[ii], duration of moisture stress [iii], and severity of soil moisture stress [iv]. Normal level of soil moisture is defined as the historical average of volumetric soil moisture within the growing season.**

[Figure]

100

**Figure 2. Soil texture affects normal moisture levels. The sandy soil has the lowest normal level while the clay has the highest normal levels.**

***Specific comments:***

105 *Lines 1-2: I would suggest changing the title to something like "Quantifying the compounding effects of soil moisture and heat on crop yield" The paper does not talk about the impacts on irrigation water demand, and none of the figures show results looking at water demands.*

We agree that this section may disrupt the flow of the paper. To improve the flow of the paper and to focus on the main message, we decided to follow the reviewer's suggestion and eliminate the "irrigation

110 demand" section.

*Line 12: Are high-resolution and fine-scale intended to mean different things?*

No. We did not mean different things by using these two separate terms. In the revised version, we only use "fine-scale" throughout the paper to avoid confusion.

115

*Introduction: The introduction section needs better arrangement for better flow of the ideas. I would recommend focusing on the importance of this work, what is the current state of the knowledge in this space, how this differs or builds on previous efforts, the key novelties it adds to the field, and the specific science/research questions it is trying to tackle. All of this is pretty much there, but it needs to flow*

120 *better, and certainly results should be omitted from the introduction section to avoid redundancies.*

Many thanks for highlighting the relevance of the work. This section has been shortened and re-written according to these recommendations. We have also removed the results and summary contents from the Introduction section. We have included some of the text from the section "Empirical concerns" to provide adequate background on the models and metrics of individual and compound hydroclimatic extremes for predicting corn yields. We limit the text on the state of the art in the statistical prediction of corn yields to highlight current shortcomings. We also kept the text on the description of the objectives to give a clear view of the originality of the research.

*Lines 22-32: this section is redundant with some of the content of the abstract and talks about the approach and key messages before even articulating the importance of the work. I would recommend omitting.*

We have omitted these lines.

*Lines 57-62: "We show that the coefficient . . ." Avoid throwing results in the middle of the introduction section to avoid redundancy. I would suggest omitting.*

We have omitted these lines.

*Line 79: it is a bit weird to talk about concerns before even talking about the approach.*

This section has largely been moved, with key items moved to the Methods section and some are moved to the Introduction.

*Line 82: spell out Sec for consistency sake.*

According to the Manuscript Preparation Guideline, "the abbreviation 'Sect.' should be used when it appears in running text and should be followed by a number unless it comes at the beginning of a sentence". However, we make them consistent by putting them all at the beginning of the sentences here. We have also removed many of the Section references, as they are not needed.

*Line 83: Do you mean "background: Key factors impacting yield" or something along that line?*

This section has been moved to the Supplementary Materials and other relevant sections. This type of section is standard in the econometric literature from which the methods are mainly derived, but we agree it does not fit in the flow of the paper here.

*Line 85: "before starting our discussion" please rephrase.*

This section has been revised and moved to the Methods section.

155

*Line 94: "we will briefly talk about" please rephrase*

This section has been removed.

*Lines 97-101: I would suggest omitting this paragraph. Water is discussed in section 2.3, and here the*
160   *focus is on spatial aggregation.*

These lines have been omitted, with key details succinctly described in the methods section.

*Line 100: a sample of what? The sentence is somewhat vague.*

This paragraph is now eliminated.

165

*Lines 105-110: "we construct our. . ." this reads like a methodology section and should be part of section 3.*

We removed this part in the revised version. The data construction is explained in the Methds.

170   *Line 111: "another empirical challenge" This reads like you are talking about a different challenge than what was discussed in the above two paragraphs under 2.1. I would suggest separating this section as '2.2 Degree of temporal aggregation' and keeping the previous sub-section on the spatial aspect only.*

Omitted, with key details succinctly described in the methods section.

175   *Line 130: if you are going to end of each challenge with how this study tackles this challenge or differs from previous efforts, then I would suggest that this is done here as well, and at the end of each of the other challenges discussed in section 2.*

Omitted, with key details succinctly described in the methods section.

180   *Line 144: "To undertake. . ." please omit sentence. It does not add much.*

This sentence is omitted in the revised version.

*Lines 145-147: Omit. I would suggest not throwing results at this stage. Plus, the reader does not know anything about WBM yet.*

185   These lines are omitted in the revised version.

*Line 166: "a fixed effect panel regression" I am not sure what that means. Please explain. Also, in the following sentence, what coefficients are you referring to? Please be specific.*

Thanks for raising the need for clarification about this method. We added a brief description of the fixed-effect panel regression. Also, we removed this term in any text before the methods section. The
190    following sentence is added in the description of variables in the Model (1)

"The fixed effect variable (also termed the unobserved individual effect) allows us to control for other biophysical or economic characteristics of each location which are not varying over time and can potentially explain the yield differences between counties."

Also, we added the following in the estimation strategy section:

195    "A panel fixed-effect approach is a statistical method for analyzing two-dimensional (e.g. time and location) panel data. This method is helpful for analyzing data that is collected repeatedly for the same locations over time with a relatively short time span (Wooldridge, 2016). As our data set contains information for counties over time, a panel data analysis is appropriate. In addition, a fixed-effect model is appropriate as there are unique biophysical and economic
200    attributes of counties that can explain yield differences across counties and are not changing over time. When we conduct a statistical test (Hausman test), it rejects the random effects model in favor of the fixed effect models we use."

References:

Wooldridge, Jeffrey M. *Introductory Econometrics: A modern approach*. Nelson Education, 2016.

205

*Line 182: rephrase "as we prefer to take care of. . ."*

This sentence is omitted, reworded in the methods.

*Line 189: this section needs a concluding sentence to connect the dots.*

210    The section has been dropped from the revised manuscript.

*Line 217: having read through section 2, it leaves the reader wondering what all of this has to do with compounding extremes. I wonder if section 2 can be shrunk or moved to a later section after describing the method section of the paper to improve the flow of the paper.*

215    Thanks for your suggestion. To improve the flow of the paper, we have shortened the content of this section and moved much of the material to the Supplementary, Methods, or other relevant sections. Here are some of the major changes:

Line 84-92: shortened and moved to the Methods.

Line 93-116: omitted.

220     Line 117-126: shortened and moved to the Introduction.

Line 127-130: moved to the Methods.

Line 131-147: shortened and moved to the Introduction.

Line 148-163: shortened and moved to the Introduction.

Line 164-171: shortened and moved to the Introduction.

225     Line 172-177: shortened and moved to the Methods.

Line 178-189: shortened and moved to the Introduction.

Line 190-217: omitted.

*Line 219: "we introduce two models." What kind of models? Please specify.*

230     Specified now as "statistical models"

*Lines 219-221: why design the two models in this manner? Explain the logic.*

A brief introduction to why these two models are used has been added to the beginning of the Methods section. Here is the related text:

235     "Model 1 assumes the impacts of heat and water on corn yields are separable. This model considers metrics of individual extremes (heat stress and water availability). … Within this framework, we investigate which indicator of individual extremes is a better predictor of corn yields. Relaxing the separability assumption, model 2 assumes the yield impacts of heat and water are mutually interdependent. Model 2 considers indicators of compound extremes."

240

*Line 226-227: "in summary," omit. You just started talking about the model here.*

Omitted.

*Line 229: "as reported by WBM" omit since the reader has not read about WBM yet unless you go with*
245     *my recommendation to have section 3.3 moved to 3.1 as explained later.*

The sentence is omitted.  We have also moved section 3.3 to 3.1 following your suggestion.

*Lines 227-233: these equations (1a-d) need to be shown here. They are core to the whole paper and deserve more attention in the paper.*

250     We have discussed the models in the estimation strategy. In the revised version, the following is added:

"Considering the exposure to each temperature interval to capture the marginal impact of heat and water on crop yields, we estimate the following for model (1-a):

$$y_{it} = \alpha D_{it}^{10-29} + \beta D_{it}^{29} + \delta_a P_{it} + \delta'_a P_{it}^2 + \lambda_s t + \lambda'_s t^2 + c_i + \varepsilon_{it} \tag{1}$$

where $i$ is an index for counties, $t$ is the index of time, $s$ is the index for states, $y_{it}$ is the log corn yields, $D_{it}$ represents growing degree day variables, $P$ shows cumulative precipitation over the growing season, $t$ shows the time trend variable ($t$ = year – 1950), $c_i$ is a time-invariant county fixed effect, $\varepsilon$ is the residual, and $\alpha, \beta, \delta, \lambda$ are the regression parameters showing the marginal impacts. The subscript $a$ is used to show the water coefficients ($\delta$) are related to metrics in Model (1-a).

To evaluate the importance of soil moisture metrics in Model (1-b), we estimate the following:

$$y_{it} = \alpha D_{it}^{10-29} + \beta D_{it}^{29} + \delta_b M_{it} + \delta'_b M_{it}^2 + \lambda_s t + \lambda'_s t^2 + c_i + \varepsilon_{it} \tag{2}$$

where the variables are defined as Model (1-a) except for the water availability metric. Here $M$ shows the seasonal mean soil moisture index calculated as average daily root zone soil moisture from the first day of April to the end of September. The subscript $b$ is used for $\delta$ to distinguish the water coefficients in Model (1-b).

For Model (1-c) we estimate the following model:

$$y_{it} = \alpha D_{it}^{10-29} + \beta D_{it}^{29} + \delta_c N_{it}^{def} + \delta'_c N_{it}^{sur} + \lambda_s t + \lambda'_s t^2 + c_i + \varepsilon_{it} \tag{3}$$

where we replace seasonal mean or cumulative metrics with two new metrics to control the impacts of water extremes on corn yields. Here, $N^{def}$ is the number of days that soil moisture is under 25 mm below normal levels (deficit); and $N^{sur}$ is the number of days that soil moisture is higher than 25 mm above normal levels. The rest of the variables are defined as Model (1-a). The subscript $c$ shows $\delta_c$ is specific to Model (1-c).

Finally, we estimate the following equation for Model (1-d):

$$y_{it} = \alpha D_{it}^{10-29} + \beta D_{it}^{29} + \delta_d M_{it}^{pos} + \delta'_d M_{it}^{neg} + \lambda_s t + \lambda'_s t^2 + c_i + \varepsilon_{it} \tag{4}$$

where $M^{pos}$ is a cumulative measure of positive soil moisture deviations compared to the normal levels (equivalent to A+B+C in Figure 1). And $M^{neg}$ is the cumulative measure of negative soil moisture deviations compared to the normal levels (equivalent to D+E+F in Figure 1). The subscript $d$ distinguished estimated $\delta$ from previous models."

*Lines 225-234: are metrics, indicators, and water variables the name thing here?*

In the revised paper we only use "water metric" when writing specifically about methods used in this paper.

*Line 243: I would suggest making this sub-section (3.3 Data) as the first sub-section in the methods section for better flow. Sub-section 3.4 builds nicely on what's covered under the first two subsections, and the data comes in the middle and breaks the flow.*

We have moved Data to Section 2.1 (first methods section)

*Line 250: "Daily interaction" how is this defined or calculated? Is this a term in equation 2? If so, then please state so.*

295     We added more details on the daily interaction. Here is the text in the estimation strategy discussion:

> "For Model (2), we consider the daily interaction of heat and soil moisture as the compound metric. The interaction term is defined when the marginal impact of an explanatory variable depends on the magnitude of yet another explanatory variable (Wooldridge, 2016). Here, the marginal impact of heat on yield depends on water availability; also, the marginal impact of
300     water on yield depends on heat. This is called conditional marginal impact."

The methods section has been substantially reorganized, including changing the order of how the data, model equations, and estimation strategy are described. We have also omitted cross-references to sections.

345

*Line 363: "This section provides estimation results for different representations of Model (1)" well the authors discuss results from Model (2) as well (starting around line 389).*

This sentence has been removed. The results section is re-organized to focus on the main results and to improve the flow of the paper. Here is the new order:

350
       3.1. Model (1): predicting yield responses to individual extremes

       3.2 Model (2): predicting yield responses to compound extremes

       3.3 Model comparison

       3.4 Decomposing the variation in US corn yields

       3.5 Robustness checks

355

*Line 384: so what does all of this mean? Which is the 'best' model formulation, on what basis, and how does this compare with previous findings?*

We have added a section on the model comparison, and section titles have been added for clarity. The performances of the models are compared based on AIC and BIC. While R-squared is not necessarily the
360 best measure for model comparison, we have reported it for interested readers.

"A comparison of model performance metrics is given in Table 5, along with a description of the water metric and the extreme metric used in each model. We find that for Models 1b-d and Models 2a-d the coefficients on the soil moisture metrics are significant and with expected signs. Comparing the models' performance suggests that Model (1-b), with mean soil moisture, performs better than the Model (1-a), with cumulative precipitation. Also, Model (1-d), with the extreme soil moisture metrics, outperforms all previous models (with cumulative precipitation or with mean soil moisture). The best corn yield predictor is from Models (2-a) and (2-b), considering compound extremes through the daily interaction of heat and soil moisture."

*Line 387: "the deficit and by 2300" – delete 'and'*

Deleted.

*Line 397: "The figure shows that Model (1) would. . ." It was not clear that the intent was to compare the two models (1 and 2) to get at the compounding aspect. Some articulation of that upfront would help the reader follow through.*

Many thanks for your comment which helped us focus on the central finding of the paper. As this is a significant point, we have talked about it at the beginning of the Results section. We have added the following:

"Here we describe the regression results from each individual model, and compare their performance to identify which metrics are important to include in the statistical estimate of corn yields. The central finding is that measures of soil moisture extremes are statistically significant, and models including intensity, duration, and severity metrics (as illustrated in Fig. 1) better capture both mean and year-to-year variation in U.S. corn yields. This point is illustrated in Figure 7, which compares Model 1 (a-d range) to Model 2a: each model estimates the percentage change in corn yields assuming an additional 10 degree-days above 29˚C and no change in mean soil moisture. The figure shows that Model (1) would significantly underestimate the damage for conditions with extreme water surplus or extreme water deficit."

*Line 44: "from previous models" which models are you referring to? 1a,b,c,d, 2a?*

Thanks for pointing to this issue. We have clarified the sentence as:

"This is not significantly different from previous models (1-a, 1-b, 1-c, 1-d, and 2-a)."

*Lines 409-422: I would suggest moving this to be part of the results section. And to change the title for section 5 to be simply "Discussion" and then to jump to 5.1 directly.*

Thanks for your suggestion. This section is shortened and moved to the "Model comparison" subsection of the Results.

*Lines 416-420: please expand on this section to explain what you found out from these additional analyses that are in the appendix. Currently, they come across as throw away sentences.*

400    We have moved these sentences to the "Robustness check" as a subsection of the Results.

"The Supplementary Materials provide several robustness checks. The goal is to investigate whether different assumptions can improve the predictive power of Model (1) such that it outperforms Model (2). We answer three questions. First, are the estimation results from Model (1) different from those using alternative water metrics from WBM output? Second, are the

405    estimates in Model (1) different from those obtained using a model considering growth stages? And third, do the main findings change if we alter the geographical scope of the study?

For the first robustness question, alternative water metrics, we re-estimate Model (1) using daily evapotranspiration (which is related to the water requirements of plants) and soil moisture fraction (soil moisture content divided by field capacity). Overall, the findings remain robust to

410    alternative soil moisture metrics from WBM including the mean of soil moisture fraction, the seasonal mean of evapotranspiration as well as within season standard deviation of them. We also look at the results using an alternative interpolation of WBM data to PRISM resolution (nearest neighbor versus bilinear interpolations). We reject the null hypothesis that the coefficient on yield response to heat is different between these two metrics. Also, we reject the

415    null hypothesis that the prediction power across these models is higher than Model (2).

To test the second robustness question, time separability, we re-estimate Model (1-b) for two-month intervals (Apr-May, Jun-Jul, Aug-Sep), and the findings remain robust. We find that considering bi-monthly variables does not change the yield response to heat. Although this alternative formulation does improve the predictive power of Model (1-b) a little bit, the

420    performance is not better than the original Models (2-a) and (2-b) with compound extremes.

To test the sensitivity of our findings to the geographical area, we re-estimate the models for the Eastern US and the Western US. We find that the estimated coefficients of Models (1-a) and (1-b) are not robust to the geographical choice, while those of Model (2) remain robust."

425    *Lines 420-422: "Finally, we have provided . . ." I would omit these two sentences.*

The sentence is omitted, and the whole section is re-organized.

*Line 432: "we recommend the use of soil . . ." I can't tell if this recommendation is based on the findings in this study, or simply an opinion based on past efforts/studies. Please clarify.*

430    We have shortened the discussion section and focused on the main messages and central findings. This is the revised Discussion:

In this paper, we have identified new water availability metrics that improve the predictive power of statistical corn yield models. While predictive power is an important outcome of this

435 analysis, the insights gained from incrementally adding higher temporal-resolution metrics of water extremes to the models are also valuable for understanding the drivers of corn yield variability, and for revealing the resolution of water availability data required to capture future extremes under climate change scenarios.  Statistical crop models have been used to both elucidate drivers of crop yield trends and variability, and to evaluate potential climate change impacts on crop production in the future (e.g., Lobell and Burke, 2010; Diffenbaugh et al. 2012).

440 However, these models typically use seasonally averaged water availability metrics (e.g., total growing season precipitation), and utilize precipitation more often than soil moisture. Generally, if the location of the study does not expect a significant change in the within-season distribution of the soil moisture, a mean soil moisture index will work. However, if there is an expected change in this distribution, using the mean variable will create biased yield projections. Because

445 climate models project significant changes in the frequency and intensity of both extreme precipitation and temperature (Myhre et al., 2019; Zscheischler et al., 2018; Manning et al., 2019; Bevacqua et al., 2019; Poschlod et al., 2020; Potopová et al., 2020; Wehner, 2019), the results presented here show that the mean metrics of water availability – especially mean precipitation - are not sufficient to capture the impacts on yields. It is necessary to consider the

450 metrics of extreme events as illustrated in Figure 1. As we find that the coefficient on extreme heat is significantly different when considering soil moisture, it is possible that previous climate impact studies have over- or under-estimated the yield impacts. Further, farm management practices can alter soil moisture – and therefore yields – independent of precipitation. Supplemental irrigation, as well as no-till farming, cover cropping, and soil conservation, can

455 increase soil moisture.  These adaptations may occur in places predicted to face higher mean precipitation coupled with more extreme water events. The results of these management practices cannot be captured by statistical models looking at precipitation metrics alone. Such precipitation-based studies could potentially lead to over-estimation of yield damages under future climate extremes by not accounting for human adaptations designed to conserve soil

460 moisture.

*Line 454: "Model (2)" 2a or 2b?*

This part is omitted in the revised version.

465 *Line 454: "while Model (3) predicts ..." do you mean model 1 here?*

This part is omitted in the revised version.

*Lines 484-499: Subsection 5.4 comes as a surprise to the reader. It also reads more like a methods section. I would suggest dropping this subsection.*

470 Thanks for your comment. To improve the flow of the paper, we followed your suggestion. This part is dropped in the revised version.

*Lines 500-509: The first sentence is redundant. The subsection is relatively shallow as compared to previous subsections. Also, it is not clear if there is any conclusion that can be drawn from Figure 10. I wonder if this would fit better if moved to the results section instead of being a discussion subsection.*

The redundant section has been removed, and the remainder has been moved to the results. We have also clarified the methodology.

"To show the significance of weather variation for crop yields, we estimated the historical impacts of heat and water using Model (2-a). The trend is estimated assuming no variation in heat and water availability. Then, we predicted the impact of heat on yields considering observed variation in heat and assuming normal soil moisture. Finally, we predicted the yield considering observed variation in heat and simulated variation in soil moisture. The residual is not reported."

*Lines 510-536: Subsection 5.6 is another big surprise to the reader. I was not expecting this as this was never baked in the framing of the paper in the initial sections. All the previous sections including the data subsection focused on the US. Though this extends the work globally, which begs the question of how the extrapolation was done. I would suggest omitting this from this paper and keeping it for a follow-on paper.*

Thanks for your suggestion. This section is omitted in the revised section.

*Figure 6: what are the units for the y-axis and for the color bar on the far right?*

Thanks for pointing to the missing units. It is the ratio of soil moisture to normal soil moisture. We have corrected this in the revised version.
**Overall:** *The paper is about a well-designed study aiming to elaborate individual and compound extreme event impacts on corn yields in the USA using statistical approach. The significance of extreme events on yield anomalies were studied using various indicators of soil moisture (representing water stress) as well. The outcomes of the paper can be insightful for further studies of predicting crop yield anomalies and assessing impacts of extreme weather conditions to crop yields. Consequently, the paper is worth for publishing with some revisions.*

We would like to thank the referee for his/her helpful comments that helped to improve the manuscript. We have revised the paper accordingly and provided overall and specific answers below. Also, many thanks for the positive feedback on the technical details and the significance of the paper.

As the majority of the comments are around the organization of the paper, we have revised the flow of the paper and transitions within the sections. We have dropped the sections identified less relevant by the referees. This has resulted in a substantial re-ordering of the material presented, and these changes have substantially shortened the paper as requested by the reviewer. Now, the paper is focused on the main messages. The manuscript introduces the problem by stating the research gap as "current statistical models of crop yield prediction ignore the compound extreme". And we establish the discussion around the main finding that "statistical models ignoring compound hydroclimatic extremes will significantly underestimate the yield response to water in hot days while they will significantly overestimate the yield response to water in moderate days". The referee's comments also helped us identify the unclear terms and less critical ideas. They helped us to improve the cohesion of the writings by providing clarifying definitions for unfamiliar terms and by removing the ideas not critical for the argument. The background information has been moved to the Supplementary Materials. We have also clarified the methods, moved some parts of the appendix to the text, and moved some parts of the Methods section to the Supplementary. These are major changes:

Introduction: We have included some of the text from the section "Empirical concerns" to provide adequate background on the models and metrics of individual and compound hydroclimatic extremes for predicting corn yields. We limited the text on the state of the art in the statistical prediction of corn yields to highlight current shortcomings. We kept the text on the description of the objectives to give a clear view of the originality of the research. We have removed the sentences more relevant to the Results and Conclusion.

Empirical concerns: A shortened version of this section has been merged into "Methods" and "Introduction" sections as follows. The sentences regarding the Schlenker and Roberts (2009) model are moved to the Methods section making the base for our model with individual extremes. The sentences regarding spatial aggregation are removed, we only kept our method for spatial aggregation in the Methods section. The sentences regarding average versus extreme metrics of water availability are moved to the introduction as they show the shortcomings in the current literature and how we are going to address them in the paper. The sentences regarding "interaction of soil moisture and heat" are shortened, rephrased, and moved to the introduction as they are base for our arguments about compound extreme. We have also clarified the meaning of the statistical term "interaction" when it first appeared in the manuscript. Finally, the sentences regarding measurement errors and endogeneity concerns are moved to Supplementary.

Methods: This section has some minor changes. We re-order the sub-sections introducing the data before the models. Also, technical terms are described including the "panel fixed effect" method, "daily interaction of heat and soil moisture", and "conditional marginal impact". Figures 1-3 are improved to support definitions and methods.

45    Results: The results from Model 1 (individual extremes) and Model 2 (compound extremes) have not changed. However, we added a couple of sentences to provide a comparison with previous studies. We added two critical subsections here. A new sub-section on "Model comparison" compares the performance of each model in predicting yields and to illustrate why we have estimated different models with different assumptions and different water metrics. It clearly shows the advantages of using a model with compound

50    extremes. Also, a new sub-section on "Robustness checks" describes why we do these checks and what we learn. Figures 4-6 are moved to the Results section with more details.

Discussion: This section is substantially shortened. We dropped contents about methods and results. The section on "implications for climate studies" and the related text is dropped. The section on "implications for irrigation water demand" and the related text is dropped. Based on our findings we argue that "As we find

55    that the coefficient on extreme heat is significantly different when considering soil moisture, it is possible that previous statistical studies have over- or under-estimated the yield impacts". The revised Discussion section is provided below.

In the following sections, we offer detailed responses to each comment.

60    *My major comments on the paper are:*

*1- The paper needs to be re-structured/re-written. First, it is too lengthy including textbook information (e.g. Figure 1b, and Figure 2) which are not necessary for the reader (peer knowledge). Second, its structure is chaotic: the introduction chapter includes results and discussions points etc; it is like a short summary of the whole paper; the discussion section includes equations, methods, results and data sources. The authors claim*

65    *to include results/conclusions which are too diverse and out of scope of the analysis (e.g. irrigation, farm soil management, marginal value, decision making as specified in the abstract). The framework of analysis do not support to make conclusions about these topics. The authors should revise their goals and associated conclusions accordingly. The paper is about compound vs individual extreme events on crop yield and comparison of different soil moisture indicators. Other conclusions not taken from this analysis can be*

70    *excluded. Furthermore, the empirical concerns are relevant however too lengthy for readers. It can be reduced and can be removed to SI.*

Overall response: Thanks for these excellent suggestions. These comments helped us to improve the organization of the paper. To minimize redundancies and maximize the audience engagement, we re-organized the manuscript. We omitted the less relevant parts in order to focus on the main message. This has

75    resulted in a substantial re-ordering of the material presented, and substantially shortened the paper.

*Comment: "it is too lengthy"*

Response: Regarding the length of the paper, we have shortened the paper substantially from 52 pages (around 19,000 words) to 29 pages (around 10,000 words).

80   *Comment: "including textbook information (e.g. Figure 1b, and Figure 2) which are not necessary for the reader (peer knowledge)"*

Response: Regarding the textbook information, we have dropped panel b from figure 1. Figure 2 and 3 are revised to illustrate the critical concepts and definitions necessary for this study. We have moved figures 2 and 3 to the supplementary.

Comment: *"the introduction chapter includes results and discussions points etc; it is like a short summary of the whole paper"*

Response: The flow of the Introduction section has been revised as you will see from the following responses. We have omitted the contents related to conclusion, discussion and summary from the Introduction. The first
90   paragraph and the last paragraph are omitted too.

Comment: *"the discussion section includes equations, methods, results and data sources"*

Response: The Discussion section has been revised substantially as you will see below. We have omitted the equations, methods, and results type of content from it.

Comment: *"The authors claim to include results/conclusions which are too diverse and out of scope of the analysis (e.g. irrigation, farm soil management, marginal value, decision making as specified in the abstract). The framework of analysis do not support to make conclusions about these topics. … Other conclusions not taken from this analysis can be excluded."*

100   Response: We agree that some of the discussions required further details and their relevance to the main message were not well-defined. Hence, we have focused on the main message and omitted the discussions about marginal value, farm soil management, supplemental irrigation. Below we have included the shortened and revised Discussion section.

105   Comment: *"The authors should revise their goals and associated conclusions accordingly. The paper is about compound vs individual extreme events on crop yield and comparison of different soil moisture indicators."*

Response: Thanks for this very helpful comment. We have revised the flow of the paper focusing on the significance of compound extreme metrics and their advantage over the individual extreme metrics.

110   Comment: *"Furthermore, the empirical concerns are relevant however too lengthy for readers. It can be reduced and can be removed to SI".*

Response: Thanks for highlighting the relevance of this material. The content of this section is shortened and moved to SI and other relevant sections. Below, we will describe the changes in more details.

115    *2- The authors claim that "marginal value of water" will be calculated and utilized in the paper. There is nothing about it in the method and result section (only shown in the discussion section – a short paragraph without any substantial info). I think having this goal of economic analysis is not relevant and beyond the scope the. It is better to exclude this part of the analysis so that the paper is coherent and consistent with its framework.*

120    It is true that the paper does not provide details on the implications for irrigation water demand. While the paper could potentially talk about economic and agronomic water demand, it only briefly discussed the economic demand. To improve the flow of the paper and to focus on the main message, we decided to cut the "irrigation demand" section.

125    *3- Discussion sections were boldly written (e.g. like for climate change discussion and farmer management). I recommend drawing conclusions only if it is supported by the data and analysis.*

Thanks for this comment that helped us focus on the critical findings. We omitted the climate change implications. We have omitted the contents are not critical to our main message. Also, we have revised the conclusion and discussion to only draw the conclusions supported by our analysis. This is the revised

130    Discussion:

"In this paper, we have identified new water availability metrics that improve the predictive power of statistical corn yield models. While predictive power is an important outcome of this analysis, the insights gained from incrementally adding higher temporal-resolution metrics of water extremes to the models are also valuable for understanding the drivers of corn yield variability, and for revealing

[revised manuscript text omitted]

185 ***For more-detailed comments:***

*1) Abstract*

*- which crops were addressed in the article? Please specify. It is important to mention corn here.*

The paper is focused on corn in the US, we have added this in the revised abstract.

190     *- "the value of water experiences a four-fold increase on hot days": not clear, what do the authors refer to by "value of water"? Is this volume? Value of water is generally associated with significance, importance, true cost etc.*

This sentence is omitted from the abstract. This term was used to refer to economic value, but the related section and discussions are removed from the revised paper.

195

*- This paper also improves our understanding of the conditional marginal value (or damage)". Which way? And what is conditional marginal value? It is important to provide necessary descriptions in the text as well.*

This sentence is related to a section which is omitted from the revised paper. However, the concept of conditional marginal value has been defined in the paper. This is added in the text:

200     "Marginal impact and conditional marginal impact are two statistical concepts equivalent to partial derivatives in mathematics. When the partial derivative of one variable does not depend on other variables, we use the term "marginal impact". When it depends on other variables, we use "conditional marginal impact". A conditional marginal impact shows the impact of a compound extreme. A non-conditional marginal impact can show the impact of individual extremes."

205

*2) Introduction*

*- The first paragraph was written like a conclusion section (after line 26). It includes a short summary, reminding "an abstract". This part needs revision or can be completely excluded (or moved to discussion/conclusion sections).*

210     This paragraph is excluded in the revised paper.

*- Ln 33: there can be other factors affecting crop yield significantly such as soil, management, nutrients etc.*

This is completely right. The word "variation" was missing. We revised the sentence to the following:

    "In agricultural production, water and heat extremes are key determinants of yield variations".

215

*- Ln 37-38: "Other metrics of extreme water conditions", please specify.*

We revised the sentence as:

    "While soil moisture is a more appropriate measure of water availability for crops, extreme water indicators based on soil moisture have been only minimally explored".

220

*- Ln 38-39: "Current statistical studies had limited success in statistically capturing the yield response to soil moisture metrics", please explain why.*

We added the following explanation:

> "There are several potential reasons for the limited success of previous statistical studies in capturing yield response to soil moisture. Direct measures of soil water availability include complex biophysical and hydrological processes that are difficult to capture in a rather simple statistical model. On the other hand, seasonal mean soil moisture is highly correlated to seasonal precipitation. Thus, including an average of soil water content may not add value to a statistical model."

225

230 *- Ln 43: "the impact of climate change on soil moisture". The paper is about individual extreme response of yield vs compound. It is not clear why the authors refer to CC studies.*

This is omitted. The climate change section is dropped now, so this sentence is no longer relevant.

*- Ln 46: "conditional marginal impact". Please explain what this means.*

235 See explanation above.

*- Ln 50: please explain "wet-heat stress"*

Wet heat stress or moist heat stress are the terms have been used in different disciplines to talk about hot and humid or moist conditions (Buzan and Huber, 2020). Soil water can exacerbate the heat stress under
240 conditions of high humidity. This is not a prevalent condition. However, it can arise in the context of complex meteorological, hydrological, and agronomic interactions. In the US Midwest, a combination of heatwave and corn sweat can create "moist heat stress" which is dangerous for people, animals, and plants.

280  *- Equation 1: please describe what exactly each letter in the equation refers to? For example please refer last variable in the equation as error and describe g(h) function?*

Thanks for catching this. We have added the description for the missing variables. Here, g(h) is a general function showing the yield growth as function of heat.

285  *- Ln 126: "measure the value of water". Not clear what the authors refer to as "value of water". Please clarify.*

This part is omitted in the revised version.

*- Figure 1: this is nothing new, a known information– like a textbook. Excluding this figure does not change anything about the paper. I recommend not to include it.*

290 We have dropped panel b of the Figure 1. We believe that Figure 1-a illustrates the concepts that are central to the Methods. While illustration itself might look like textbook information, it helps us to define the metrics of soil moisture extremes. To distinguish this from a common-knowledge figure, we have modified it as follows:

[Figure]

**Figure 1. Soil moisture dynamics within a typical growing season. Some soil moisture conditions can be harmful to crops including excess wetness [i], moisture stress intensity[ii], duration of moisture stress [iii], and severity of soil moisture stress [iv]. Normal level of soil moisture is defined as the historical average of volumetric soil moisture within the growing season.**

300 *- Ln 134: "Many researchers have acknowledged the need for soil moisture data to predict the response of crop yields to variations in water availability." Please provide references to those researchers.*

This sentence has been omitted in the revised version.

*- Ln 171: please provide references to those studies.*

305 This sentence has been rephrased and moved to the introduction:

"It has become a standard practice either to focus on a limited geographical area (Rizzo et al., 2018; Wang et al., 2017) or to employ a proxy variable like precipitation, evapotranspiration, or vapor pressure deficit estimates (Comas et al., 2019; Roberts et al., 2013)."

*- This section is too long. Please shorten it and provide detailed information in SI.*

We have substantially revised the organization and transitions within the Methods section. The section is re-
325   organized to focus on the critical parts of the methods and to improve the flow of the paper. Here is the new order:

2.1. Data

2.2 Model (1): individual extremes

2.3 Model (2): compound extremes

330   2.4 Estimation strategy

*- Equation 2: please define each variable and function used in the equation.*

Thanks for pointing to the missing definitions. We have corrected it.

"where $y_{it}$ is the crop yield, $g(h, m)$ is the yield response function to each combination of soil
335   moisture level, $m$, and temperature (heat), $h$; $\varphi(h, m)$ is the distribution of soil moisture and heat; $\overline{m}$ and $\underline{m}$ are maximum and minimum soil moisture; $\overline{h}$ and $\underline{h}$ are maximum and minimum temperature; and $c_i$ is a time-invariant county fixed effect. Here, we do not separate the impact of heat from water. In other words, the marginal impact of heat depends on water; and the marginal impact of water depends on heat."

340

*- Ln 230: "some indicators", please clarify which indicators.*

We have clarified this as:

> "In Model (1-c), we consider the number of days that soil moisture is either too high or too low. The
> model with these metrics of soil moisture extremes further improves the fit, revealing a negative
> 345     marginal relationship associated with the number of days with low/high soil moisture."

*- Please provide numbers to the equations.*

Thanks for your comment. We added equation numbers in the revised version.

350    *- Ln 277: g(Ws), please define the parameter*

This description has been added to the text: "and $g(W_s)$ is 1 for all crops, while it is an exponential function
of soil moisture depth for non-crop soil areas."

*- Ln 290-295: this is a result of the analysis, not related to data/method or assumptions.*

355   Thanks for your comment. We have moved this figure to the Supplementary Material. This information is
important to ensuring that soil moisture is a different metric than precipitation. This information is added,
and the statement re-contextualized and rephrased.

> "In a statistical study, a natural first step is to look at the correlation between these variables. To
> show that mean soil moisture is a different metric than mean precipitation, we have plotted the
> 360     annual mean soil moisture versus annual cumulative precipitation in Fig. S1. This figure is a scatter
> plot for US counties for the growing season from 1981 to 2015. The simple correlation coefficient
> between them is 0.44. This rejects the hypothesis that soil moisture is highly correlated with
> precipitation. As mean precipitation has a linear relationship with cumulative precipitation, the
> results show that mean soil moisture is a different metric than cumulative or mean precipitation."

365

*- Figure 4,5 and 6 are outcomes of the model/analysis. They can be presented in the result section.*

These figures have moved to the results section. We have also added more explanations about the figures
and their messages.

> "The overall simulation results from WBM are illustrated in Fig. 4-6, showing the gridded historical
> 370     mean for the cultivated continental US, average annual variations for the cultivated continental US,
> and bivariate distribution of soil moisture and heat for the corn growing grid cells. To illustrate the
> spatial heterogeneity, Fig. 4 shows the growing season mean soil moisture content (in mm in 1000
> mm topsoil) as calculated based on daily root-zone soil moisture level from Apr-Sep for 1981-2015 at
> 2.5 x 2.5 arcmin grids excluding non-cultivated area. Average growing season soil moisture is

375   heterogeneous across the Continental US, with distinct regional patterns (see Fig. 4). For the corn belt, the soil moisture level is relatively high compared to other regions. The mean of volumetric soil moisture ranges from below 50 mm in southern California to above 250 mm in the Corn Belt and around Mississippi.

380   To compare the variation of simulated soil moisture and precipitation, Fig 4 illustrates the weighted average soil moisture and precipitation over the cultivated US for 1981-2015. In general, variation in soil moisture average is higher than in that of precipitation (Fig. 5), showing how this new water metric is different from previous approaches. One interesting finding is that for some years the mean precipitation and the mean soil moisture move in opposite directions. For example, in 1990 the mean precipitation declined by around 5% while mean soil moisture increased by around 13%.

385   To show the dynamics of soil moisture and heat, Fig. 6 shows their bivariate distribution by month based on daily information for all the cultivated grid cells in the US Corn Belt for 1981-2015. Heat and soil moisture combinations vary through the growing season (Fig. 6) The data shows significant month-to-month variation, with the second half of the season facing hotter and dryer days. Also, July has the highest variation in soil moisture deviation with high probability of compound extremes as
390   the distribution moves toward the lower right. "

*5) Results*

*- Ln 363/364: "We will discuss the implications of these results in Sect. 5." The authors use lots of cross*
395   *references between the sections as seen in here. This is not necessary, since discussion section means discussion of the results by definition. Please through the entire text and remove unnecessary cross-section references.*

Good point. By cutting the length of the manuscript an improved flow of the paper, there is no need to these references. Thus, the superfluous section cross-references have been removed.

400

*- Table 2: note section is repetition of the previous sections, thus it is not necessary.*

The table notes have been removed or shortened for all the Tables.

*- Ln 404: "This indicates that water is up to four times more valuable in hot weather." The authors can*
405   *consider revising the sentence and be more explicit, "value of water" may mean several things.*

As we omitted the value of water section, we have revised this as follows:

"The estimated parameters show the yield response to changes in soil water content. Comparing the parameter values can show the difference in yield response to soil moisture in hot weather and moderate weather…. This indicates that the average yield response to water is up to four times
410   higher in hot weather."

*- Model (2-a) and Model (2-b) were mentioned here for the first time. Please describe the differences between these models in method/data section.*

The Methods section is revised to consider this comment. We have introduced the models in the relevant subsections on the Methods section. Here is the new order:

415         3.1. Model (1): predicting yield responses to individual extremes

        3.2 Model (2): predicting yield responses to compound extremes

        3.3 Model comparison

        3.4 Decomposing the variation in US corn yields

        3.5 Robustness checks

420

*6) Discussion*

*- Ln 410/411: this is related to differences between model 1 & 2, right? Please clarify which model outcome supports (or all?) the statement.*

These lines are omitted. The clarification has been added in subsection 3.2 "Model (2): predicting yield
425   response to compound extremes".

*- Performance: does this mean best correlation between indicators of extreme events yield anomalies? Please clarify.*

For comparing the models, we have Looked at statistical criteria. We have added Table 5 to compare the
430   performance metrics of the models.

**Table 5: Performance metrics for Models 1(a-d) and 2(a-d).**

| Model | Water metric | Extreme metric | R-squared | AIC (Akaike's information criterion) | BIC (Bayesian information criterion) |
|-------|--------------|----------------|-----------|--------------------------------------|--------------------------------------|
| 1-a | Avg. precipitation | Precipitation sqr | 0.469 | -21,238 | -21,201 |
| 1-b | Avg. soil moisture | Soil moisture sqr | 0.471 | -21,612 | -21,576 |
| 1-c | Avg. soil moisture | Number of days with low/high soil moisture | 0.480 | -22,697 | -22,660 |
| 1-d | Avg. soil moisture | Avg soil moisture deficit/surplus | 0.491 | -24,303 | -24,267 |
| 2-a | Avg. soil moisture | T binned by extreme deficit/surplus | 0.492 | -24,402 | -24,328 |
| 2-b | normal soil moisture x T | extreme deficit/surplus x T | 0.501 | -25,582 | -25,509 |

*- First paragraph: what about model 1-c ?*

This section is omitted. We have presented the results from model 1-c in subsection 3.1 " Model (1):
predicting yield response to individual extremes".

> "Regarding Model (1-c), the coefficient on the number of days with low moisture is also significant
> and negative. Our estimation sample shows on average 26 days of high soil moisture and 27 days of
> low soil moisture. The implication is that eliminating 25 days of high soil moisture and 25 days of low
> soil moisture can improve the corn yields by up to 12.6%."

*- Model 2 a-b were not defined in the previous parts of the paper. Please check consistency.*

The Methods section is revised to consider this issue. As mentioned above, we have introduced the models in
the relevant subsections on the Methods section.

> "First, we construct a binning estimator based on daily interaction on heat and soil moisture in model
> (2-a). …. We estimate a coefficient for each combination of excess heat and soil moisture; i.e., we
> estimate a model with metrics of degree days while controlling for soil moisture. The model provides
> the conditional marginal impact of excess heat as:

$$y_{it} = \alpha D_{it}^{10-29} + \left\{ \sum_m \beta_m D_{mit}^{29} \right\} + \delta M_{it} + \delta' M_{it}^2 + \lambda_s t + \lambda_s' t^2 + c_i + \varepsilon_{it} \tag{11}$$

> where $i$ is the county index, $t$ is the time index, $m$ is an index of soil moisture condition (high, low,
> normal), $s$ is an index for states, $y$ is average corn yields, $D$ represents conditional growing degree day
> variables, $M$ shows the seasonal mean soil moisture content, $T$ stands for the time trend variable, $c_i$ is
> a time-invariant county fixed effect. Here, $\beta$ is indexed by $m$; i.e., the marginal impact of heat is
> conditional to soil moisture conditions. $\alpha, \beta, \delta, \lambda$ are the regression parameters showing the marginal
> impacts.

> Second, we estimate a model with metrics of soil moisture while controlling for temperature in
> model (2-b). We define an index of soil moisture when the temperature is above the threshold and
> an index of soil moisture when the temperature is below the threshold. In this model, the soil
> moisture is separated by a temperature threshold $H^*$.

$$y_{it} = \alpha D_{it}^{10-29} + \beta D_{it}^{29} + \left\{ \sum_m \delta_m M_{mit} \big|_{H<H^*} + \delta_m' M_{mit} \big|_{H>H^*} \right\} + \lambda_s t + \lambda_s' t^2 + c_i + \varepsilon_{it} \tag{12}$$

> where $i$ is the county index, $t$ is the time index, $m$ is an index of soil moisture condition, $s$ is an index
> for states, $y$ shows average corn yields, $D$ represents growing degree day variables, $M$ shows
> conditional seasonal mean soil moisture, $t$ stands for the time trend variable, $H$ is the average daily
> temperature, $H^*$ is the temperature threshold, and $c_i$ is a time-invariant county fixed effect. Here, we
> define $\delta$ and $\delta'$ to test whether the marginal impact of soil moisture depends on heat. The soil

moisture metrics are calculated from daily gridded data and aggregated to county and growing season. This includes the index of normal soil moisture (*SM* 0-25+ mm around normal) when $H > H^*$, the index of normal soil moisture when $H < H^*$, the index of moisture deficit (*SM* 25+ mm below normal) when $H > H^*$, index of moisture deficit when $H < H^*$, the index of moisture surplus (*SM* 25+ mm above normal) when $H > H^*$, and the index of moisture surplus when $H < H^*$. *α, β, δ, λ* are the regression parameters showing the marginal impacts. "

*- Ln 416-421: These are newly introduced topics. None of these research goals (including why to have them), methods and results were mentioned in the previous sections of the paper (e.g. new interaction model, why do you have that and this was never mentioned in the paper). It is like Appendix is another paper with its own results, methods and goals. Please revise the paper accordingly.*

We have substantially shortened and revised the Discussions and Appendix sections. The paper has been revised to focus on the main contribution and major messages. Thus, we dropped Model 2-c and 2-d as well as the discussions on "Implications for irrigation water demand and subsurface drainage" and "Implications for climate studies."

*- Ln 424-428: Is this an outcome supported by the results? If so, please indicate how. It is more like a general knowledge.*

This paragraph is omitted. We have revised the discussion section around the advantages of using the metrics of individual and compound extremes.

*- Ln 429-430: Please provide supporting data/result from the analysis.*

We have removed this paragraph as it requires further investigations which are not related to the main message of the paper.

*- Ln 433: what are the other metrics suggested in the literature?*

This section is omitted in the revised version.

*- Ln 434-438: Is this a conclusion related to compound vs individual extreme weather event analysis? Can we say the same if we use other metrics of water stress than soil moisture?*

Thanks for your helpful question. This section is omitted in the revised version. However, we used your suggestion in revising the paper. We focused on comparing models with individual extremes and models with compound extremes. This has improved the flow of the paper and highlighted the significance of this study.

500    *- Ln 465-469: I question that the authors' research is critical for climate change studies. First, their analysis was based on historical data and says nothing about counterfactual analysis. This is not the first time impacts of a compound event was researched and like other studies this paper shows stronger impact of a compound event. It does not bring anything to climate change impact studies.*

We have omitted this subsection in the revised version and briefly talked about it in the revised manuscript.
505    However, we believe that the findings are critical for climate impact studies for several reasons. First, the current literature follows methods like Schlenker and Roberts (2009) by modelling yield response functions looking only at average water conditions. They ignore individual and compound extremes related to water. As we find that the coefficient on heat stress variable is significantly different when considering soil moisture and compound extremes, it is possible that previous climate impact studies have over- or under-estimated
510    the yield impacts of climate change. Second, we are introducing simple but operational metrics of individual and compound extremes that can be constructed using hydroclimatic models for the future. These metrics can improve the prediction of crop yields. We are not aware of any other study suggesting such a simple yet powerful prediction framework.

515    *- Ln 472: please clarify benefit of this collaboration. In which way it helps to solve the challenge.*

We believe that collaboration between hydrologists, climate scientists, and statisticians can improve data generating processes and leads to better models and metrics to help better decisions among people and policymakers. Here is the revised text:

"Applying this framework to climate impact studies will face a key challenge —namely projecting the
520    future compound extremes with the high temporal resolution of Model 2. It requires collaboration between hydrologists, climate scientists, and statisticians (Zscheischler et al., 2020). For future yield projections, we need reliable future projections of daily temperature (maximum and minimum) and soil moisture. Unfortunately, to the best of our knowledge, available data sets including predictions of future soil moisture have a relatively coarse spatial and temporal resolution, and rely on climate
525    model projections with known difficulties representing daily temporal resolution events (Hempel et al., 2013). Further research is required to improve the ability of climate models and impact models in projecting the bivariate distribution of heat-moisture (Sarhadi et al., 2018)."

535    *- Ln 479-ln 483: this recommendation is not related to the sub-section heading. The authors stated a discussion point which is out of scope of their analysis and not supported with the overall goal of the paper. Recommendations can be given to farmers etc; however their model/research is not aimed for decision - support guidance. Please remove this section of revise it.*

Thanks for your comment. We have omitted this part.

540

*- Section 5.4: This section includes literature, method, data and equation related to an estimation. This is not a discussion section. Please previse it accordingly. This additional analysis doesn't bring anything to the value of the paper. I would recommend excluding this analysis from the paper in order to keep its coherence and consistency.*

545     Thanks for your comments which helped to improve the flow of the paper. We have omitted this subsection.

*- Ln 501: "We find that the average damage from excess heat has been up to four times more severe when combined with water stress" what is the damage, yield losses?*

Thanks for your comment. Originally benefits and damages were considered from an economics point of
550     view. In the revised version, we removed the economic analysis of the value of soil moisture. Now we have revised and clarified the sentence as:

>       "Finally, the marginal impact of heat index on crop yields depends on the soil moisture level. We
>       show the average yield damage from heat stress is up to four times more severe when combined
>       with water stress; and therefore the value of water in maintaining crop yield is up to four times larger
555     on hot days."

*- Line 517-525: the CC knowledge and analysis were not included in previous parts (method, data, results) section of the paper. Please include info about this analysis in adequate sections.*

To improve the flow of the paper and reduce the redundancy, the climate change material is omitted.

560

*- Line 525- 535: There is almost no economic analysis thus the paper does not contribute to CC economics. No policy analysis or research were provided either; also paper does not say/bring anything to regional resilience of agroecosystems, global food security, and as well as future climate impacts. These two paragraphs have to be re-written. These claims are bold and cannot be taken from the research as described in the paper.*

565     Thanks for your comment. As we have dropped the subsection, theses paragraphs are also omitted.

**Track-change color guides:**

Inserted text

5    Moved to

[revised manuscript text omitted]

**S.2. A.1. Correlation of mean seasonal soil moisture and other variables**

The soil moisture output from WBM is informed mainly by soil moisture memory, heat, precipitation, and many other time-variant and time-invariant information. In a statistical study, a natural first step is to look at the correlation between these variables. To show that mean soil moisture is a different metric than mean precipitation, we have plotted the annual mean soil moisture versus annual cumulative precipitation in Fig. S2. This figure is a scatter plot for US counties for the growing season from 1981 to 2015. The simple correlation coefficient between them is 0.44. This rejects the hypothesis that soil moisture is highly correlated with precipitation. As mean precipitation has a linear relationship with cumulative precipitation, it shows that mean soil moisture is a different metric than cumulative or mean precipitation.

We have taken two other variables from WBM including soil moisture fraction and evapotranspiration (ET). Also, we have interpolated WBM soil moisture using an alternative method (nearest neighbor method). Section A.6 will provide the estimation results when using these variables to show the robustness of the results to variable selection. Here, we plot these variables against the volumetric soil moisture content to illustrate the correlation and differences. As shown in Fig. S3 A1 two interpolations of soil moisture are closely correlated by R= 0.9997. Figures S4 A2 and S5 A3 are the scatter plots of seasonal ET and seasonal mean soil moisture fraction against volumetric soil moisture. The figures show the seasonal variables are not following a simple linear relationship. Figure S6 A4 shows the scatter plot of cumulative growing degree days above 10°C versus mean soil moisture for US counties for the growing season from 1981 to 2015. This indicates the soil moisture output is not a simple linear transformation of heat data.

**S.3. Are the results different with alternative water metrics? A.6. Robustness check: other metrics from WBM outputs (soil moisture fraction and ET)**

Here, wWe re-estimate Model (1) with other related metrics of water availability to crops including simulated daily evapotranspiration of rainfed corn (ET) from WBM; daily soil moisture fraction (SMF) from WBM; and soil moisture content from different spatial interpolation of WBM grid cells to PRISM (nearest neighbor method versus original bilinear method). The soil moisture fraction index considers the volumetric soil moisture content divided by field capacity. We have also considered the within-season standard deviation of ET and SMF. Note that we keep the degree days above 29°C as an indicator of heat stress and the degree days from 10°C to 29°C as an indicator of beneficial heat to corn.

Table S1A7. reports regression results for these models. Columns 1 and 2 show a significant relationship with the mean of soil moisture fraction, its square term, and its within season standard deviation. Columns 3 and 4 with mean ET and within-season SD of ET also show a significant relationship. Column 5 shows that the other interpolation of soil moisture has a very close marginal coefficient and standard error compared to our original Model (1). The important finding is the marginal relationship for beneficial and harmful heat remains significant and not significantly different from Model (1). Overall, the main findings of the paper remain robust to the choice of alternative seasonal metrics of water availability.

**S.4. Are the estimates different when considering the stages of plant growth? A.5. Robustness check: bi-monthly metrics of soil moisture**

[revised manuscript text omitted]

The results of model (2-b) for Eastern, Western, and whole US are shown in Table S7. As in column (3) of Table S7, the coefficient on normal soil moisture conditional to hot weather is 0.00010. The coefficient on normal soil moisture conditional to moderate weather is 0.00002. This indicates that yield response to water  is up to four times more  in hot weather. The marginal impact on soil moisture deficit index is 0.00008 in hot weather and is 0.00002 in moderate weather. This also supports the finding that the yield response to water is up to four times more  in hot weather. Also, the results suggest that the damage from excess water is up to two times bigger in hot weather.

1310

**Table A1. Corn yield estimation controlling for normal soil moisture**

[revised manuscript text omitted]

Standard errors are in parenthesis
*** p<0.01, ** p<0.05, * p<0.1

1335    **Notes: Table lists regression coefficients and shows standard errors in brackets. Temperature is in degree Celsius and soil moisture in mm in 1000 mm topsoil. The soil moisture is obtained from WBM at 6 arcmin output while precipitation and temperature are taken from PRISM at 2.5 arcmin. They are aggregated from grid cells to counties based on crop area weight. Yield data is acquired from the USDA. The constant term and coefficients on the interaction of each state and time trends are not reported.**

Table S4.A4. Estimation of Model (1) for the East

| | (1-a) Log CornYield | (1-b) Log CornYield | (1-d') Log CornYield | (2-c) Log CornYield |
|---|---|---|---|---|
| Degree days from 10°C to 29°C | .0003108*** (.0000936) | .0003152*** (.0000868) | .0003072*** (.0000724) | .0002308** (.000088) |
| Degree days above 29°C | -.0056293*** (.0007259) | -.0054707*** (.0007343) | -.0052882*** (.0006442) | -.0056523*** (.000946) |
| Cumulative precipitation Apr-Sep (mm) | .0009245*** (.0002502) | | | |
| Square of cumulative precipitation Apr-Sep | -7.000e-07*** (2.000e-07) | | | |
| Mean daily soil moisture content (mm) | | .00319*** (.0006763) | | |
| Square of mean daily soil moisture content | | -.0000158*** (3.000e-06) | | |
| Index of extreme deficit | | | .0000379*** (5.700e-06) | .0000183*** (5.300e-06) |
| Index of extreme surplus | | | -.0000381*** (2.700e-06) | -.0000225*** (6.800e-06) |
| Index of normal soil moisture | | | .0000292** (.0000112) | .0000433*** (.0000107) |
| Degree days from 10°C to 29°C x S1 | | | | .0001296** (.00006) |
| Degree days above 29°C x S2 | | | | .0010785 (.000888) |
| Observations | 62094 | 62094 | 62094 | 62094 |
| R-squared | .4997799 | .4989592 | .5205428 | .5277292 |
| Akaike's Crit | -20126.6 | -20024.8 | -22756.9 | -23690.7 |
| Bayesian Crit | -20090.4 | -19988.6 | -22711.8 | -23627.5 |

Standard errors in parenthesis
*** p<0.01, ** p<0.05, * p<0.1

**Notes: Table lists regression coefficients and shows standard errors in brackets. Model (1-d') is slightly different from Model (1-d) considering extreme deficit and extreme surplus metrics. Temperature is in degree Celsius and soil moisture in mm in 1000 mm topsoil. The soil moisture is obtained from WBM at 6 arcmin output while precipitation and temperature are taken from PRISM at 2.5 arcmin. They are aggregated from grid cells to counties based on crop area weight. Yield data is acquired from the USDA. The constant term and coefficients on the interaction of each state and time trends are not reported.**

**Table S5. Estimation of the Model (1) for the West**

|  | (1-a) Log CornYield | (1-b) Log CornYield | (1-d') Log CornYield |  |
|---|---|---|---|---|
| Degree days from 10°C to 29°C | .0004426*** (.0000829) | .0004484*** (.0000823) | .0004539*** (.0000862) |  |
| Degree days above 29°C | -.0020381*** (.000423) | -.0023744*** (.0004911) | -.0022938*** (.0004752) |  |
| Cumulative precipitation Apr-Sep (mm) | .0005768 (.0003372) |  |  |  |
| Square of cumulative precipitation Apr-Sep | -3.000e-07 (5.000e-07) |  |  |  |
| Mean daily soil moisture content (mm) |  | .0078908** (.0027432) |  |  |
| Square of mean daily soil moisture content |  | -.0000848** (.0000326) |  |  |
| Index of extreme deficit |  |  | .0000255 (.0000271) |  |
| Index of extreme surplus |  |  | -9.800e-06 (7.600e-06) |  |
| Index of normal soil moisture |  |  | .0000762** (.0000309) |  |
|  |  |  |  |  |
|  |  |  |  |  |
| Observations | 7829 | 7829 | 7829 |  |
| R-squared | .2784229 | .2768284 | .2772401 |  |
| Akaike's Crit | -3050.8 | -3033.5 | -3035.9 |  |
| Bayesian Crit | -3022.9 | -3005.6 | -3001.1 |  |

Standard errors are in parenthesis
*** p<0.01, ** p<0.05, * p<0.1

**Notes: Table lists regression coefficients and shows standard errors in brackets. Model (1-d') is slightly different from Model (1-d) considering extreme deficit and extreme surplus metrics.** ~~**Temperature is in degree Celsius and soil moisture in mm in 1000 mm topsoil. The soil moisture is obtained from WBM at 6 arcmin output while precipitation and temperature are taken from PRISM at 2.5 arcmin. They are aggregated from grid cells to counties based on crop area weight. Yield data is acquired from the USDA. The constant term and coefficients on the interaction of each state and time trends are not reported.**~~

1350

**Table S6.A8. West versus East in Corn yield estimation with the interaction of heat and soil moisture (Model 2-a)**

| | (US)
log
CornYield | (West)
log
CornYield | (East)
log
CornYield |
|---|---|---|---|
| Degree days from 10˚C to 29˚C | .0003083***
(.0000685) | .0004344***
(.0000847) | .0002963***
(.0000736) |
| dday29˚C & SM 75+ mm below normal (extreme deficit) | -.0082398***
(.0014372) | -.0074467*
(.0035727) | -.0082928***
(.0014365) |
| dday29˚C & SM 25-75 mm below normal (deficit) | -.0062069***
(.0009793) | -.0033152*
(.001627) | -.0061966***
(.0009797) |
| dday29˚C & SM 0-25 mm around normal (normal) | -.0037559***
(.0004045) | -.0024412***
(.0005053) | -.0041335***
(.0004376) |
| dday29˚C & SM 25-75 mm above normal (surplus) | -.0055709***
(.0012041) | -.004754*
(.0024763) | -.005625***
(.0011677) |
| dday29˚C & SM 75+ mm above normal (extreme surplus) | -.0140295***
(.0019083) | .0095881
(.0128016) | -.0143573***
(.0018101) |
| Mean daily soil moisture content (mm) | .0026635***
(.0008153) | .0080027**
(.0028858) | .0025636***
(.0008324) |
| Square of mean daily soil moisture content | -.0000161***
(2.600e-06) | -.0000844**
(.0000326) | -.0000156***
(2.600e-06) |
| Observations | 69923 | 7829 | 62094 |
| R-squared | .4921263 | .2777862 | .5149811 |
| Akaike's Crit | -24401.6 | -3035.9 | -22034.8 |
| Bayesian Crit | -24328.3 | -2980.2 | -21962.5 |

Standard errors in parenthesis
*** p<0.01, ** p<0.05, * p<0.1

**Notes: Table lists regression coefficients and shows standard errors in brackets.** ~~Temperature is in degree Celsius and soil moisture in mm in 1000 mm topsoil. The soil moisture is obtained from WBM at 6 arcmin output while precipitation and temperature are taken from PRISM at 2.5 arcmin. They are aggregated from grid cells to counties based on crop area weight. Yield data is acquired from the USDA. The constant term and coefficients on the interaction of each state and time trends are not reported.~~

1355

**Table S7.A9. West versus East in Eestimation of corn yields while splitting the soil moisture indicators (Model 2-b)**

|  | (US) log CornYield | (West) log CornYield | (East) log CornYield |
|---|---|---|---|
| Degree days from 10˚C to 29˚C | .0003154*** | .0004451*** | .0002983*** |
|  | (.0000689) | (.0000919) | (.000074) |
| Degree days above 29˚C | -.004044*** | -.0020707*** | -.0044516*** |
|  | (.0005384) | (.0005793) | (.0005981) |
| Index of normal soil moisture when T > T* | .0001199*** | .0001805 | .0001034*** |
|  | (.0000342) | (.0001426) | (.0000358) |
| Index of extreme moisture surplus when T > T* | -.0000628*** | -.0001173 | -.0000586*** |
|  | (.0000151) | (.0001071) | (.0000149) |
| Index of extreme moisture deficit when T > T* | .000092*** | -.0000526 | .0000817*** |
|  | (.0000234) | (.0000978) | (.0000229) |
| Index of extreme moisture deficit when T < T* | .0000209*** | .0000287 | .0000223*** |
|  | (7.100e-06) | (.0000337) | (7.000e-06) |
| Index of extreme moisture surplus when T < T* | -.0000326*** | -5.700e-06 | -.0000334*** |
|  | (3.200e-06) | (6.500e-06) | (3.200e-06) |
| Index of normal soil moisture when T < T* | .000028** | .000063** | .0000247** |
|  | (.0000105) | (.0000249) | (.0000102) |
| Observations | 69923 | 7829 | 62094 |
| R-squared | .5006312 | .2782242 | .5262193 |
| Akaike's Crit | -25582.4 | -3040.6 | -23490.5 |
| Bayesian Crit | -25509.2 | -2984.9 | -23418.2 |

Standard errors in parenthesis
*** p<0.01, ** p<0.05, * p<0.1

**Notes: Table lists regression coefficients and shows standard errors in brackets.** ~~Temperature is in degree Celsius and soil moisture in mm in 1000 mm topsoil. The soil moisture is obtained from WBM at 6 arcmin output while precipitation and temperature are taken from PRISM at 2.5 arcmin. They are aggregated from grid cells to counties based on crop area weight. Yield data is acquired from the USDA. The constant term and coefficients on the interaction of each state and time trends are not reported.~~

1360

1365

**Figure** S12. Soil texture affects normal moisture levels. The sandy soil has the lowest normal level while the clay has the highest normal levels.
1370

[Figure]

1375 **Figure S2.9. WBM mean soil moisture versus PRISM cumulative precipitation for 1981-2015 by US counties.**

[Figure]

**Figure S3.A1.** County-level mean seasonal soil moisture based on bilinear interpolation versus alternative interpolation (nearest-neighbor) from WBM 6 arcmin grids to PRISM 2.5 arcmin resolution for the 1981-2015 period.

1380

[Figure]

**Figure S4. County-level mean soil moisture versus mean ET aggregated from WBM for the 1981-2015 period.**

[Figure]

1385    **Figure S5.A3. County-level mean volumetric soil moisture content versus mean of soil moisture fraction aggregated from WBM for the 1981-2015 period.**

[Figure]

**Figure S6.A4. County-level seasonal mean soil moisture versus seasonal heat index aggregated from WBM and PRISM for the 1981-2015 period.**

---

## Author Response (AR3)

**Final responses "Quantifying the Impacts of Compound Extremes on Agriculture and Irrigation Water Demand" by Iman Haqiqi et al.**

**Comments and Responses to Anonymous Referee #2**

**Overall:**

5    **The paper has been significantly improved. However, it still needs further refinement.**

The authors would like to thank anonymous referee #2 for his/her helpful comments and for acknowledging the improvements made.

**My major comments are:**

10    **1- The first two sections (introduction and method) are still chaotic. It is not easy to follow, many bits and pieces were presented without a flow/structuring. This makes it difficult to read and capture the essence.**

These two sections are revised substantially to improve the flow of the paper as described below.

**2- On the contrast, the results, discussion and conclusion sections were really neatly written/revised. The**
15    **difference in writing, organization/structure and clarity between those two parts is very obvious. I think the authors can further improve the first two sections as they did for the last three.**

Thanks for acknowledging the improvements of the Discussion and Conclusion sections.

**Introduction:**

**3- Line 19: the starting sentence to the article is awkward. I recommend the authors first guide the reader**
20    **through the subject and later define what they did. Also, please use past tense, not present, in describing what was done. I also recommend not to use "we" type of personal pronouns.**

a)    This paragraph has been revised. So, the paper starts with an introduction to the subject.

b)    Also, the manuscript is revised to employ the past tense for describing what was done.

c)    As advised, the use of first person is avoided in most of the parts of the revised manuscript.

25

**4- Line 33: this is a repetition. The reader can understand what was written before, peer understanding.**

This sentence has been removed.

**5- The use of language in telling the narrative in the introduction section is confusing. It is like a combined bits**
30    **and pieces but lacks flow of a narrative. This section still needs a thorough (not grammatical) editing to detect**

**such issues. There are repetitive sentences and statements. The first paragraph can be divided into two or three as well.**

The introduction section is revised substantially in an attempt to provide a better flow. The repetitions are eliminated and the first paragraph is shortened.

35

**6- Line 52: please avoid using "we", and who do you mean by "we"?**

The authors tried to use third person sentences in the revised manuscript.

**7- After line 52: the authors in this revised form again starts the reader education about "textbook" information.**
40 **I don't see why any reader would like to read such information. If someone is interested in fundamental agricultural soil moisture relations, they can read a book. In this form, the article is still too long and many unnecessary statements.**

The paragraph is shortened in the revised manuscript trying to avoid the use of textbook information.

45 **8- "Beneficial heat is less beneficial without sufficient soil moisture. On the other hand, soil moisture is not beneficial without sufficient heat for plant growth. Harmful heat can be less harmful when there is enough soil" These statements are like taken from a guidebook.**

This sentence is removed.

50 **9- The authors are telling us limitations of previous studies a lot, however no insight about if their study response to those limitations.**

The paragraph on the research gap is shortened. Only relevant limitations are discussed to show how the current study fills the gap.

55 **10- Line 70: this is a conclusion statement!!!**

This sentence is removed.

**Methods:**

**11- First sentence: how do the readers supposed to know what the referred studies did and what they are**
60 **about? Please revise this section.**

This section is revised. The revised starting paragraph of the Methods section provides sufficient information about those studies.

**12- Line 76-78: these two sentences are not needed. The information given after is enough for the reader.**

This sentence is removed.

**13- First paragraph: according to the author's description, the 3rd option is conditional, please correct it.**

This sentence is removed. Instead, the metrics are introduced in the data processing section.

**14- Line 88: which metrics?**

This sentence is removed. The compound metrics are introduced in the data processing and model sections.

**15- Line 98: the section number should be 2.1.1, not 3.3.1, please correct all section numbering**

The section numbers are corrected in the revised manuscript.

**16- In data section, many equations were provided. This section is a mix of data, assumption and method. Please either change the section heading or only provide data sources. Furthermore, this section is too detailed for the main text, most of it can be taken to SI.**

The Data section is divided into two parts. Section 2.1 describes the data sources and section 2.2 explains the data processing. Also the soil moisture Module is moved to the SI.

**17- Estimation strategy section can be taken from here and described in SI. Same for decomposition method.**

These sections are moved to the SI.

**18- It is very challenging for the reader to follow the content in this section. . The authors can re-organize this section by first describing the models used, their differences, input data resources. Then provide fundamental equations. The rest can be provided as SI. An overview of the models, what they refer to, major differences, metrics used, source of data can be provided in a table so that reader can follow easily the results by checking what has been done in each modelling effort. In the current, flow of the text is confusing.**

The Methods section is shortened and reorganized. As mentioned above, the soil moisture module, estimation strategy, and decomposition methods are moved to the SI. Further, Table 1 is revised to show an overview of the models and the major metrics used in each model.

**Results:**

**19- Line 280: Figure 2 doesn't show precipitation.**

Thanks for catching this. It is corrected to Fig 3.

**20- This section was improved significantly and nicely written.**

Thanks for acknowledging the improvement of the Results section.

**Discussion:**

**21- Line 397: is it a comma or a stop?**

Thanks for catching this mistake. It should be a stop.

**Conclusions:**

**22- Nicely written.**

Thanks for acknowledging the improvement of the Conclusion section.

---

## Author Response (AR4)

**"Quantifying the Impacts of Compound Extremes on Agriculture and Irrigation Water Demand" by Iman Haqiqi et al.**

**Responses to the Editor**

Overall response: We would like to thank the Editor and reviewers for their positive comments about the relevance and the significance of this work. Their detailed suggestions throughout the review process have helped us to improve the paper significantly. Following their comments also helped us to enhance the clarity, flow, and structure of the manuscript. Below, final corrections are described in response to the "minor revisions".

**Comment: The panels in Figure 4 are preferably ranked from left to right (instead of top to bottom) to make the time progression more clearly visible.**

Response: Thanks for your suggestion. We agree that the left to right ranking would better reflect the time. So, the revised Figure 4 provides "April, May, June" at the top and then "July, August, September" at the bottom.

**Comment: And in line 322 a crucial typo is found (the period should be a comma, I believe)**

Response: Thanks for catching this typo. This should be a comma. It is corrected in the revised manuscript.